# UBR4/POE facilitates secretory trafficking to maintain circadian clock synchrony

Sara Hegazi [1,2], Arthur H. Cheng [1,2], Joshua J. Krupp [1], Takafumi Tasaki [3,4], Jiashu Liu[1,2], Daniel A. Szulc [5,6], Harrod H. Ling[1,2,10], Julian Rios Garcia[1,2], Shavanie Seecharran [1,2], Tayebeh Basiri[7], Mehdi Amiri[7], Zobia Anwar[1,2], Safa Ahmad[1], Kamar Nayal[1,2], Nahum Sonenberg[7], Bao-Hua Liu[1,2], Hai-Ling Margaret Cheng [5,6,8], Joel D. Levine [1,2,9 ✉] & Hai-Ying Mary Cheng [1,2 ✉]

Ubiquitin ligases control the degradation of core clock proteins to govern the speed and resetting properties of the circadian pacemaker. However, few studies have addressed their potential to regulate other cellular events within clock neurons beyond clock protein turnover. Here, we report that the ubiquitin ligase, UBR4/POE, strengthens the central pacemaker by facilitating neuropeptide trafficking in clock neurons and promoting network synchrony. *Ubr4*-deficient mice are resistant to jetlag, whereas *poe* knockdown flies are prone to arrhythmicity, behaviors reflective of the reduced axonal trafficking of circadian neuropeptides. At the cellular level, *Ubr4* ablation impairs the export of secreted proteins from the Golgi apparatus by reducing the expression of Coronin 7, which is required for budding of Golgi-derived transport vesicles. In summary, UBR4/POE fulfills a conserved and unexpected role in the vesicular trafficking of neuropeptides, a function that has important implications for circadian clock synchrony and circuit-level signal processing.

[1] Department of Biology, University of Toronto Mississauga, Mississauga, ON L5L 1C6, Canada. [2] Department of Cell and Systems Biology, University of Toronto, Toronto, ON M5S 3G5, Canada. [3] Division of Protein Regulation Research, Medical Research Institute, Kanazawa Medical University, Uchinada, Ishikawa 920-0293, Japan. [4] Department of Medical Zoology, Kanazawa Medical University, Uchinada, Ishikawa 920-0293, Japan. [5] Institute of Biomedical Engineering, University of Toronto, Toronto, ON M5S 3G9, Canada. [6] Ted Rogers Centre for Heart Research, Translational Biology & Engineering Program, University of Toronto, Toronto, ON M5G 1M1, Canada. [7] Department of Biochemistry, Goodman Cancer Research Center, McGill University, Montreal, QC H3A 1A3, Canada. [8] The Edward S. Rogers Sr. Department of Electrical & Computer Engineering, University of Toronto, Toronto, ON M5S 3G4, Canada. [9] Department of Ecology and Evolutionary Biology, University of Toronto, Toronto, ON M5S 3B2, Canada. [10]Deceased: Harrod H. Ling. ✉email: joel.levine@utoronto.ca; haiying.cheng@utoronto.ca

Circadian clocks evolved to enable organisms to align their physiology and behavior to cyclical changes in their environment. This timekeeping mechanism in most multicellular eukaryotes is based on transcription–translation feedback loops (TTFLs) that oscillate autonomously with a ~24 h period and can synchronize with the day-night cycle. In general, elements of the positive limb of the primary TTFL (BMAL1 and CLOCK in mammals; CYC and CLOCK in *Drosophila*) drive the transcription of elements of the negative limb (*Per* and *Cry* in mammals; *per* and *tim* in *Drosophila*), whose protein products eventually feedback to auto-repress their gene expression[1]. The molecular clockwork is present in many cells, including the central pacemaker in the brain, which orchestrates circadian rhythms throughout the body in addition to receiving and responding to photic temporal cues. The neurons that comprise the central clock—the suprachiasmatic nuclei (SCN) in mammals or the clusters of dorsal and lateral neurons in *Drosophila*—oscillate synchronously as a result of secreted neuropeptides that couple clock cells together, thereby enabling circuit-level pacemaking[2–8].

In addition to the clock proteins that comprise the TTFLs, E3 ubiquitin ligases have been implicated in the regulation of circadian timekeeping. E3 ubiquitin ligases recognize protein substrates and promote their ubiquitination and consequent degradation by proteasomes[9,10]. Several E3 ligases have been identified that control the turnover of clock proteins, a process that is critical for determining the period of circadian oscillations and the ability of the clock to reset to light cues[1,11,12]. Recently, we identified ubiquitin protein ligase E3 component N-recognin 4 (UBR4) as a clock- and light-regulated protein in the murine SCN[13]. UBR4 is a UBR protein of the N-degron pathway (formerly known as the N-end rule pathway), a conserved, ubiquitylation-dependent proteolytic system that relies on the recognition of N-terminal degradation signals, or N-degrons, by UBR proteins[14–17]. Of the seven known UBRs in mammals, only four—UBR1, UBR2, UBR4, and UBR5—are known to be involved in the N-degron pathway[15]. UBR4 has been implicated in such diverse processes as proteasomal degradation, autophagy, apoptosis, membrane morphogenesis, yolk sac development, neuronal migration, muscle hypertrophy, and virus budding[18–24]. However, the molecular and cellular functions of UBR4 remain poorly understood, due in part to the fact that, unlike most UBRs, UBR4 lacks a canonical ubiquitylation domain and thus may act in a manner that is novel to other members of the N-degron pathway[15].

Here we examine the function of UBR4 in the circadian timing systems of mice and flies and generalize our findings to its role in cellular physiology. Mice deficient for *Ubr4* in the SCN were prone to the desynchronizing effects of constant light and exhibited faster recovery from jetlag, whereas flies in which the orthologous gene, *purity of essence* (*poe*), was silenced in clock neurons were susceptible to behavioral arrhythmicity. Unexpectedly, in both mice and flies, trafficking of neuropeptides important for clock network synchrony was impaired, resulting in their accumulation in the cell body. Using a human kidney cell line lacking UBR4, we further showed that UBR4 is required for the timely exit of secretory cargo from the Golgi complex, mirroring our findings in animal models. Quantitative proteomic analysis revealed Coronin 7 (CRN7) to be a candidate mediator of the trafficking defect in *Ubr4*-deficient cells. Restoring CRN7 expression in *Ubr4*-ablated cells rescued the impairment in Golgi export. Collectively, our findings identify UBR4 as a key regulator of protein trafficking through the biosynthetic-secretory pathway, a function that is conserved across multiple species (human, mouse, fly) and impacts the circadian timing system by disrupting neuropeptide-mediated communication.

## Results

### Ablation of UBR4 in GABAergic neurons impairs activity rhythms under constant light

To investigate the role of UBR4 in the murine SCN, we conditionally ablated the *Ubr4* gene in GABAergic neurons by breeding *Ubr4*[fl/fl] mice, which possess loxP sites flanking exon 1, with those that express the Cre recombinase under the control of the vesicular GABA transporter (*Vgat*) gene (*Vgat*-Cre). Given that all SCN neurons are GABAergic, the resulting *Vgat-cre;Ubr4*[fl/fl] mice (hereafter referred to as *Ubr4* cKO) are devoid of *Ubr4* expression in neurons throughout the SCN, as shown by in situ hybridization (Fig. 1a). Western blot analysis confirmed the absence of UBR4 protein of the expected size, ~570 kDa, in the SCN of *Ubr4* cKO mice (Fig. 1b). Interestingly, several bands between 100 and 400 kDa that were present in *Ubr4* wild-type (control) SCN were not observed in the *Ubr4* cKO samples (Fig. 1b), suggesting the existence of multiple protein isoforms of UBR4.

Next, we examined the effects of *Ubr4* ablation on circadian rhythms of behavior by analyzing wheel-running activities of *Ubr4* cKO mice and control animals (*Ubr4*[fl/fl] and *Vgat*[cre/+] mice) under a fixed 12-h light:12-h dark (12:12 LD) schedule followed by constant darkness (DD) (Fig. 1c). *Ubr4* cKO mice were similar to controls with respect to period, amplitude, and activity levels under LD and DD (Fig. 1c–f and Supplementary Table 1). Based on our previous observation that UBR4 is a light-inducible protein[13], we assessed the ability of our mutant mice to acutely reset their clock phase in response to nocturnal light (Fig. 1c). A brief light pulse in the early subjective night (circadian time [CT] 15) triggered a phase delay of similar magnitude between *Ubr4* cKO mice and controls (Fig. 1g). These data indicate that UBR4 is dispensable for light-induced phase shifts and period length determination under DD.

To test whether UBR4 is required for the disruptive and period-lengthening effects of constant light (LL), we exposed mice to a long-term LL paradigm in which the light intensity was increased in a stepwise fashion every 2–3 weeks (5, 10, 20, 40, 80, and 120 Lux) (Fig. 1h). Only 64% (7/11) of *Ubr4* cKO mice were rhythmic at 80 Lux, in contrast to control mice that were all rhythmic (11/11) at that light intensity (Supplementary Table 1). Rhythmic control and *Ubr4* cKO mice displayed increasingly longer free-running rhythms as the light intensity increased (Fig. 1h, i). However, the period of *Ubr4* cKO mice under 10, 20, and 40 Lux LL was significantly shorter than that of controls (Fig. 1h, i). The amplitude of rhythms and daily activity levels in LL were also reduced in *Ubr4* cKO mice (Fig. 1h, j–l). Collectively, these data suggest that UBR4 buffers the SCN clock against the disruptive effects of constant light.

### The absence of UBR4 in the murine SCN promotes entrainment to acute and chronic jetlag

Behavioral rhythmicity under LL has been shown to be positively correlated with intercellular synchrony in the SCN[25]. Hence, the LL-induced behavior of *Ubr4* cKO mice may be a consequence of altered SCN synchrony, which should also affect their ability to entrain to jetlag schedules. When subjected to an abrupt 7-h advance or delay of the LD schedule, both *Ubr4* cKO and control mice were able to re-entrain to the shifted cycle (Fig. 2a). However, *Ubr4* cKO mice re-entrained to an advanced LD schedule at an accelerated rate when compared to controls, requiring fewer days to reach stable entrainment (Fig. 2b, c). In contrast, the kinetics of re-entrainment to a delayed LD schedule were not different between genotypes (Fig. 2d, e).

Next, we investigated the consequence of *Ubr4* ablation on an animal's ability to entrain to repeated shifts of the LD schedule. The ChrA[6/2] chronic jetlag paradigm, in which the LD schedule is

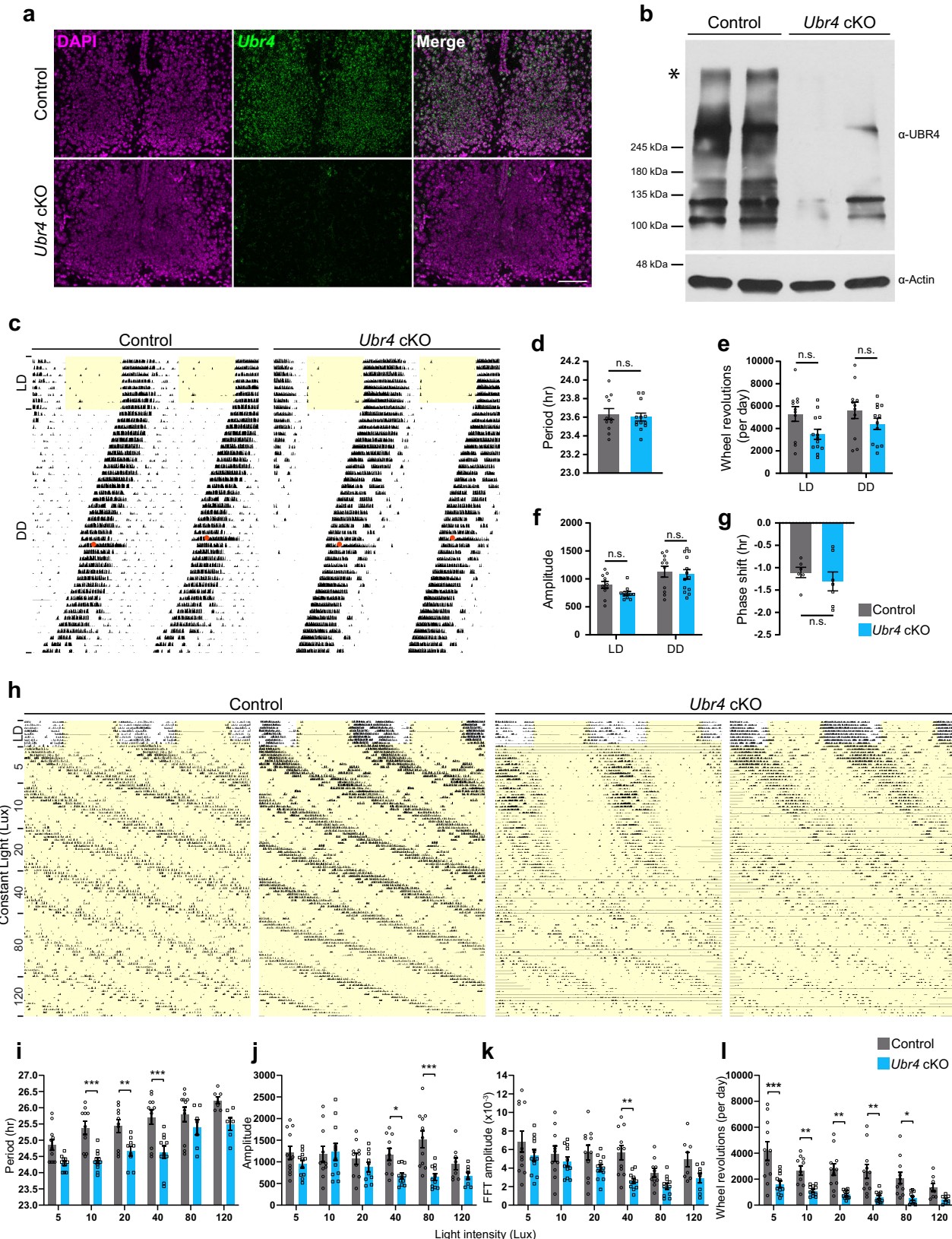

advanced by 6 h every 2 days, has been shown to trigger a state of internal desynchrony in a subset of C57Bl/6 mice, resulting in the emergence of two activity rhythms: a short-period component of ~21 h that signifies entrainment to the LD schedule and a long-period component of >24 h that is not entrained[26]. Under the ChrA[6/2] protocol, 44% (4/9) of control mice displayed a single, entrained component with a period of 20.98 ± 0.02 h, and the remaining 56% (5/9) showed desynchronized behavior with two rhythmic components (period: 21.00 ± 0.00 h, 24.55 ± 0.20 h) (Fig. 2f, g). In contrast, 85% (11/13) of Ubr4 cKO mice had a single, entrained component (period: 21.00 ± 0.00 h); the remaining 15% (2/13) were desynchronized but with a "long-period"

**Fig. 1 Ablation of UBR4 in GABAergic neurons impairs activity rhythms under constant light. a** In situ hybridization analysis of *Ubr4* (green) mRNA expression in the SCN of control and *Ubr4* cKO mice. DAPI, magenta. Scale bar, 100 μm. **b** Western blots of UBR4 and actin in SCN lysates from control and *Ubr4* cKO mice. Full-length UBR4 (~570 kDa), asterisk. **c**, **h** Representative actograms displaying wheel-running activities of control and *Ubr4* cKO mice under LD and DD conditions (**c**), or under LL conditions where the light intensity was increased in a stepwise fashion (**h**). CT 15 light pulse, orange circle (in **c**). Periods of light are shaded in yellow. **d** Period length under DD. **e**, **f** Daily wheel-running activity (**e**) and $X^2$ periodogram amplitude (**f**) under LD and DD. **g** Phase shift following a CT15 light pulse. **i–l** Period length (**i**), $X^2$ periodogram amplitude (**j**), Fast Fourier Transformation (FFT) amplitude (**k**), and daily wheel-running activity (**l**) under LL as a function of light intensity. Values represent mean ± SEM. Exact sample size and *p*-value for all behavioral measures are provided in Supplementary Table 1. *$p < 0.05$, **$p < 0.01$, ***$p < 0.001$ vs. control by two-tailed unpaired *t* test (**d**, **g**) or linear mixed-effects modeling with Bonferroni's post hoc (**e**, **f**, **i–l**). n.s. not significant.

---

component that was less than 24 h (period: 21.00 ± 0.00 h, 23.42 ± 0.34 h) (Fig. 2f, g). The ChrA^6/2 paradigm did not trigger a change in the free-running period of *Ubr4* cKO mice relative to controls upon their release into DD (Fig. 2h). However, the increase in circadian amplitude observed in control mice following exposure to the ChrA^6/2 paradigm was absent in *Ubr4* cKO animals (Fig. 2i). Altogether, these data strongly indicate that the absence of UBR4 in the SCN increases the resistance to jetlag.

**UBR4 is essential for the proper trafficking of VIP and AVP in the SCN and for mPER2 expression**. SCN neurons secrete a variety of neuropeptides within the tissue to regulate intercellular communication and synchrony[8]. The unusual circadian behavior of *Ubr4* cKO mice suggested a potential defect in the synchronization process and motivated us to examine the expression of two prominent neuropeptides in the SCN. Vasoactive intestinal neuropeptide (VIP) is synthesized by ventrolateral (core) SCN neurons, which send dense projections to the dorsomedial (shell) SCN region (Fig. 3a)[27,28]. In contrast, shell SCN neurons express arginine vasopressin (AVP) and project primarily to neurons within the same compartment (Fig. 3a)[27,28]. Analysis of VIP and AVP immunoreactivity (IR) revealed no difference in mean expression intensity in the whole SCN of *Ubr4* cKO mice relative to controls (Fig. 3b–d, i). However, the spatial distribution of both neuropeptides was noticeably altered in the mutant SCN in a manner that was independent of time-of-day (Fig. 3b, c). In control mice, VIP-IR was most prominent in the axons projecting to the shell, in contrast with *Ubr4* cKO mice in which VIP-IR was highly concentrated in the cell soma situated in the core SCN (Fig. 3b). These distribution patterns were reflected in the mean expression intensity measurements in the core and shell regions, as well as the number of cell bodies expressing high levels of VIP (Fig. 3e–h). Similarly, *Ubr4* cKO mice demonstrated intense AVP-IR in the cell soma along with a greater number of cell bodies with high AVP expression (Fig. 3c, j–l). Importantly, these effects could not be attributed to increased expression at the transcript level, as *Vip* mRNA was unaltered and *Avp* mRNA was actually reduced in the SCN of *Ubr4* cKO mice (Supplementary Fig. 1a). These data suggest that UBR4 is critical for the proper trafficking of VIP and AVP neuropeptides within SCN neurons.

To determine whether *Ubr4* ablation impacts the molecular clock, we analyzed the circadian expression profile of PERIOD2 (PER2) protein in the SCN of *Ubr4* cKO and control mice (Fig. 3m). Relative to controls, *Ubr4* cKO mice exhibited damped oscillations of PER2, as evident by the reduction in mean PER2-IR intensity and the number of cells with high PER2 expression in the SCN at the peak of the rhythm (Fig. 3n, o). Along these lines, knocking down *Ubr4* expression in murine neuroblastoma (Neuro-2a) cells also reduced the abundance of ectopically expressed V5-tagged PER2 (Supplementary Fig. 1b). However, these differences at the protein level are not due to altered transcription or mRNA stability, as mRNA abundance of *Per2*, as well as of other clock genes, in the SCN at CT 9 was comparable between *Ubr4* cKO and control mice (Supplementary Fig. 1a).

Collectively, these findings reveal that UBR4 is crucial for neuropeptide trafficking within the SCN and modulation of PER2 protein rhythms. These functions may underlie the resistance of *Ubr4* cKO mice to experimental jetlag and their susceptibility to the disruptive effects of constant light.

**Poe knockdown in Drosophila melanogaster clock neurons impairs behavioral rhythms under constant darkness**. To determine whether UBR4 has a conserved function in circadian rhythms that extends beyond mammals, we investigated the effects of knocking down *purity of essence* (*poe*), the *Drosophila* homolog of UBR4, on invertebrate clock timing. *Poe* is widely expressed in the adult fly brain including the main pacemaker neurons, the pigment-dispersing factor (PDF)-positive small and large ventral lateral neurons (s-LN$_v$ and l-LN$_v$, respectively) (Fig. 4a and Supplementary Fig. 3a). To assess the function of *poe* in circadian timekeeping, we used the clock-specific drivers, *timeless (tim)*-GAL4 and *Pdf*-GAL4, to induce the expression of *poe^RNAi* either in all clock neurons or specifically in the s-LN$_v$ and l-LN$_v$ neurons, respectively. The efficacy of the *poe^RNAi* line that we used in knocking down *poe* expression was confirmed by in situ hybridization analysis of s-LN$_v$ and l-LN$_v$ neurons from *Pdf>Dcr2; poe^RNAi* flies and by reverse transcription–quantitative polymerase chain reaction (qRT-PCR) analysis of whole-brain tissues from flies with pan-neuronal expression of *poe^RNAi* using the *elav*-GAL4 driver (Fig. 4a, b).

We assessed the consequence of *poe* knockdown (KD) in either *tim*- or *Pdf*-positive clock cells on circadian rhythms of locomotor activity. Under a 12:12 LD schedule, *tim>Dcr2; poe^RNAi* flies exhibited advanced evening and no morning anticipatory behavior (Supplementary Fig. 2a–d). Upon release into DD, all *tim>Dcr2; poe^RNAi* flies showed complete and immediate arrhythmicity, which was further indicated by a reduction in their rhythm strength when compared to *UAS-poe^RNAi* (+>*poe^RNAi*) and *tim>Dcr2* controls (Fig. 4c, d, Supplementary Fig. 2a, and Supplementary Table 2). In contrast, *Pdf>Dcr2; poe^RNAi* flies displayed multiple phenotypes under free-running conditions (Fig. 4e and Supplementary Fig. 2e). The majority (72%) of *Pdf>Dcr2; poe^RNAi* flies, which we have termed "delayed arrhythmic," remained rhythmic in the first ~3–6 days of DD before transitioning to complete arrhythmicity (Fig. 4e, f). This phenotype is consistent with the reduction in rhythm strength of these flies in the second week of DD (DD 7–12) compared to the first (DD 1–6) (Fig. 4g and Supplementary Table 3). In addition, 16% of *Pdf>Dcr2; poe^RNAi* flies exhibited immediate arrhythmicity (termed "arrhythmic") and 12% displayed rhythms with fluctuating period lengths (termed "rhythmic-unstable period") throughout DD (Fig. 4e, f). *Pdf>Dcr2; poe^RNAi* flies were indistinguishable from controls in terms of their ability to anticipate the onset of morning and evening under LD conditions (Supplementary Fig. 2e–h).

Collectively, these results reveal that *poe*, like its mammalian homolog, serves an important role in the control of circadian rhythms. Specifically, *poe* is critical for the generation or regulation of free-running rhythms in flies.

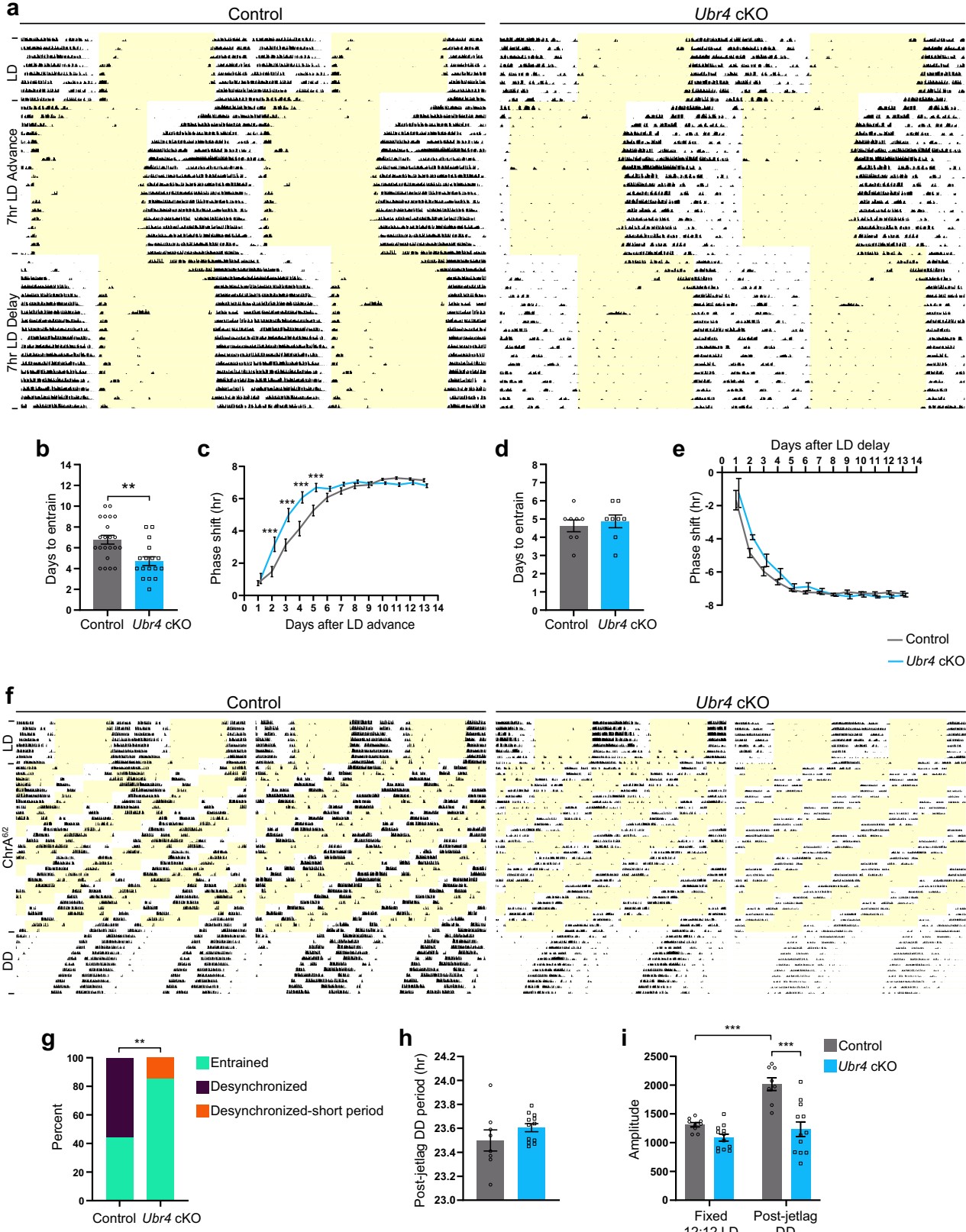

**PDF-specific *poe* knockdown alters PDF trafficking and damps dPER rhythms in fly clock neurons**. The behavioral phenotype of *Pdf>Dcr2; poe^RNAi* flies under DD partially mimics that of *Pdf*-null (*Pdf^01*) flies with respect to the delayed arrhythmicity[4]. Given the impaired axonal trafficking of VIP, the vertebrate functional homolog of PDF, in SCN neurons, we speculated that

altered PDF trafficking may underlie the deficits in circadian behavior of these flies. To address this, we examined the abundance of PDF protein in cell bodies and axonal projections at CT2 and CT14 (Fig. 5a). The s-LN_v neurons extend projections to the dorsal region of the brain, whereas the l-LN_v neurons project to the ipsilateral and contralateral optic lobes (OL), the latter via the

**Fig. 2 Absence of UBR4 in the murine SCN promotes entrainment to acute and chronic jetlag. a, f** Representative actograms displaying wheel-running activities of control and *Ubr4* cKO mice subjected to a 7-h LD advance followed by a 7-h LD delay (**a**), or to chronic jetlag (ChrA[6/2]) (**f**). Periods of light are shaded in yellow. **b, d** Days to re-entrain to the 7-h advance (**b**) or delay (**d**) of the LD schedule. For **b**, $n = 22$ control and 17 *Ubr4* cKO; **$p = 0.0017$, two-tailed Mann-Whitney $U$ test. For **d**, $n = 8$ per genotype; two-tailed unpaired $t$ test. **c, e** Daily phase shifts to the 7-h advance (**c**) or delay (**e**) of the LD schedule. For **c**, $n = 22$ control and 17 *Ubr4* cKO; ***$p < 0.0001$ (days 2–4), ***$p = 0.0006$ (day 5). For **e**, $n = 8$ per genotype. Linear mixed-effects modeling with Bonferroni's post hoc (**c, e**). **g** Percentage of mice displaying "entrained" (one rhythm, ~21 h period), "desynchronized" (two rhythms, ~21 h and >24 h periods), or "desynchronized-short period" (two rhythms, ~21 h and < 24 h periods) behavior under the ChrA[6/2] schedule. $n = 9$ control and 13 *Ubr4* cKO; **$p = 0.0072$, two-sided chi-square test. **h** Period length under DD after exposure to ChrA[6/2]. $n = 8$ control and 12 *Ubr4* cKO; two-tailed unpaired $t$ test. **i** $X^2$ periodogram amplitude under pre-ChrA[6/2] LD (fixed 12:12 LD) and post-jetlag DD. $n = 9$ control (fixed LD), 8 control (DD), 12 *Ubr4* cKO. ***$p = 0.0002$, control (LD) vs. control (DD); ***$p < 0.0001$, control vs. *Ubr4* cKO (DD); linear mixed-effects modeling with Bonferroni's post hoc. Values represent mean ± SEM (**b–e, h, i**) or percent (**g**). "$n$" represents the number of mice.

posterior optic tract (POT)[29,30]. *Pdf>Dcr2; poe^RNAi* flies possessed the expected number of PDF-positive small and large $LN_v$ neurons, indicating normal development of these cells (Supplementary Fig. 3b, c). At CT2, PDF expression within the OL, POT, and dorsal projections (DP) was drastically reduced in *Pdf>Dcr2; poe^RNAi* flies compared to *+>poe^RNAi* and *Pdf>Dcr2* controls (Fig. 5b–d). However, there was no effect of *poe* KD on PDF expression in the dorsal terminals (DT) at CT2 (Fig. 5e). At CT14, *Pdf>Dcr2; poe^RNAi* flies continued to exhibit reduced PDF expression in the POT but not in the OL or DP (Fig. 5b–d). Intriguingly, we observed a significant accumulation of PDF in the DT of *Pdf>Dcr2; poe^RNAi* flies at CT14 when compared to controls (Fig. 5e). In terms of expression in the cell body, PDF levels were significantly greater in the soma of s-$LN_v$ neurons of *Pdf>Dcr2; poe^RNAi* flies compared to controls, but were not different in the l-$LN_v$s, regardless of time point (Fig. 5f, g). *Pdf* mRNA expression in the small and large $LN_v$s of *Pdf>Dcr2; poe^RNAi* flies was not markedly different from that of *+>poe^RNAi* control flies (Supplementary Fig. 3d, e), indicating that post-transcriptional mechanisms underlie the observed changes in PDF distribution and abundance. Moreover, using a GAL4-driven membrane-targeted marker, CD2-HRP, we determined that *poe* KD in PDF neurons did not affect neuronal morphology, thus ruling out the possibility that altered PDF distribution in the *Pdf>Dcr2; poe^RNAi* flies was due to structural defects of the axonal projections (Supplementary Fig. 3f).

Next, we examined the effects of *poe* KD on the rhythmic expression of the core clock protein, dPER, in PDF-positive neurons (Fig. 5h, i and Supplementary Fig. 3g). In the s-$LN_v$s of control flies, dPER expression was strongly rhythmic with a peak in the late night-to-early day (Fig. 5j). In contrast, the expression of dPER was markedly reduced in the s-$LN_v$s of *Pdf>Dcr2; poe^RNAi* flies; furthermore, rhythms were severely damped and peaked in the mid-to-late day (Fig. 5j). dPER levels in the l-$LN_v$s were not affected by *poe* KD (Fig. 5k). We also observed drastically reduced and non-rhythmic expression of dPER in the small and large $LN_v$s of *tim>Dcr2; poe^RNAi* flies; however, interpretation of these results was complicated by the fact that these flies had fewer $LN_v$s than *tim>Dcr2* and *+>poe^RNAi* controls (Supplementary Fig. 4). Collectively, these data argue for a conserved role of POE/UBR4 in the regulation of neuropeptide trafficking and PER expression within circadian clock neurons of flies and mice.

**The absence of UBR4 impairs cargo transport along the secretory pathway and alters Golgi morphology.** To understand the cellular basis for the neuropeptide transport defect arising from *Ubr4/poe* ablation, we conducted trafficking studies that utilized a human embryonic kidney (HEK) 293 T cell line in which the *UBR4* gene had been knocked out (KO) using CRISPR/Cas9 technology[24]. Immunocytochemistry revealed the punctate distribution of UBR4 protein in the cytoplasm, and Western blot

analysis confirmed its absence in our *UBR4* KO cell line (Supplementary Fig. 5a, b). *UBR4* KO and wild-type (WT) cells were then transfected with a reporter construct expressing neuropeptide Y (NPY) fused to enhanced green fluorescent protein (GFP), and the subcellular localization of NPY-GFP was determined by immunofluorescence (Fig. 6a). NPY-GFP was distributed evenly throughout the cytoplasm of WT cells, whereas in *UBR4* KO cells it was largely restricted to the Golgi apparatus, as shown by the high degree of co-localization with the *cis*-Golgi and *trans*-Golgi network (TGN) markers, GM130 and p230, respectively (Fig. 6a, c, d and Supplementary Fig. 5c). Additionally, the *cis*-Golgi and TGN compartments exhibited an abnormally distended morphology in *UBR4* KO cells, which was reflected in an increase in their area and aspect ratio (H/W) relative to WT cells (Fig. 6b, e, f and Supplementary Fig. 5d, e). These results suggest that protein trafficking through the Golgi may be delayed or impaired in the absence of UBR4.

Next, we examined the kinetics of neuropeptide trafficking using the RUSH system to synchronize the transport process[31]. This system uses a Streptavidin "hook" to mediate ER retention of a cargo protein (in our case, NPY-GFP) that is fused to Streptavidin-binding peptide (SBP). The addition of biotin initiates the release of the cargo-SBP fusion protein from the hook and subsequent trafficking along the secretory pathway. Using this method, *UBR4* WT and KO cells transfected with RUSH plasmids were harvested at several time points following the addition of biotin and processed by immunofluorescence (Fig. 6g). At $t = 0'$ (no biotin added), NPY-GFP was evenly dispersed in *UBR4* WT and KO cells, indicating ER localization (Fig. 6g). ER-to-Golgi transport was accelerated in the absence of UBR4, as shown by the higher level of NPY-GFP expression in the Golgi relative to the rest of the cell and the greater proportion of cells with NPY-GFP expression in the Golgi in *UBR4* KO cultures 15 min following the addition of biotin (Fig. 6h, i). Once NPY-GFP was fully trafficked to the Golgi ($t = 30'$), export from the Golgi was markedly slower in *UBR4* KO cells compared to WT, as evident by the more gradual decline in relative NPY-GFP expression in the Golgi from $t = 60'$ to $t = 120'$ and the larger proportion of cells that retained NPY-GFP expression within the Golgi at $t = 120'$ (Fig. 6h, i). To confirm these findings, we combined the RUSH system with live-cell imaging to track the kinetics of NPY-GFP trafficking in individual cells (Fig. 6j and Supplementary Movie 1). The results are consistent with an impairment or delay in Golgi exit in the absence of UBR4, given the persistent presence of NPY-GFP signal within the Golgi of *UBR4* KO cells as late as 144 min following addition of biotin (Fig. 6j and Supplementary Movie 1). Analysis of a constitutively secreted cargo, glycosylphosphatidylinositol-anchored enhanced GFP protein (GPI-GFP), using the RUSH assay showed a similar effect whereby ER-to-Golgi transport was faster but Golgi exit was slower in the absence of UBR4 (Supplementary Fig. 5f–h). Interestingly, large intracellular vesicles containing GPI-GFP were

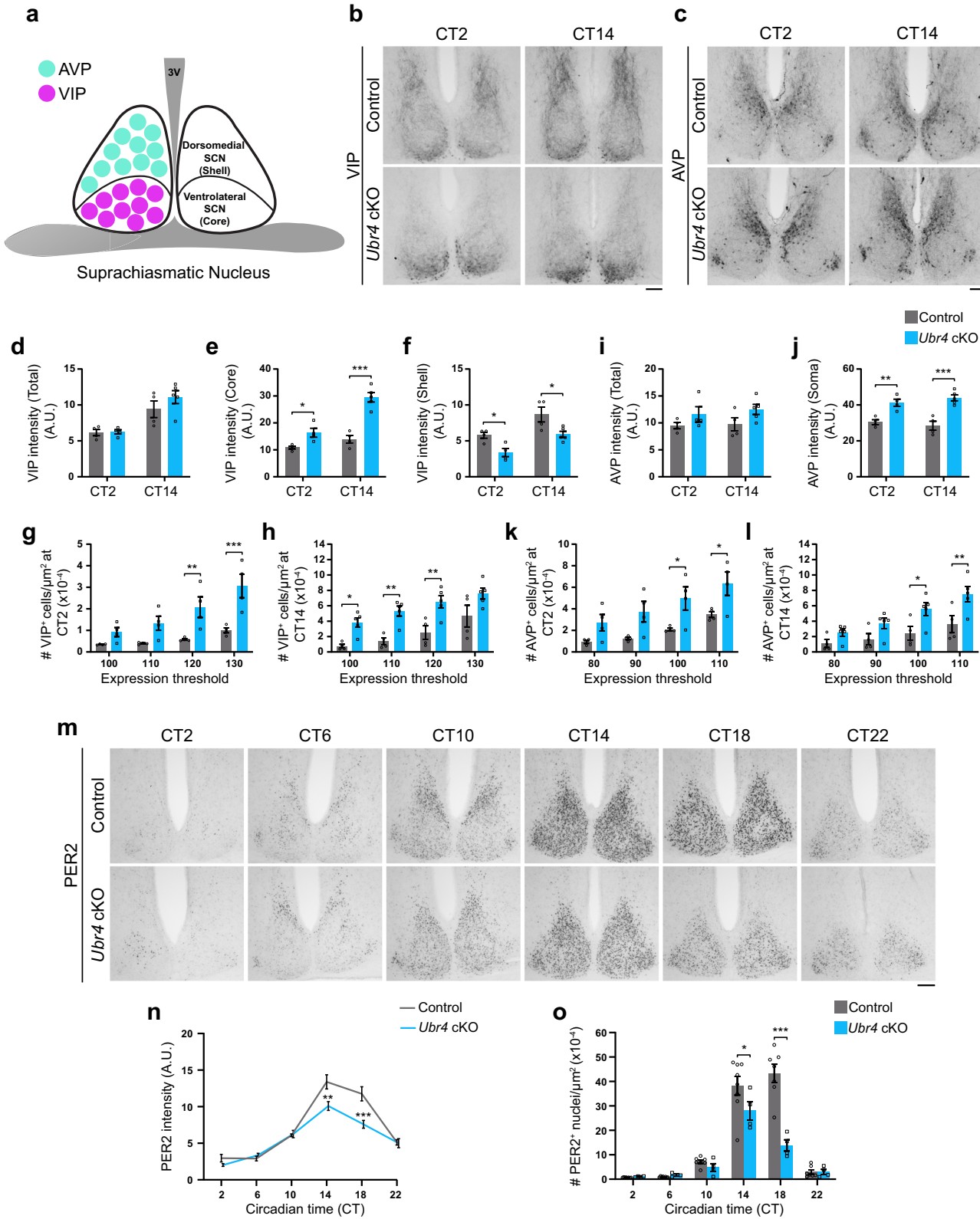

prevalent in the cytoplasm of *UBR4* KO cells and preceded the appearance of GPI-GFP at the plasma membrane (Supplementary Fig. 5f).

Lastly, we determined whether the effects of UBR4 on cargo export from the Golgi are dependent on the UBR-box domain, which is required for the recognition of N-degrons[32]. Over-expression of full-length (FL), wild-type UBR4 in *UBR4* KO

HEK293T cells rescued the Golgi export phenotype, as evident by the broad distribution of NPY-GFP throughout the cytoplasm and reduced localization in the Golgi relative to the rest of the cell (Fig. 6k, l and Supplementary Fig. 5i). Surprisingly, the phenotype was also rescued by overexpression of two mutant forms of UBR4 in which the function of the UBR-box was abrogated: one mutant contained a partial deletion of the UBR-box domain (Del) while

**Fig. 3 UBR4 is essential for the proper trafficking of VIP and AVP in the SCN and for mPER2 expression. a** Schematic depicting the location of AVP and VIP neurons in the SCN. **b, c, m** Photomicrographs of VIP (**b**), AVP (**c**), and mPER2 (**m**) immunoreactivity in the SCN of control and *Ubr4* cKO mice at defined CTs. Scale bar, 100 µm. **d–f** Immunoreactive intensity of VIP in the whole (**d**), core (**e**), or shell (**f**) SCN. **g, h** Density of VIP-IR cells in the SCN at CT 2 (**g**) or CT 14 (**h**) as a function of expression intensity. **i, j** Immunoreactive intensity of AVP in the whole SCN (**i**) or in the soma of shell SCN neurons (**j**). **k, l** Density of AVP-IR cells in the SCN at CT 2 (**k**) or CT 14 (**l**) as a function of expression intensity. Low threshold values (on the *x*-axis in **g, h, k, l**) indicate strong expression, whereas high threshold values indicate weak expression. $n = 4$ control (CT 2 and CT 14), 4 *Ubr4* cKO (CT 2), 5 *Ubr4* cKO (CT 14) (**d–l**). *$p = 0.048$, ***$p < 0.0001$ (**e**); *$p = 0.0449$ (CT 2), *$p = 0.0159$ (CT 14) (**f**); **$p = 0.0058$, ***$p = 0.0002$ (**g**); *$p = 0.0378$, **$p = 0.0052$ (110), **$p = 0.0044$ (120) (**h**); **$p = 0.0037$, ***$p = 0.0001$ (**j**); *$p = 0.0329$ (100), *$p = 0.0338$ (110) (**k**); *$p = 0.0438$, **$p = 0.008$ (**l**). **n** Immunoreactive intensity of mPER2 in the whole SCN. $n = 8$ control and 4 *Ubr4* cKO; **$p = 0.0083$, ***$p = 0.0005$. **o** Density of mPER2-IR cells in the SCN. In ascending order of CT: $n = 8, 8, 7, 8, 7, 7$ for control; $n = 4, 3, 4, 4, 4, 4$ for *Ubr4* cKO. *$p = 0.0233$, ***$p < 0.0001$. Values represent mean ± SEM. "*n*" represents the number of mice. Two-way ANOVA with Bonferroni's post hoc. A.U. arbitrary units.

the other carried multiple His/Cys→Ala mutations in the UBR-box (Ala) (Fig. 6k, l and Supplementary Fig. 5i). The inefficient export of NPY-GFP from the Golgi of *UBR4* KO cells ultimately impacted secretion, as shown by the reduced amount of NPY-GFP in the growth media of *UBR4* KO cultures compared to WT controls (Supplementary Fig. 5j).

Collectively, our data strongly suggest that UBR4 has a critical role in controlling the kinetics of protein trafficking through the regulated and constitutive secretory pathways. Importantly, the absence of UBR4 impedes the rate of cargo exit from the Golgi and thus secretion from the cell, effects which we believe can have serious consequences on timely communication between cells within a network.

**UBR4 regulates the abundance of CRN7 to promote cargo export from the Golgi**. We next sought to identify the molecular mechanism by which UBR4 promotes the exit of cargo proteins from the Golgi. Based on our hypothesis that UBR4 regulates the abundance of a specific mediator of Golgi export, we used label-free quantitative mass spectrometry (MS) to analyze the proteomes of *UBR4* KO and WT HEK293T cells. Out of 4,026 unique proteins that were accurately quantified, we identified 471 differentially expressed proteins (DEPs) (11.7%) based on a BH.q value cutoff of 0.05 (Fig. 7a and Supplementary Data 1). Of these, 191 were upregulated and 280 were downregulated in *UBR4* KO cells relative to WT (Fig. 7a and Supplementary Data 1). As expected, Gene Ontology (GO) enrichment analysis of the upregulated and downregulated proteomes revealed significant enrichment of GO terms that are related to intracellular protein transport (Fig. 7c and Supplementary Data 2). The list of significantly enriched biological process (BP) GO terms for the upregulated DEPs included "intra-Golgi vesicle-mediated transport" and "Golgi vesicle transport", whereas for the downregulated DEPs the list encompassed "regulation of protein targeting" and "positive regulation of intracellular protein transport" (Fig. 7c). Furthermore, the upregulated DEPs were significantly enriched for the cellular component (CC) GO terms "endoplasmic reticulum", "Golgi apparatus", and "endocytic vesicle lumen", while the GO-CC terms that are associated with the downregulated DEPs included "extracellular vesicle", and "membrane-bound vesicle" (Fig. 7c).

To identify a candidate mediator of the trafficking phenotype in *UBR4* KO cells, we carefully evaluated the list of DEPs for proteins that had previously been implicated in Golgi export. CRN7, a protein that binds to the cytosolic side of Golgi complex membranes, has been shown to mediate cargo export from the Golgi and is required for normal Golgi morphology[33–35]. Consistent with our quantitative MS data, Western blot analyses revealed a significant downregulation of CRN7 in *UBR4* KO HEK293T cells and in SCN tissues extracted from *Ubr4* cKO mice (MS: Log$_2$(fold-change (FC)) of CRN7 $= -0.9$; HEK293T WB: CRN7 FC $= 0.62$; SCN WB: CRN7 FC $= 0.39$) (Fig. 7b, d, e).

Importantly, elevating the levels of CRN7 in *UBR4* KO cells by transient transfection of a Flag-CRN7 construct rescued the deficit in Golgi export (Fig. 7f and Supplementary Fig. 6a). Compared to pcDNA-transfected controls, *UBR4* KO cells expressing Flag-CRN7 showed a marked reduction in the accumulation of NPY-GFP within the *cis*-Golgi and TGN relative to the rest of the cell (Fig. 7f–h and Supplementary Fig. 6a). The distribution of NPY-GFP in *CRN7*-overexpressing *UBR4* KO cells was indistinguishable from that of WT controls (Fig. 7f and Supplementary Fig. 6a). Overexpression of Flag-CRN7 in WT cells did not affect the localization of NPY-GFP, indicating that levels of endogenous CRN7 in these cells are sufficient to ensure proper neuropeptide trafficking (Fig. 7f and Supplementary Fig. 6a).

It does not appear that UBR4 promotes the accumulation of CRN7 by enhancing its gene transcription or mRNA stability, or by stabilizing the protein. First, transcript levels of *CRN7* were not different between *UBR4* WT and KO HEK293T cells (Supplementary Fig. 6b). Second, proteasomal inhibition using MG-132 for up to 18 h did not increase the level of CRN7 in *UBR4* KO or WT HEK293T cells, consistent with reports that CRN7 is a highly stable protein[36] (Supplementary Fig. 6c). Even the combination of proteasomal and lysosomal inhibition, the latter using Bafilomycin A1 and E64d/Pepstatin A, failed to restore CRN7 abundance in *UBR4* KO cells to WT levels (Supplementary Fig. 6d). These data suggest that UBR4 may instead be regulating CRN7 abundance at the translational level. Consistent with this idea, polysome profiling indicated that the sedimentation of *CRN7* mRNA was shifted towards the lighter density polysome fractions in sucrose density gradient centrifugation in the *UBR4* KO HEK293T lysates compared to the WT, while the distribution of *GAPDH* mRNA was not affected (Supplementary Fig. 6e, f). This result suggests that the rate of translation of *CRN7* is selectively reduced in the absence of UBR4. Altogether, our findings show that UBR4 facilitates cargo export from the Golgi complex by positively regulating the abundance of CRN7.

**Restoring CRN7 abundance in *Ubr4*-deficient SCN neurons rescues VIP trafficking and promotes desynchronized behavior under chronic jetlag**. So far, we have used HEK293T cells to identify a mechanism that potentially underlies the neuropeptide trafficking phenotype in central clock neurons of mice and flies. To show unequivocally that CRN7-mediated cargo export from the Golgi is impaired in *Ubr4*-deficient SCN neurons, we first examined the SCN of *Ubr4* cKO mice for evidence of cargo retention within the Golgi using transmission electron microscopy (TEM). Unlike the controls, the SCN of *Ubr4* cKO mice was characterized by the frequent presence of distended Golgi—a likely consequence of cargo retention—and a corresponding increase in the width of Golgi cisternae (Fig. 8a, b).

To demonstrate that cargo proteins are retained in SCN neurons in a cell-autonomous fashion, we prepared dissociated SCN neuronal cultures from *Ubr4*$^{fl/fl}$ neonates and virally

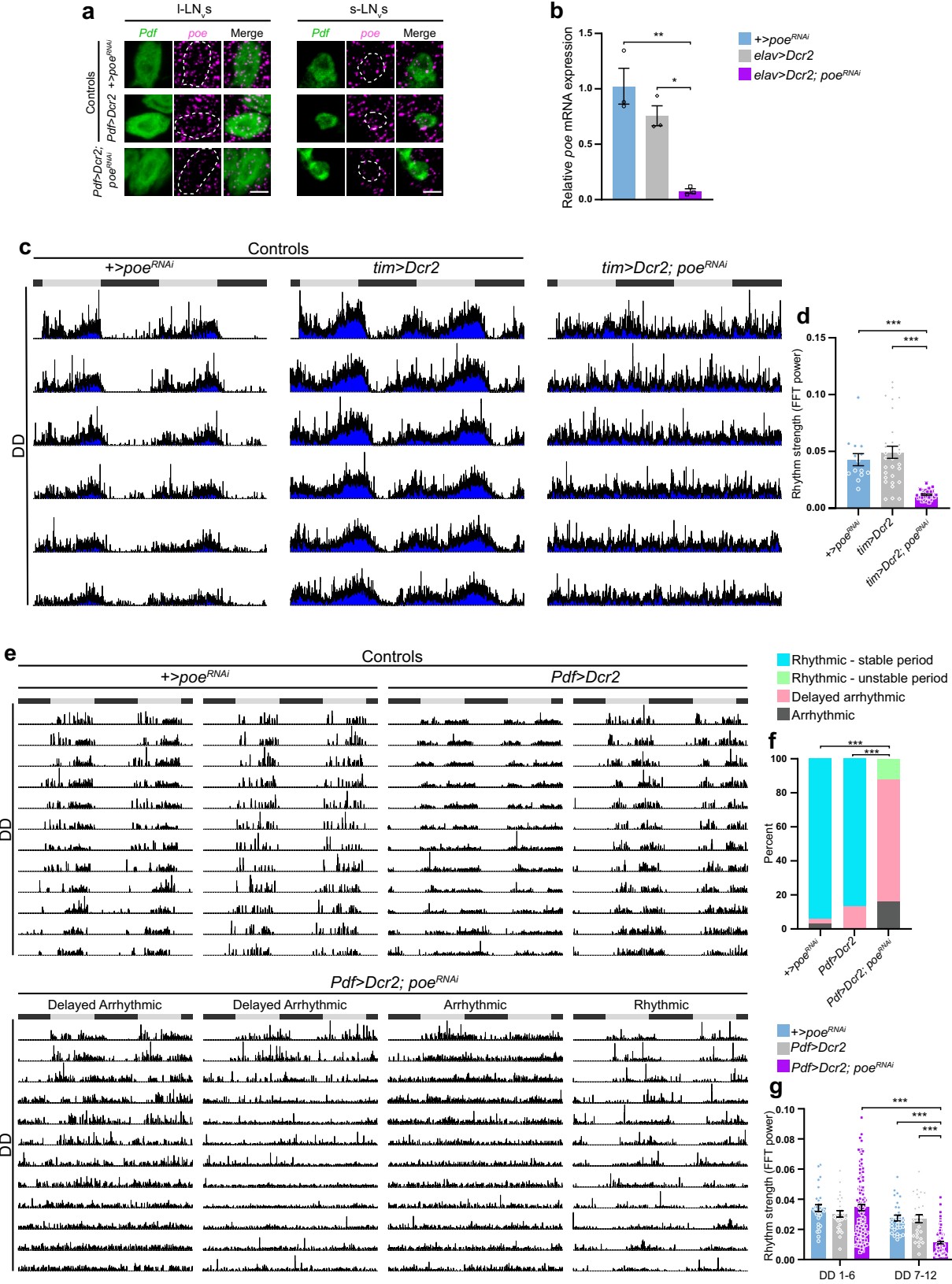

transduced them with Cre recombinase to ablate *Ubr4* (Supplementary Fig. 7a). In Cre-transduced cultures, we could easily identify cells with endogenous VIP expression, as VIP was highly localized to the Golgi (Supplementary Fig. 7b). In contrast, we had difficulty finding VIP-expressing neurons in GFP-transduced control cultures, as the VIP signal was very weak and diffuse,

presumably as a result of efficient trafficking and secretion (Supplementary Fig. 7b). To quantify VIP localization, we performed dual transduction using adeno-associated viral (AAV1) constructs that expressed Cre recombinase and the murine *Vip* gene under the control of the human Synapsin (hSyn) promoter (Fig. 8c). Similar to endogenous VIP, ectopically

**Fig. 4 Poe knockdown in D. melanogaster clock neurons impairs behavioral rhythms under constant darkness. a** In situ hybridization analysis of *Pdf* (green) and *poe* (magenta) mRNA expression in l-LN$_v$s and s-LN$_v$s. Scale bar, 5 µm. **b** qRT-PCR analysis of relative *poe* mRNA expression in whole fly brains of +>*poe*$^{RNAi}$, *elav*>*Dcr2*, and *elav*>*Dcr2; poe*$^{RNAi}$ flies. n = 3 samples per group; *p = 0.0127, **p = 0.0024, one-way ANOVA with Bonferroni's post hoc. **c** Averaged actograms displaying activity rhythms of *tim*>*Dcr2; poe*$^{RNAi}$ flies and controls under DD. **d** Rhythm strength (FFT power) of *tim*>*Dcr2; poe*$^{RNAi}$ flies and controls under DD. n = 14 +>*poe*$^{RNAi}$, 31 *tim*>*Dcr2*, 31 *tim*>*Dcr2; poe*$^{RNAi}$. ***p < 0.0001, Kruskal–Wallis with a Dunn's post hoc. **e** Individual actograms displaying activity rhythms of *Pdf*>*Dcr2; poe*$^{RNAi}$ flies and controls under DD. Gray and black bars represent the light and dark phases, respectively, of the previous LD schedule (**c**, **e**). **f** Percentage of *Pdf*>*Dcr2; poe*$^{RNAi}$ flies and controls displaying different behaviors under DD. n = 31 +>*poe*$^{RNAi}$, 30 *Pdf*>*Dcr2*, 111 *Pdf*>*Dcr2; poe*$^{RNAi}$. ***p < 0.0001, two-sided chi-square test. **g** Rhythm strength of *Pdf*>*Dcr2; poe*$^{RNAi}$ flies and controls during the first (DD1-6) and second (DD7-12) week of DD. n = 30 +>*poe*$^{RNAi}$, 28 *Pdf*>*Dcr2*, 104 *Pdf*>*Dcr2; poe*$^{RNAi}$. ***p < 0.0001, linear mixed-effects modeling with Bonferroni's post hoc. Values represent mean ± SEM (**b**, **d**, **g**) or percent (**f**). "n" represents the number of samples (**b**) or the number of flies (**d**, **f**, **g**).

expressed VIP was highly concentrated in the Golgi of Cre-transduced neurons relative to GFP-transduced controls (Fig. 8c, d). Importantly, in triple transduction experiments, ectopically expressed VIP was no longer localized in the Golgi of Cre-transduced neurons when murine CRN7 was overexpressed (Fig. 8e, f and Supplementary Fig. 7a). These results demonstrate that the absence of UBR4 in SCN neurons impairs the export of cargo proteins from the Golgi through a CRN7-dependent mechanism.

Lastly, to determine whether defects in CRN7-mediated protein trafficking in the SCN underlie the resistance of *Ubr4* cKO mice to jetlag, we examined the effects of CRN7 over-expression in the ChrA$^{6/2}$ jetlag paradigm. *Ubr4* cKO mice received bilateral SCN injections of AAV1 constructs encoding CRN7-2A-mCherry under the control of the CMV promoter or a control AAV1-CMV-mCherry vector prior to activity monitoring. Post-mortem analysis revealed robust expression of mCherry in the majority of, but not all, SCN neurons (Supplementary Fig. 7c). All *Ubr4* cKO mice (9/9) injected with the mCherry control vector showed a single, entrained component of ~21 h (period: 21.00 ± 0.01 h) under the ChrA$^{6/2}$ schedule (Fig. 8g, h). In contrast, only 50% of *Ubr4* cKO mice (4/8) injected with the CRN7 vector were entrained to the ChrA$^{6/2}$ schedule (period: 21.02 ± 0.02 h), while the remaining 50% (4/8) exhibited desyn-chronized behavior with two rhythmic components (period: 20.96 ± 0.02 h and 23.69 ± 0.15 h) (Fig. 8g, h and Supplementary Fig. 7d). This dichotomy in the behavioral response to chronic jetlag is reminiscent of the behavior of wild-type mice, although the period of the long component is slightly shorter in the CRN7-overexpressing *Ubr4* cKO mice, potentially as a result of the mosaicism in CRN7 transgene expression (Fig. 2g). Collectively, our findings demonstrate that UBR4 regulates the abundance of CRN7 in SCN neurons to facilitate neuropeptide trafficking from the Golgi. This UBR4-CRN7 pathway is crucial for maintaining appropriate behavioral rhythms under chronic jetlag.

## Discussion

Here, we have used the circadian clock system in mice and flies to uncover a conserved role of UBR4/POE in the regulation of secretory protein trafficking. The absence of UBR4/POE in clock neurons reduces the efficiency of cargo export from the Golgi, resulting in abnormal accumulation of neuropeptides in the soma. UBR4 is required to maintain high expression of CRN7, which in turn mediates or facilitates neuropeptide trafficking out of the Golgi. At the level of behavior, knocking out (or knocking down) *Ubr4/poe* in clock neurons results in perturbed activity rhythms that are suggestive of defects in clock network syn-chrony, despite the phenotypic differences exhibited by our two model organisms.

In eukaryotic cells, the biosynthetic secretory pathway is responsible for the synthesis, processing, and transport of all transmembrane proteins, secreted proteins, and resident proteins of the endomembrane system (i.e., ER, Golgi apparatus,

lysosome)[37,38]. In particular, the Golgi serves the important function of sorting and packaging proteins into transport vesicles for eventual delivery to their intended destination[37]. Convergent lines of evidence from mice, flies, and human cells strongly suggest that UBR4/POE is crucial for the efficient export of secretory proteins from the Golgi. First, in contrast to control animals, *Ubr4/poe* mutants show preferential accumulation of AVP, VIP, or PDF in the soma of clock neurons rather than axonal projections. Second, SCN neurons and HEK293T cells deficient for UBR4 have markedly distended Golgi cisternae that are suggestive of cargo retention. Third, in cultured HEK293T cells or dispersed SCN neurons, ablating UBR4 causes abnormal accumulation of NPY-GFP or VIP, respectively, in the Golgi. Fourth and most importantly, in trafficking assays, the cargo proteins NPY-GFP and GPI-GFP are present in the Golgi for a much longer time in *UBR4*-deficient cells compared to control cells. Collectively, these data underscore the requirement for UBR4 in efficient Golgi export of cargo proteins, which in turn should impact extracellular release, as we have also demonstrated in our secretion assay. It is worth noting that the effects of UBR4 on Golgi export do not depend on the presence of a functional UBR-box domain and thus appear to be unrelated to UBR4's canonical role as an N-recognin. Furthermore, the absence of UBR4 impacts both the regulated and constitutive secretory pathways, raising the possibility that the trafficking of other proteins, in addition to neuropeptides, may be affected in our animal models.

Mechanistically, we show that UBR4 facilitates Golgi export by supporting the high expression of CRN7. Mammalian CRN7 is a cytosolic protein that is recruited to the cytoplasmic side of Golgi membranes where it promotes normal Golgi organization and biogenesis of TGN-derived transport carriers required for Golgi export[33–36,39,40]. The depressed levels of CRN7 in *Ubr4*-deficient SCN neurons and HEK293T cells, coupled with the rescue of VIP or NPY-GFP export from the Golgi once CRN7 is overexpressed in these cells, respectively, indicate that CRN7 is causal to the Golgi trafficking defect observed in the absence of UBR4. The shift in *CRN7* transcripts towards the light polysome fraction in *UBR4* KO HEK293T cells suggests that UBR4 enhances the translation of CRN7. UBR4 does not appear to affect the stability of CRN7 protein, consistent with the observation that UBR4 mutants in which the UBR-box domain is disabled are still cap-able of rescuing the Golgi export phenotype. Future experiments are needed to confirm the effects of UBR4 ablation on the translation of CRN7 and to establish precise molecular mechan-isms. Although our data suggest that it is unlikely that the N-recognin activity of UBR4 is required for the regulation of CRN7, given the effects of the UBR-box mutants on Golgi export, we cannot entirely rule out the possibility that the link between UBR4 and CRN7 translation is of a more indirect nature and involves an intermediary protein whose stability is directly regulated by UBR4. In addition to translation, UBR4 might promote the recruitment of CRN7 to Golgi membranes, a

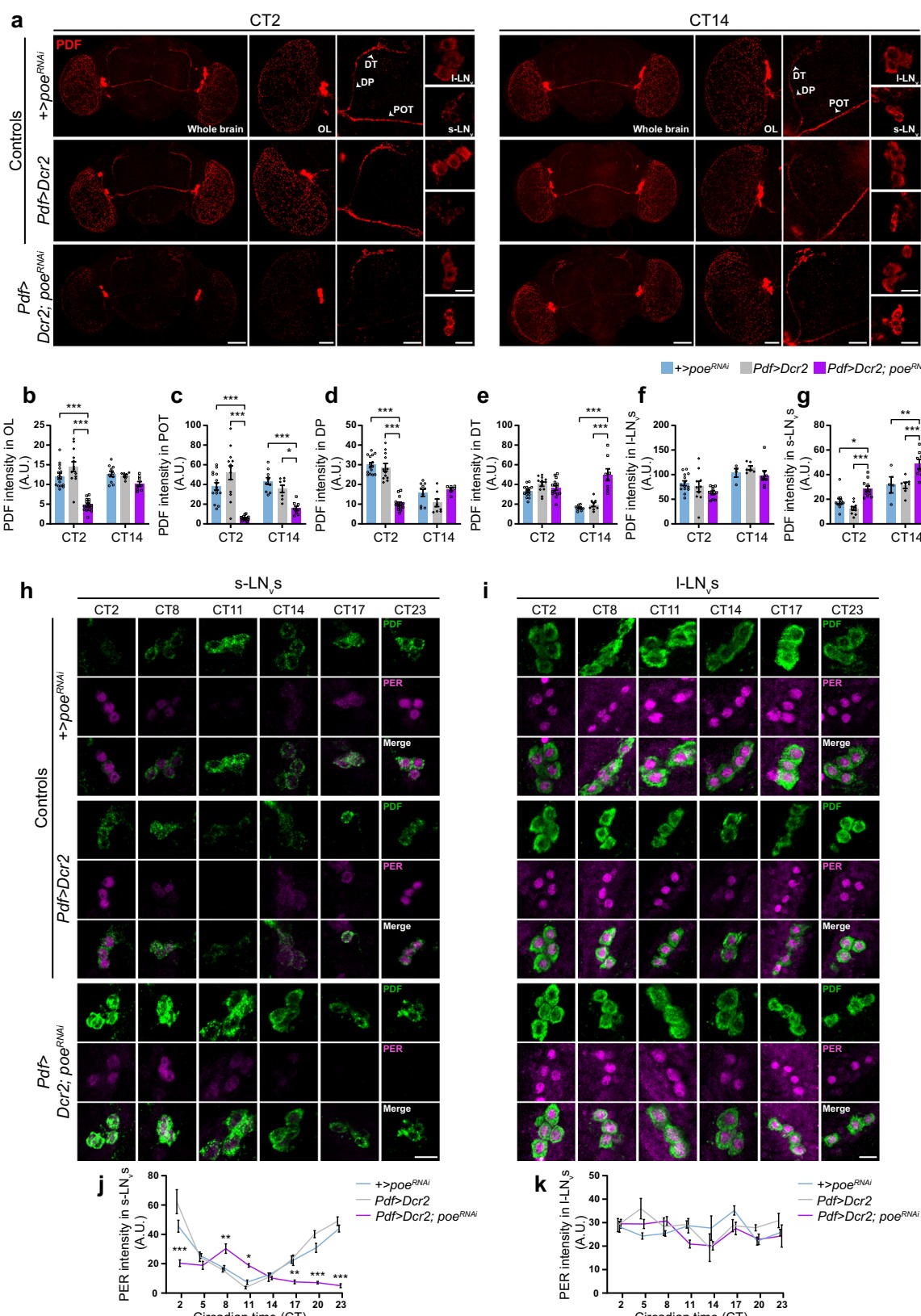

possibility that merits further scrutiny. Interestingly, even though its function remains unclear, POD-1, the *Drosophila* homolog of CRN7, possesses motifs that are required for targeting Golgi membranes[35]. This raises the tantalizing possibility that a UBR4-CRN7 pathway may be a phylogenetically conserved mechanism regulating Golgi trafficking.

Although we elected to focus on the mechanism underlying the Golgi export phenotype, there is some evidence to suggest that UBR4 may affect other vesicular trafficking events. For example, ER-to-Golgi trafficking in HEK293T cells appears to be accelerated in the absence of UBR4. Exocytosis might also be reduced or delayed, given the conspicuous accumulation of large, post-Golgi,

**Fig. 5 PDF-specific *poe* knockdown alters PDF trafficking and damps dPER expression in fly clock neurons. a** PDF immunoreactivity (red) in the whole fly brain, optic lobe (OL), dorsal projections (DP), dorsal terminals (DT), posterior optic tract (POT), and large (l-LN$_v$) and small (s-LN$_v$) ventral lateral neurons at CT 2 and CT 14. Scale bar, 60 μm (whole brain), 30 μm (OL, POT, DP, DT), and 10 μm (l-LN$_v$, s-LN$_v$). **b–g** PDF-IR intensity in the OL (**b**), POT (**c**), DP (**d**), DT (**e**), and the soma of l-LN$_v$s (**f**) or s-LN$_v$s (**g**). $n = 15$ (**b–e**), 14 (**f**), or 11 (**g**) for +>*poe*$^{RNAi}$ (CT 2); $n = 10$ (**b–e**) or 4 (**f, g**) for +>*poe*$^{RNAi}$ (CT 14). $n = 13$ (**b, c**), 12 (**d, e**), or 10 (**f, g**) for *Pdf>Dcr2* (CT 2); $n = 7$ (**b, g**), 8 (**c–e**), or 6 (**f**) for *Pdf>Dcr2* (CT 14). $n = 16$ (**b**), 15 (**c–e**), or 11 (**f, g**) for *Pdf>Dcr2; poe*$^{RNAi}$ (CT 2); $n = 8$ (**b, c, e–g**) or 7 (**d**) for *Pdf>Dcr2; poe*$^{RNAi}$ (CT 14). ***$p < 0.0001$ (**b, d, e**); *$p = 0.0385$, ***$p < 0.0001$ (CT 2), ***$p = 0.0006$ (CT 14) (**c**); *$p = 0.0211$, **$p = 0.0058$, ***$p = 0.0001$ (CT 2), ***$p = 0.0005$ (CT 14) (**g**). **h, i** dPER (magenta) and PDF (green) immunoreactivity in s-LN$_v$s (**h**) and l-LN$_v$s (**i**) under DD. Scale bar, 10 μm. **j, k** d-PER-IR intensity in s-LN$_v$s (**j**) and l-LN$_v$s (**k**) as a function of CT. In ascending order of CT: $n = 6, 10, 10, 8, 4, 7, 6, 8$ for +>*poe*$^{RNAi}$ (**j**); $n = 6, 11, 8, 8, 4, 7, 5, 9$ for +>*poe*$^{RNAi}$ (**k**); $n = 5, 9, 5, 9, 4, 8, 5, 9$ for *Pdf>Dcr2* (**j, k**); $n = 7, 11, 10, 8, 4, 7, 6, 7$ for *Pdf>Dcr2; poe*$^{RNAi}$ (**j**); $n = 7, 11, 10, 9, 3, 7, 7, 7$ for *Pdf>Dcr2; poe*$^{RNAi}$ (**k**). *$p = 0.0485$, **$p = 0.0017$ (CT 8), **$p = 0.0054$ (CT 17), ***$p < 0.0001$ (**j**). Values represent mean ± SEM. "$n$" represents number of flies. Two-way ANOVA with Bonferroni's post hoc. A.U. arbitrary units.

GPI-GFP$^+$ membrane-enclosed compartments in the cytoplasm of *UBR4* KO HEK293T cells, the reduced secretion of NPY-GFP from these cells into the culture media, and the abnormally high accumulation of PDF in the dorsal terminals of *Pdf>Dcr2; poe*$^{RNAi}$ flies at CT14. Further investigations are warranted to fully characterize the effects of UBR4 along the entire biosynthetic secretory pathway.

The pattern of behavior elicited by *Ubr4/poe* deficiency suggests that clock network synchrony is perturbed. However, the extent to which reduced vesicular transport in central clock neurons impacts the circadian behavior of *Ubr4* cKO mice and *poe* KD flies has not been directly examined, except in the case of the chronic jetlag paradigm. While we believe that the defect in the trafficking and secretion of neuropeptides contributes to at least some of the behavioral phenotypes of our animal models, we by no means claim that it is the sole determinant or that all phenotypes (or lack thereof) are equally impacted by it. Under chronic jetlag, *Ubr4* cKO mice entrain efficiently to the Zeitgeber cycle and only upon overexpression of *Crn7* in the SCN will a subset of these animals desynchronize. These results, along with our *Crn7* rescue experiments in SCN neuronal cultures, suggest that defective neuropeptide trafficking in *Ubr4* cKO mice is responsible for their inability to desynchronize under extreme conditions of chronic jetlag. If we consider forced desynchrony under the ChrA$^{6/2}$ paradigm as desynchronization of a dual oscillator system, the resistance of *Ubr4* cKO mice to desynchrony suggests that *Ubr4* ablation affects the range of entrainment in such a way that the Zeitgeber period is now within the range of entrainment of both oscillators. The entrainment range depends on various factors including the amplitude and period of the oscillator as well as neuropeptide-mediated coupling of oscillators[41,42]. Given that the amplitude of PER2 rhythms is mildly reduced in the SCN of *Ubr4* cKO mice, we cannot exclude the possibility that the chronic jetlag phenotype arises from the combined effects of altered oscillator properties and impaired communication between oscillators.

Our fly model reaffirms the effects of *Ubr4/poe* ablation on neuropeptide trafficking, but also provides strong evidence for its effects on the molecular clock. Unlike *Ubr4* cKO mice, *Pdf>Dcr2; poe*$^{RNAi}$ flies exhibit profound changes in the amplitude and phase of PER oscillations in the s-LN$_v$s. The majority of *Pdf>Dcr2; poe*$^{RNAi}$ flies display delayed-arrhythmic behavior under DD, emerging after a brief period of rhythmicity: this phenotype has also been observed in *Pdf*$^{01}$ mutants[4]. While it may be tempting to infer that impaired PDF signaling is the cause of the delayed arrhythmicity of *Pdf>Dcr2; poe*$^{RNAi}$ flies, it is possible that this behavior reflects the aggregate effects of damped molecular oscillations and less PDF signaling. Interestingly, selective disruption of *per* in PDF neurons does not lead to arrhythmicity in the majority of flies, suggesting that the changes in PER oscillations in *Pdf>Dcr2; poe*$^{RNAi}$ flies, on their own, might not be sufficient to trigger delayed arrhythmic behavior

under DD[43,44]. It is also important to point out that not all phenotypes exhibited by *Pdf*$^{01}$ mutants, namely, lack of morning anticipation and advanced evening anticipation under LD as well as short DD period, are observed in *Pdf>Dcr2; poe*$^{RNAi}$ flies. The fact that *Pdf>Dcr2; poe*$^{RNAi}$ and *Pdf*$^{01}$ flies do not perfectly mirror each other with respect to their behavior is not surprising, given that (1) *poe* ablation is unlikely to result in the complete loss of PDF signaling, and (2) other mechanisms besides aberrant PDF signaling may also help to shape the behavior of *Pdf>Dcr2; poe*$^{RNAi}$ flies under different environmental light conditions.

Further investigations are needed to decipher the causal mechanisms underlying the individual circadian phenotypes of *Ubr4* cKO and *poe* KD flies. For example, it would be worthwhile to determine whether overexpression of *Crn7* is sufficient to rescue the acute jetlag or LL phenotypes of *Ubr4* cKO mice and whether the fly homolog, POD-1, plays a similar role in the s-LN$_v$s to regulate PDF trafficking and circadian behavior. Although our current study focused on the effects of *Ubr4* ablation on the transport of neuropeptides, it is clear that both the regulated and constitutive secretory pathways are perturbed in *UBR4* KO HEK293T cells. Thus, there may be other proteins whose trafficking is also impaired in *Ubr4/poe*-deficient clock neurons and which contribute to the behavioral phenotype. Beyond the role of UBR4 in protein trafficking, it is important to determine the contributions of damped PER oscillations to the behavior of *Ubr4* cKO mice and *poe* KD flies, and to identify the molecular underpinnings for the decrease in PER protein abundance in these animal models. Plausible mechanisms include reduced translation of *Per* transcripts and reduced stability of PER proteins. If UBR4 indeed regulates PER stability, it would be important to ask whether this is a direct effect that relies on the N-recognin activity of UBR4, or alternatively an indirect effect stemming from improper neuropeptide trafficking[45]. On the other hand, if UBR4 regulates *Per* translation, as it appears to do for *Crn7*, a full examination of the potential links between UBR4 and the translational machinery, including its regulators, is warranted. Finally, a deeper exploration of the 471 DEPs in *UBR4* KO cells may reveal additional factors that mediate the effects of UBR4 on vesicular trafficking or other cellular processes.

Our study of the circadian clock system has unveiled an unexpected role of UBR4 in the regulation of vesicular trafficking, a fundamental process in all eukaryotic cells. Although it is highly speculative, we believe that this particular function of UBR4 may explain some of the biological phenomena that have been documented when *Ubr4* or its homologs are ablated or mutated. *Ubr4*$^{-/-}$ mice are embryonic lethal at E9.5-E10.5 due to defects in neurogenesis and cardiovascular development[46]. These defects were linked to the depletion of many cell surface proteins, including cell adhesion proteins, from the plasma membrane (PM)[46]. Although the authors speculated that their depletion at the PM may be due to misregulated protein turnover, in light of our findings with the constitutive cargo GPI-GFP, a deficit in

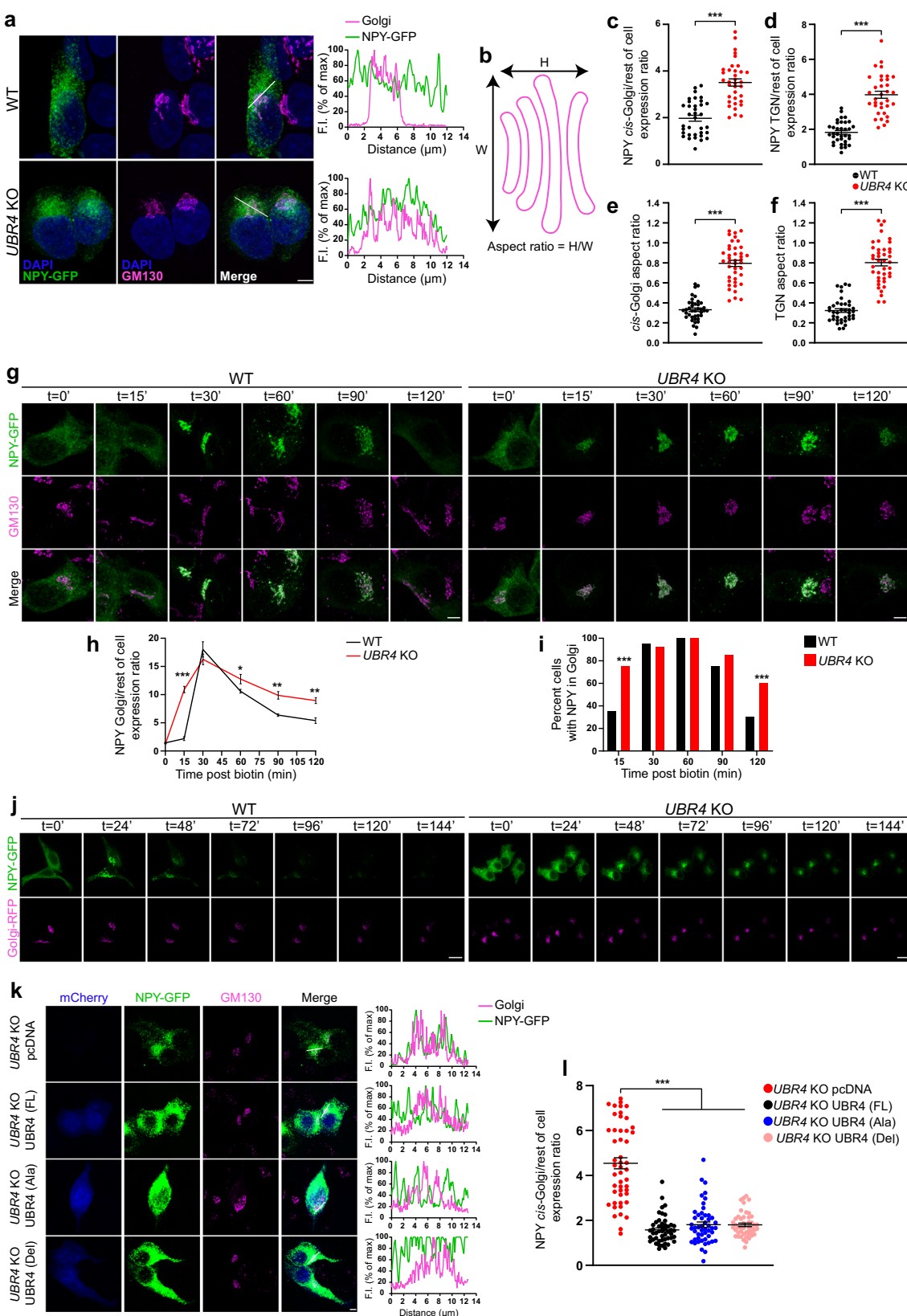

anterograde trafficking of these proteins to the PM is equally plausible. Along similar lines, disease-linked variants of UBR4 have been identified in patients suffering from episodic ataxia (EA), a neurological disorder that is associated with altered biosynthesis and trafficking of calcium and potassium channels to the PM[47–52]. Mutations in *poe* (also known as *pushover*) result in

enhanced neuronal excitability at the fly larval neuromuscular junction (NMJ), which becomes hypersensitive to the effects of the potassium channel blocker, quinidine[53]. The NMJ phenotype along with the observed male sterility in these *poe* mutants may be explained by impaired delivery of PM proteins that are essential for controlling membrane excitability and

**Fig. 6 The absence of UBR4 impairs cargo transport along the secretory pathway and alters Golgi morphology. a** NPY-GFP (green) and GM130 (magenta) immunofluorescence in *UBR4* WT and KO HEK293T cells. DAPI, blue. Profile plots (right) show NPY-GFP and GM130 fluorescence intensity (F.I.) along the reference axis (white line, merged panel). Scale bar, 5 μm. **b** Depiction of how the aspect ratio is calculated. **c, d** Ratio of NPY-GFP abundance in the *cis*-Golgi (**c**) or TGN (**d**) relative to the rest of the cell. $n = 35$ WT, 35 *UBR4* KO (**c**), 34 *UBR4* KO (**d**). **e, f** Aspect ratio of *cis*-Golgi (**e**) or TGN (**f**). $n = 39$ WT, 41 *UBR4* KO; ***$p < 0.0001$ (**c–f**), two-tailed unpaired *t* test. **g–i** Trafficking of NPY-GFP (green) in *UBR4* WT and KO HEK293T cells using the RUSH assay (**g**). Cells were fixed at the indicated times post-biotin and CHX addition, and immunostained for NPY-GFP and GM130 (magenta). Scale bar, 5 μm. Ratio of NPY-GFP abundance in the Golgi relative to the rest of the cell (**h**). In ascending order of time: $n = 14, 36, 35, 42, 32, 25$ WT; $n = 19, 35, 38, 45, 29, 44$ *UBR4* KO. *$p = 0.031$, **$p = 0.0034$ (90 min), **$p = 0.002$ (120 min), ***$p < 0.0001$; two-way ANOVA with Bonferroni's post hoc. Percentage of cells with NPY-GFP in the Golgi (**i**). ***$p < 0.0001$, two-sided chi-square test. **j** Live-cell images of NPY-GFP (green) trafficking in *UBR4* WT and KO HEK293T cells. Golgi, magenta. Scale bar, 20 μm. **k, l** NPY-GFP (green) and GM130 (magenta) immunofluorescence in *UBR4* KO HEK293T cells that had been co-transfected with empty vector (pcDNA) or with UBR4 (FL), UBR4 (Ala), or UBR4 (Del) constructs co-expressing mCherry (blue) (**k**). Profile plots are shown (right). Scale bar, 5 μm. Ratio of NPY-GFP abundance in the *cis*-Golgi relative to the rest of the cell (**l**). $n = 49$ pcDNA, 49 UBR4 (FL), 50 UBR4 (Ala), 47 UBR4 (Del). ***$p < 0.0001$, Kruskal–Wallis with a Dunn's post hoc. Values represent mean ± SEM (**c–f**, **h**, **l**) or percent (**i**). "$n$" represents the number of cells.

spermatogenesis, respectively[53,54]. Lastly, mutations in *BIG*, the homolog of *Ubr4* in *Arabidopsis*, result in elongation defects owing to impaired polar transport of the plant hormone, auxin[55]. In *BIG* mutants, the PM-resident auxin efflux carrier was mis-localized to an unidentified intracellular compartment[55], an observation that is consistent with a vesicular trafficking defect. The extent to which the N-recognin activity of UBR4 and the consequent effects on protein stability contribute to any aspect of its role in membrane trafficking remains open to further exploration.

In conclusion, our study has revealed a hitherto unappreciated role of UBR4 in the regulation of the biosynthetic secretory pathway, particularly in the export of cargo proteins from the Golgi complex. UBR4 enhances the rate of Golgi export by promoting the expression of CRN7 through a non-proteolytic mechanism. This function of UBR4 is conserved across species, and its disruption in clock neurons impairs the trafficking of neuropeptides that are required for clock network synchrony, ultimately perturbing behavioral rhythms.

## Methods

**Mice**. Details of mouse strains used in this paper are listed in Supplementary Table 4.

All animal handling and experimental procedures were performed at the University of Toronto Mississauga (UTM) Animal Facility and were approved by the UTM Animal Care Committee, complying with guidelines established by the University of Toronto Animal Care Committee and the Canadian Council on Animal Care. The following mouse strains were purchased from The Jackson Laboratory (Bar Harbor, ME, USA) and bred in-house to generate the appropriate genotypes for this study: homozygous $Ubr4^{fl/fl}$ mice in which the first coding exon of $Ubr4/p600$ is flanked by loxP sequences ($Ubr4^{tm1.2Nkt}$); homozygous $Vgat$-IRES-Cre ($Vgat^{cre/cre}$) knock-in mice in which the IRES-Cre recombinase cassette is inserted downstream of the stop codon of the endogenous vesicular GABA transporter ($Vgat$) gene ($Slc32a1^{tm2(cre)Lowl}$); and C57Bl/6 J mice. $Vgat^{cre/cre}$ mice were bred to $Ubr4^{fl/fl}$ mice, and a breeding colony was maintained by mating $Ubr4^{fl/fl}$ and $Vgat^{cre/+}$;$Ubr4^{fl/+}$ animals. Littermate controls were used wherever possible for experiments. $Vgat^{cre/cre}$ mice were also bred to C57BL/6 J mice to generate $Vgat^{cre/+}$ mice. For the acute jetlag (7-h LD advance) and PER2 immunostaining experiments, data from $Vgat^{cre/+}$ and $Ubr4^{fl/fl}$ controls were pooled as they were phenotypically indistinguishable from one another. All other experiments utilized only $Ubr4^{fl/fl}$ mice as controls. Unless otherwise specified, mice were bred and maintained on a fixed 12-h light:12-h dark (12:12 LD) schedule in which lights on and lights off corresponded to 7 a.m. and 7 p.m. Eastern Standard Time, respectively. The animals were maintained at 40–60% humidity and 20–24 °C. Lighting conditions in animal housing rooms were 100–150 Lux (white light) measured at room floor level. Lighting conditions in behavioral cabinets were as defined in each experiment. Air exchanges in the animal housing rooms and behavioral cabinets were maintained at 15–20 ACH. $Vgat^{cre/+}$;$Ubr4^{fl/fl}$ mice were born at the expected Mendelian ratio, were viable, and appeared grossly normal despite having a slightly reduced body weight compared to their sex-matched littermates at the time of weaning.

**Fly strains and rearing**. Details of fly lines used in this paper are listed in Supplementary Table 4.

All fly strains were reared on food containing agar, glucose, sucrose, yeast, cornmeal, wheat germ, soya flour, molasses, propionic acid, and Tegosept on a 12:12 LD schedule at 25 °C, 50% relative humidity. UAS and GAL4 controls were used as heterozygotes after crossing to $w^{1118}$ as the wild-type strain. Unless otherwise stated, male flies were used for all experiments. The genotypes of experimental flies and controls are as follows:

$w^{1118};UAS\text{-}poe^{RNAi}/+;+/+$ ($+>poe^{RNAi}$),
$UAS\text{-}Dcr2; tim\text{-}GAL4/+;+/+$ ($tim>Dcr2$),
$UAS\text{-}Dcr2; tim\text{-}GAL4/UAS\text{-}poe^{RNAi};+/+$ ($tim>Dcr2; poe^{RNAi}$),
$w^{1118}; Pdf\text{-}GAL4/+; UAS\text{-}Dcr2/+$ ($Pdf>Dcr2$),
$w^{1118}; Pdf\text{-}GAL4/UAS\text{-}poe^{RNAi}; UAS\text{-}Dcr2/+$ ($Pdf>Dcr2; poe^{RNAi}$),
$elav\text{-}GAL4; UAS\text{-}Dcr2/+; +/+$ ($elav>Dcr2$),
$elav\text{-}GAL4; UAS\text{-}Dcr2/UAS\text{-}poe^{RNAi};+/+$ ($elav>Dcr2; poe^{RNAi}$),
$w^{1118}; Pdf\text{-}GAL4/UAS\text{-}CD2\text{-}HRP; UAS\text{-}Dcr2/+$ ($Pdf>Dcr2; CD2\text{-}HRP$),
$w^{1118}; Pdf\text{-}GAL4/UAS\text{-}poe^{RNAi}, UAS\text{-}CD2\text{-}HRP; UAS\text{-}Dcr2/+$ ($Pdf>Dcr2; poe^{RNAi}, CD2\text{-}HRP$).

**Stereotaxic injections**. Details of AAV1 constructs used in stereotaxic injections are listed in Supplementary Table 4.

Eight–12-week-old $Ubr4$ cKO male mice were anesthetized with ~2% (vol/vol) isoflurane in $O_2$. Toe pinch responses (or the lack of a paw withdrawal reflex) were frequently assessed to ensure that a surgical depth of anesthesia was achieved. A thin layer of eye ointment (Systane Nighttime Lubricant) was applied to the eyes to prevent them from drying. The scalp was cut open and a craniotomy of ~300–500 μm diameter was made with a 0.3 mm Tungsten bur (Gesswein 122–2008). Through this cranial window, a beveled glass pipette (tip size ~30 μm) was inserted into the SCN using the following stereotaxic coordinates (AP: 0.4 mm posterior from bregma, ML: ±0.2 mm from the midline, DV: 5.03 mm from dura with head level), with minor adjustments based on the distance between bregma and lambda. A viral particle suspension of AAV1-CMV-m-CORO7-2A-mCherry or AAV1-CMV-mCherry (control) was injected at a speed of 55 nl/min in a total volume of 350 nl into each unilateral SCN using an UltraMicroPump (UMP3, WPI) to which the glass pipette was attached. The scalp was then sutured with 6–0 silk suture (CP Medical 667S). Mice were set up in wheel cages for activity recordings two weeks after the viral injection. At the end of the ChrA$^{6/2}$ schedule, SCN tissues were harvested from the injected mice at 16–20 weeks of age.

**Circadian behavioral analyses**. Details of software used for behavioral analyses in this paper are listed in Supplementary Table 4.

Unless otherwise specified, at 5–8 weeks of age, male mice were individually housed in running-wheel cages within ventilated, light-tight cabinets under computer-controlled light schedules (Phenome Technologies). Cage-level light intensity was set at 40 Lux (0.5 μE), except where indicated, using white light-emitting diodes (LEDs) installed within the cabinets. Wheel revolutions were recorded and analyzed using Clocklab software (Actimetrics). In all experiments, mice were initially entrained to a 12:12 LD cycle for at least 2 weeks prior to the indicated changes in the light schedule. To assess free-running rhythms, entrained mice were released into constant darkness (DD) for 2 weeks. To determine the effect of a brief, early-night light pulse, mice received a 15-min light pulse (LP: 40 Lux [0.5 μE]) at CT 15 after ~2 weeks in DD. Mice were maintained in DD for an additional 2 weeks following the LP to measure the phase shift through extended fitted regression lines before and after the LP. To determine the effects of constant light (LL), mice were subjected to LL in which the light intensity was increased in a stepwise fashion (5 Lux [0.06 μE], 10 Lux [0.13 μE], 20 Lux [0.25 μE], 40 Lux [0.5 μE], 80 Lux [1 μE], 120 Lux [1.5 μE]) every ~2–3 weeks. For the acute jetlag paradigm, mice were subjected to an abrupt 7-h advance of the 12:12 LD schedule for ~3 weeks, and for some mice this was followed by an abrupt 7-h delay of the LD schedule for an additional 3 weeks. To assess the effects of chronic jetlag (ChrA$^{6/2}$), mice were subjected to a 6-h advance in the 12:12 LD schedule every

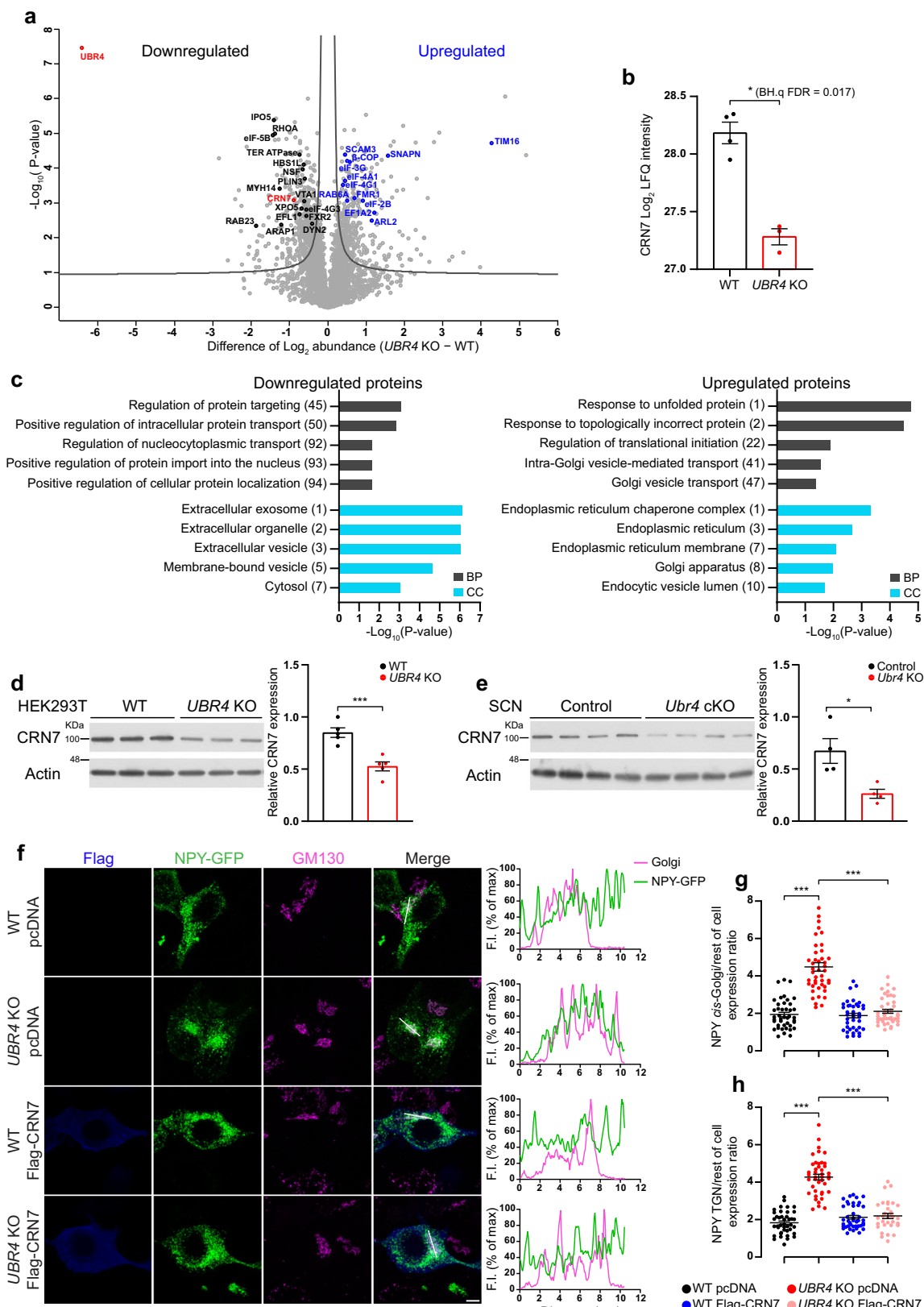

2 days for 4 weeks, after which they were released into DD. Times of activity onset were generated by Clocklab and corrected where necessary after visual inspection of the actograms. The onset of activity was considered as the first bin where 20% of maximum activity during that cycle was displayed, and the offset was the last 5-minute bin with 5% of peak activity. Depending on the light paradigm assessed, the mean of the total daily wheel revolutions was calculated by taking the average of daily wheel revolutions for 7 (LD, DD) or 12 (LL) consecutive days. The period was measured by fitting a regression line through daily activity onsets or by using the $\chi^2$ periodogram. The DD period reported in Fig. 1d was measured from the 10 days preceding the CT15 LP. Amplitude was measured by $\chi^2$ periodogram and Fast Fourier

**Fig. 7 UBR4 regulates the abundance of Coronin 7 to promote cargo export from the Golgi. a** Volcano plot showing proteins that are differentially expressed in *UBR4* KO HEK293T cells relative to WT controls (two-tailed unpaired *t* test with a Benjamini-Hochberg (BH) adjustment, BH.q FDR < 0.05, S0 = 0.1). Proteins involved in vesicular trafficking or mRNA translation are highlighted in black or blue. **b** Log$_2$ LFQ intensity values for CRN7. *n* = 4 WT and 3 *UBR4* KO. Two-tailed unpaired *t* test with Benjamini-Hochberg FDR set at 0.05. BH.q FDR = 0.017. **c** FAT GO enrichment analysis of differentially expressed proteins in *UBR4* KO HEK293T cells. Biological process, BP. Cellular component, CC. The number in parenthesis indicates the position of the corresponding term in the ranked list (one-sided Fisher's exact test, *p* < 0.05). **d**, **e** Western blot of endogenous CRN7 in *UBR4* WT and KO HEK293T cells (**d**) or SCN tissues of control and *Ubr4* cKO mice (**e**). Graphs show relative CRN7 expression normalized to actin. *n* = 5 (**d**) or 4 (**e**) per genotype. ***p* = 0.0009 (**d**), **p* = 0.0174 (**e**), two-tailed unpaired *t* test. **f** NPY-GFP (green) and GM130 (magenta) immunofluorescence in *UBR4* WT and KO HEK293T cells transfected with either pcDNA empty vector or Flag-CRN7 (blue). Profile plots (right) show NPY-GFP and GM130 fluorescence intensity (F.I.) along the reference axis (white line, merged panel). Scale bar, 5 μm. **g**, **h** Ratio of NPY-GFP abundance in the *cis*-Golgi (**g**) or TGN (**h**) relative to the rest of the cell. *n* = 42 (**g**) or 41 (**h**) for WT pcDNA, *n* = 43 for *UBR4* KO pcDNA, *n* = 40 (**g**) or 42 (**h**) for WT Flag-CRN7, *n* = 43 (**g**) or 32 (**h**) for *UBR4* KO Flag-CRN7. ***p* < 0.0001, Kruskal–Wallis with a Dunn's post hoc (**g**) or one-way ANOVA with Bonferroni's post hoc (**h**). Values represent mean ± SEM. "*n*" represents the number of samples (**b**, **d**), the number of mice (**e**), or the number of cells (**g**, **h**).

Transformation (FFT). Arrhythmicity in LL was determined by χ$^2$ periodogram. The phase angle of entrainment was measured by calculating the difference between the time of activity onset and lights-off for 6 consecutive days.

Locomotor activity of individual male flies was recorded using the *Drosophila* Activity Monitoring (DAM) System (Trikinetics). Briefly, 7-day-old male flies were individually aspirated into glass tubes containing 4% sucrose/2% bactoagar media on one end and capped with a cotton plug on the other end. The tubes were then placed in DAM monitors in a 25 °C incubator, and the flies were allowed to entrain to a 12:12 LD cycle for ~5 days before being released into DD for an additional 1–2 weeks. Activity data were recorded for individual flies as the number of infrared-beam crosses per minute (1-min bins). The fly toolbox[56] implemented in Matlab (Mathworks) was used to generate actograms, and Clocklab software (Actimetrics) was used to analyze rest:activity rhythms. The free-running period under DD was analyzed using χ$^2$ periodogram, and FFT power was used to measure the rhythm strength of individual flies. A fly was deemed to be rhythmic if the χ$^2$ periodogram showed a peak above the 95% confidence interval and the FFT power was >0.01. Behavioral scoring (Fig. 4f) was achieved using the abovementioned criteria as well as by visual assessment of the actograms by an observer blind to the genotype of the flies. LD activity line plots (Supplementary Fig. 2c, d, g, h) were generated by averaging the minute-by-minute activity of individual flies during the last 3 days of LD, converting the activity data of individual flies into 1 h bins, and then calculating the average activity of flies within each genotype. For an optimal depiction of LD anticipation, the activity levels were normalized to the maximum activity level in each genotype and presented as "relative activity." LD anticipation indices (AI) were calculated as follows: Morning AI = "sum of activity in the 3 h preceding lights-on/sum of activity in the 6 h preceding lights-on," and evening AI = "sum of activity in the 3 h preceding lights-off/sum of activity in the 6 h preceding lights-off." AI values that are significantly >0.5 represent anticipatory behavior[44].

**Tissue harvest**. Unless otherwise specified, mice (both sexes, 5–8 weeks of age) were maintained on a fixed 12:12 LD schedule and released into DD for two consecutive cycles prior to further treatment and tissue harvest on day 3 of DD. Mice were killed by cervical dislocation and their brains were rapidly dissected under dim red light. Brains were sectioned in ice-cold oxygenated media or diethyl pyrocarbonate-treated phosphate-buffered saline (PBS), pH 7.4 (for qRT-PCR and RNAscope experiments) with an oscillating tissue slicer (Electron Microscopy Sciences) to obtain an 800-μm-thick coronal slice containing the SCN. For immunostaining experiments, tissue slices were fixed in 4% paraformaldehyde (PFA) in PBS (pH 7.4) for 6 h at room temperature (RT), cryoprotected in 30% sucrose in PBS at 4 °C overnight (O/N), cut into 30-μm thin sections using a freezing microtome (Leica Microsystems), and stored in 30% sucrose at 4 °C until further use. For electron microscopy, the 800-μm coronal slice was fixed in 4% PFA and 1% glutaraldehyde in PBS (pH 7.2) for 1 h at RT, followed by an O/N incubation in fixative at 4 °C. For Western blotting and qRT-PCR experiments, the 800-μm coronal slice was placed on a dry ice-chilled glass slide and the SCN was microdissected using a sharp scalpel blade. SCN tissues were frozen immediately on dry ice and stored at –80 °C until further processing. For Western blotting and qRT-PCR experiments, each sample represented SCN tissues from 1 or 2 mice, respectively. For RNAscope in situ hybridization (ISH) experiments, tissue slices were fixed in 4% PFA in RNase-free PBS (pH 7.4) for 24 h at 4 °C, cryoprotected in 30% sucrose in PBS at 4 °C O/N, and then frozen in Tissue Tek Optimal Cutting Temperature (OCT) embedding media with dry ice. Tissue blocks were cut into 14-μm thin sections using CryoStar NX50 Cryostat (ThermoFisher Scientific) and thaw-mounted on SuperFrost Plus slides. Slides were air-dried for 2 h and stored at −80 °C until further use.

For adult fly brain dissections, all flies were maintained under a 12:12 LD cycle at 25 °C, released into DD for two consecutive cycles (if required), and dissected at the indicated Zeitgeber time (ZT) or CT (on day 3 of DD). Adult flies were anesthetized with CO$_2$, pinned down, and dissected in a dish containing ice-cold PBS (pH 7.4, RNase-free). For qRT-PCR experiments, dissected brains from ~2-4-

day-old female flies were immediately transferred to cell lysis buffer for RNA isolation and stored at −80 °C until further processing. Each sample contained a pool of ~15 brains. For ISH experiments, dissected whole brains from ~17–18-day-old male and female flies were fixed in 4% PFA in PBS (pH 7.4, RNase-free) at RT for 30 min, and then immediately processed for ISH. Each sample contained ~7–10 brains for whole-mount ISH. For immunostaining and CD2-HRP detection experiments, whole brains from ~17–18-day-old male flies were fixed in 4% PFA in PBS (pH 7.4) for 20 min on ice, followed by an additional 20 min at RT in fresh fixative. Brains were rinsed in PBS (pH 7.4), and subsequently processed for immunostaining. Each sample contained ~8–15 brains for whole-mount staining. For each experiment, all brains were processed at the same time and treated in an identical manner.

**RNAscope in situ hybridization**. Details of the RNAscope kit and ISH probes used in this paper are listed in Supplementary Table 4.

All ISH probes were purchased from Advanced Cell Diagnostics, and the RNAscope Multiplex Fluorescent Reagent Kit v2 was used according to the manufacturer's instructions. For *Ubr4* ISH, slides with mouse SCN sections were washed in PBS for 5 min, baked at 60 °C for 30 min, fixed with 4% PFA in PBS for 90 min at RT, dehydrated in increasing concentrations of ethanol (50%, 70%, 2 × 100%), and baked again at 60 °C for 10 min. Slides were treated with hydrogen peroxide for 10 min at RT. Antigen retrieval was performed by placing slides into mildly boiling target retrieval reagent for 5 min and then washed in distilled water for 15 sec and 100% ethanol for 3 min. Slides were dried and permeabilized with Protease III for 30 min at 40 °C. Next, slides were rinsed with distilled water followed by probe incubation for 2 h at 40 °C. Afterwards, slides were incubated with a series of amplifier probes and the appropriate horseradish peroxidase (HRP) solution at 40 °C (AMP1, 30 min; AMP2, 30 min; AMP3, 15 min; HRP-C1, 15 min) with 2 × 2 min washes in wash buffer in between each amplification step. Slides were incubated with Opal 570 dye diluted in TSA buffer for 30 min at 40 °C. After blocking HRP activity, sections were counterstained with DAPI and coverslipped with Fluorescence Mounting Medium.

For *poe* and *Pdf* ISH, we followed a modified protocol from the manufacturer that was optimized for whole-mount *Drosophila* brains. Following fixation, brains were washed 2 × 10 min in PBST (PBS, pH 7.4, with 0.1% Tween-20) at RT, dehydrated in increasing concentrations of methanol (25%, 50%, 75%, 100%) in PBST, incubated in 0.2 M hydrochloric acid in 100% methanol for 30 min at RT, and rehydrated in decreasing concentrations of methanol (75%, 50%, 25%) in PBST. Antigen retrieval was performed by adding mildly boiling target retrieval solution to the samples for 15 min. Brains were then immediately transferred to PBST with 1% bovine serum albumin (BSA) for 1 min, rinsed in 100% methanol for 1 min, and washed with PBST + 1% BSA for 5–10 min. Brains were permeabilized with Protease III for 20 min at 40 °C, rinsed in probe diluent, and incubated O/N in the appropriate probes at 40 °C. The next day, brains were washed 2 × 10 min in wash buffer, followed by incubation with a series of amplifier probes and the appropriate HRP solutions at 40 °C (AMP1, 30 min; AMP2, 30 min; AMP3, 15 min; HRP-C1 and HRP-C3, 15 min), with 2 × 10 min washes in wash buffer between every step. Brains were then incubated with Opal 570 and Opal 520 dyes diluted in TSA buffer for 30 min at 40 °C. HRP activity was blocked, and brains were mounted onto glass microscope slides in Fluorescence Mounting Medium.

**Immunohistochemistry (IHC), immunocytochemistry (ICC), and immunofluorescence (IF)**. Details of antibodies, including working dilutions, and other immunostaining reagents are listed in Supplementary Table 4.

For IHC, coronal brain sections were washed 5 × 5 min in PBS (pH 7.4) with 0.1% Triton X-100 (PBST), treated with 0.3% H$_2$O$_2$ in PBS for 20 min at RT, washed 5 × 5 min in PBST, and incubated for 1 h at RT in blocking solution (10% horse serum in PBST). Tissues were incubated O/N at 4 °C in primary antibody diluted in fresh blocking solution. The next day, sections were washed 5 × 5 min in

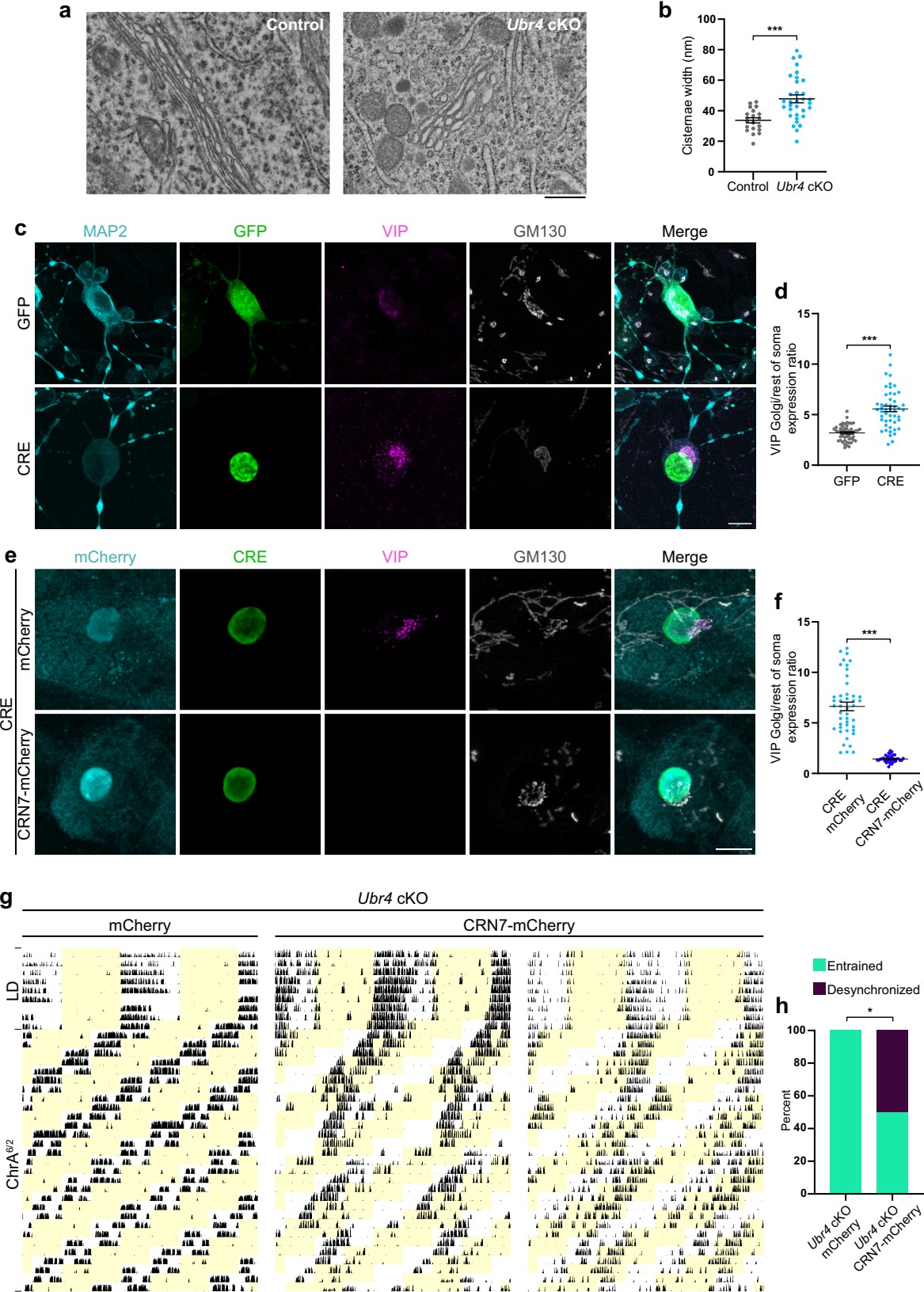

PBST, incubated for 2 h at RT with the appropriate biotinylated secondary antibody diluted in blocking solution, washed 5 × 5 min in PBST, and incubated for 45 min with Avidin-Biotinylated HRP Complexes (ABC). After washing 5 × 5 min in PBST, sections were developed with the 3,3'-diaminobenzidine (DAB) HRP substrate according to the manufacturer's instructions. For each experiment, all sections were processed concurrently and with the same DAB development time.

Sections were mounted onto gelatin-coated glass microscope slides, dehydrated, and coverslipped with Permount Mounting Medium.

To visualize endogenous CD2-HRP expression in flies, following fixation, adult fly brains were rinsed in PBS (pH 7.4), washed 4 × 20 min in PBST (PBS, pH 7.4, with 0.3% Triton X-100) at RT, and then developed with the DAB HRP substrate according to the manufacturer's instructions. For each experiment, all brains were

**Fig. 8 Restoring Coronin 7 abundance in *Ubr4*-deficient SCN neurons rescues VIP trafficking and promotes desynchronized behavior under chronic jetlag. a** TEM images showing the morphology of Golgi cisternae in the SCN of control and *Ubr4* cKO mice. Scale bar, 500 nm. **b** Cisternae width of Golgi stacks in SCN tissues. Each symbol represents data from a single cell; symbol type (circle, diamond) denotes data from different animals. $n = 20$ control and 33 *Ubr4* cKO cells from 2 mice (per genotype). ***$p = 0.0002$, two-tailed unpaired $t$ test. **c, e** Photomicrographs of *Ubr4*^fl/fl SCN neurons that were co-transduced with AAVs expressing VIP along with either Cre-eGFP (CRE) or GFP (**c**). In **e**, cells were transduced with a third AAV construct expressing either mCherry or CRN7-mCherry, in addition to the Cre- and VIP-expressing AAVs used in **c**. Neurons were immunostained for VIP (magenta), GM130 (gray), GFP or CRE (green), and MAP2 or mCherry (cyan). Scale bar, 10 μm. **d, f** Ratio of VIP abundance in the Golgi relative to the rest of the soma for the experiments shown in **c** and **e**. $n = 50$ GFP, 49 CRE, 45 CRE mCherry, 40 CRE CRN7-mCherry. ***$p < 0.0001$, two-tailed Mann–Whitney $U$ test. **g** Actograms showing wheel-running activities of *Ubr4* cKO mice subjected to chronic jetlag (ChrA^6/2) following SCN injections of AAVs expressing mCherry or CRN7-mCherry. **h** Percentage of mice displaying "entrained" (one rhythm, ~21 h period) or "desynchronized" (two rhythms, ~21 and ~24 h periods) behavior under the ChrA^6/2 schedule. $n = 9$ *Ubr4* cKO mCherry and 8 *Ubr4* cKO CRN7-mCherry. *$p = 0.0153$, two-sided chi-square test. Values represent mean ± SEM (**b**, **d**, **f**) or percent (**h**). "$n$" represents the number of cells (**b**, **d**, **f**) or the number of mice (**h**).

processed concurrently and with the same DAB development time. Brains were then dehydrated in a graded series of ethanol (30%, 50%, 70%, 90%, 95%, 100%, 100%), cleared in xylene, and mounted with Permount Mounting Medium.

For ICC, HEK293T cells and primary neurons were plated onto poly-D-lysine-coated glass coverslips in 24-well plates. Cells were washed with ice-cold PBS (pH 7.4), fixed in 4% PFA in PBS for 15 min at RT, washed 2 × 5 min in PBS, washed 5 × 5 min in PBS (pH7.4) with 0.1% Triton X-100 (PBST), incubated for 1 h in blocking solution, and incubated O/N at 4 °C in fresh blocking solution containing the primary antibody. The next day, coverslips were washed 5 × 5 min in PBST and incubated with the appropriate AlexaFluor secondary antibody (1:1000) diluted in blocking solution, in the dark for 2 h at RT. Subsequently, coverslips were washed 5 × 5 min with PBST, incubated for 10 min in DAPI (1:10,000) diluted in PBS where applicable, and washed 2 × 5 min in PBS. Coverslips were mounted onto glass microscope slides with Fluorescence Mounting Medium and sealed with nail polish.

For IF of mouse tissues, following fixation, coronal mouse brain sections were washed 5 × 5 min in PBS (pH 7.4) with 0.1% Triton X-100 (PBST), and incubated for 1 h at RT in blocking solution (10% horse serum in PBST). Tissues were incubated O/N at 4 °C in primary antibody diluted in fresh blocking solution. The next day, tissues were washed 5 × 5 min with PBST and incubated for 2 h at RT, protected from light, with the appropriate secondary AlexaFluor antibody diluted in blocking solution. Sections were washed 5 × 5 min with PBST, incubated for 10 min in DAPI (1:10,000) diluted in PBS, washed twice in PBS, and mounted onto glass microscope slides. Slides were coverslipped with Fluorescence Mounting Medium and sealed with nail polish.

For IF of fly tissues, following fixation, adult fly brains were rinsed in PBS (pH 7.4), washed 4 × 20 min in PBST (PBS, pH 7.4, with 0.3% Triton X-100) at RT, and incubated in blocking solution (5% horse serum in PBST) for 2 h at RT. Subsequently, brains were incubated in a fresh blocking solution containing primary antibodies for 2 h at RT, followed by a 4 °C incubation for 40–48 h. After washing 5 × 20 min in PBST, brains were incubated O/N with the appropriate AlexaFluor secondary antibody (1:100 or 1:1000) diluted in blocking solution, in the dark at 4 °C. The next day, brains were washed 4 × 20 min in PBST followed by 2 × 10 min in PBS (pH 7.4). All incubations were done on a rocker set at low speed. Brains were mounted onto glass microscope slides in Fluorescence Mounting Medium.

**Tissue processing for transmission electron microscopy.** Fixed coronal slices were washed 3 × 20 min in PBS (pH 7.2) to remove residual aldehydes. Sections were post-fixed in 1% osmium tetroxide in PBS (pH 7.2) for 90 min at RT and subsequently washed 2 × 30 min in PBS. Slices were dehydrated in increasing concentrations of ethanol at RT (50% ethanol for 20 min, 70% ethanol for 30 min, 90% ethanol for 60 min, 100% ethanol O/N). SCN tissues were then micro-dissected, embedded in Epon resin, and polymerized for 48 h at 40 °C. Ultrathin sections (90 nm) were cut, collected on single-slot formvar-coated copper grids, and stained in uranyl acetate and lead citrate. Ultrathin SCN sections were imaged using a Hitachi HT7700 transmission electron microscope at 80 kV.

**Cell culture, plasmid transfections, and viral transductions.** Details of reagents and AAV1 constructs used in cell culture experiments are listed in Supplementary Table 4.

Human embryonic kidney (HEK) 293T and mouse Neuro-2a cells were cultured in Dulbecco's Modified Eagle Medium (DMEM) supplemented with 10% heat-inactivated Fetal Bovine Serum (FBS) and 1% penicillin-streptomycin at 37 °C and 5% CO_2. Cells were transfected with plasmid constructs or siRNAs using Lipofectamine 3000 in DMEM and returned to complete growth media 4–6 h after transfection. Unless otherwise specified, subsequent experimental manipulations or cell harvests were performed 24–48 h post-transfection. For pharmacological studies, HEK293T cells were incubated in growth media supplemented with DMSO (vehicle), 2–20 μM MG132 for 6–18 h, 500 nM Bafilomycin A1 for 6 h, or 10 μg/mL E64D along with pepstatin A for 10 h.

For primary SCN neuronal cultures, SCN tissues from P0 *Ubr4*^fl/fl mouse pups were rapidly dissected and placed in dissection media (HBSS [Ca$^{2+}$ and Mg$^{2+}$ free] supplemented with 1 mM sodium pyruvate, 0.1% glucose, and 10 mM HEPES [pH 7.3]), dissociated with 0.125% trypsin for 20 min in a 37 °C water bath, and triturated with fire-polished glass pipets. Cells were plated onto glass coverslips placed inside the wells of a 24-well plate or onto the plastic wells directly, in plating media (MEM Eagle with Earle's BSS supplemented with 5% FBS, 2 mM glutamine, 1 mM sodium pyruvate, 0.45% glucose, and penicillin/streptomycin) as described by Beaudoin et al.[57]. Glass or plastic surfaces were coated with 1 mg/ml or 100 μg/ml poly-D-lysine, respectively, prior to plating. Unless otherwise stated, neurons were plated at densities of $2 \times 10^5$ cells per well of a 24-well plate. Three to 6 h later, the plating media was removed and replaced with neuronal growth media (Neurobasal media supplemented with 2% B27, 1 mM glutamine, and penicillin-streptomycin). A half media change was performed every 2 days. Primary neuronal cultures were transduced with AAV1 constructs on day in vitro (DIV) 3. The following multiplicities of infection (MOIs) were used: 5000 MOI for hSyn-CRE, hSyn-GFP, CMV-m-CORO7-2A-mCherry, and CMV-mCherry, and 500 MOI for hSyn-m-VIP. Unless otherwise specified, subsequent experimental manipulations or cell harvests were performed on DIV 10.

**Plasmid construction.** Details of plasmids purchased or generated in this paper are listed in Supplementary Table 4.

To generate a CMV-driven Streptavidin-KDEL/SBP-EGFP-NPY construct for the RUSH experiments, human NPY was PCR-amplified from the NPY-GFP plasmid template (Addgene plasmid #74629), and cloned into the Streptavidin-KDEL/SBP-EGFP-GPI plasmid (Addgene plasmid #65294), using the FseI and XhoI sites to place the NPY insert in-frame with the EGFP cassette. Primer sequences can be found in Supplementary Table 5.

Expression plasmids encoding wild-type or mutant human UBR4 fused to an IRES-mCherry sequence were generated using Gateway LR cloning (Supplementary Table 5). Mutant UBR4 harbors the Ubr box[15,32] encoding either multiple Ala mutations or a partially deleted sequence. To construct pENTR3C_hUBR4_ubrbox_Ala and pENTR3C_hUBR4_ubrbox_del entry clones, the previously reported entry clone pENTR3ChUBR4[32] was digested with KpnI and CsiI and ligated with a commercially synthesized DNA fragment (Genscript, Piscataway, NJ, USA) (sequences in Supplementary Table 5).

**Western blotting.** Details of antibodies, including working dilutions, and other Western blotting reagents are listed in Supplementary Table 4.

SCN tissues or cultured cells were lysed in ice-cold RIPA buffer (10 mM Tris-HCl pH 8.0, 1 mM EDTA, 1% NP-40, 0.1% sodium deoxycholate, 0.1% SDS, 140 mM NaCl, Protease Inhibitor Cocktail). Lysates were centrifuged at $13,000 \times g$ for 20 min, and the protein concentration of the supernatant was determined using the Bradford method. Protein samples were resolved by SDS-PAGE using 4–20% Mini-PROTEAN TGX precast gels and transferred onto PVDF membranes. Membranes were cut at approximately the position of the 63 kDa band of the ladder; the top half of the membrane was probed for UBR4/CRN7/V5 and the bottom half for actin. Membranes were washed 5 × 5 min in TBST (20 mM Tris base, 150 mM NaCl, 0.1% Tween-20) and incubated for 1 h at RT in blocking solution (5% skim milk in TBST). Membranes were incubated O/N at 4 °C with primary antibodies diluted in blocking solution. The next day, membranes were washed 5 × 5 min with TBST, incubated for 2 h at RT in HRP-conjugated secondary antibodies, and washed 5 × 5 min in TBST. Signals were detected by chemiluminescence using the SuperSignal West Femto Maximum Sensitivity Substrate and developed on X-ray film. To quantify relative protein expression, the polygon selection tool on the ImageJ software was used to delineate individual bands, and the "measure" function was used to determine band intensity. Results are presented as values normalized to the loading control, actin. Uncropped blots for all experiments are shown in the main figures or, alternatively, in the Source Data file.

**Enzyme-linked immunosorbent assay (ELISA)**. Details of kits and plasmids used for ELISA assays are listed in Supplementary Table 4.

WT and *UBR4* KO HEK293T cells were co-transfected with NPY-GFP and CMV-mCherry constructs. Culture media was replaced with fresh growth media 48 h post-transfection, and collected 2 h later along with the cell lysates. GFP and mCherry ELISA kits (Abcam) were used to quantify the amount of NPY-GFP in the culture media and the amount of mCherry in the cell lysates, respectively. For every experimental well, relative NPY-GFP secretion was calculated by normalizing the amount of NPY-GFP in the culture media to the amount of mCherry in the cell lysate.

**Polysome profiling**. WT and *UBR4* KO HEK293T cells were replated and grown overnight in growth media. The next day, cells were treated with 100 μg/ml cycloheximide (Sigma) for 5 min prior to lysis in a hypotonic buffer containing 2.5 mM MgCl$_2$, 5 mM Tris-HCl (pH 7.5), 1× protease inhibitor cocktail (EDTA-free; Roche), 1.5 mM KCl, 2 mM DTT, 200 U/ml RNaseIn, 0.5% (v/w) sodium deoxycholate, and 0.5% (v/w) Triton X-100. Four hundred micrograms of the isolated ribonucleoproteins were sedimented on a 10–50% sucrose gradient using ultracentrifugation at 250,000 × g in a SW40 rotor (Beckman Coulter) at 4 °C for 2 h. The sediments were then fractionated using an ISCO gradient fractionation system while continuously recording the optical density at 260 nm with a FOXO JR Fractionator. RNA was then isolated from each fraction using TRIzol (Invitrogen) according to the manufacturer's instructions and subsequently used for qRT-PCR analysis.

**RNA extraction and qRT-PCR**. Details of reagents used for RNA extraction and qRT-PCR experiments are listed in Supplementary Table 4.

Total RNA was extracted from SCN tissues and HEK293T cells using the RNeasy Micro Kit and TRIzol Reagent, respectively, according to the manufacturer's instructions. For the polysome profiling experiment, TRIzol-extracted RNA from 14 individual fractions (#1 to #14) were pooled to generate 5 fractions ('A' to 'E') for subsequent cDNA synthesis: A (#1 to #3), B (#4 and #5), C (#6 and #7), D (#8 to #10), and E (#11 to #14). cDNA was synthesized using SuperScript IV Reverse Transcriptase, and qPCR was performed using SsoFast EvaGreen Supermix on a Bio-Rad CFX384 Real-Time PCR Detection System. Except for the polysome profiling study, relative transcript abundance was calculated using the delta-delta CT method and normalized to *Gapdh*. For the polysome profiling experiment, the amount of *CRN7* or *GAPDH* transcripts in each pooled fraction was calculated as a percentage of the total amount of the transcript detected in all fractions.

Total RNA was extracted from fly brain tissues using the RNeasy Micro Kit. cDNA was synthesized using iScript cDNA Synthesis Kit, and qPCR was performed using iTaq Universal SYBR Green Supermix. *Rp49* was used to normalize relative transcript expression. All primer sequences used for qPCR analyses are listed in Supplementary Table 5.

**Time-lapse imaging**. Details of reagents and software used for time-lapse image acquisition and processing are listed in Supplementary Table 4.

HEK293T cells seeded on 4-chamber glass slides were transduced with Golgi-RFP BacMam 2.0 one day after transfection with the Streptavidin-KDEL/SBP-EGFP-NPY plasmid. The next day, cells were supplemented with fresh phenol red-free growth media, and the chamber slide was transferred to a stage-top incubator (Live Cell Instrument Inc.) where the cells were maintained at 37 °C and supplied with humidified air and 5% CO$_2$ throughout the time-lapse imaging session. At recording time = 0', the media was removed, and replaced with fresh media supplemented with cycloheximide (100 μg/mL) and D-biotin (40 μM). Confocal imaging was performed using a Leica DMi8 inverted microscope (Leica) equipped with X-Light V2 spinning disk confocal (CrestOptics), laser diode illuminator-7 (89 North), and a prime HS:sCMOS camera (Photometrics). The multipoint acquisition was achieved through the MS-2000 motorized XY stage (Applied Scientific Instrumentation), while axial sectioning was obtained with the MS-2000 piezo top-plate (Applied Scientific Instrumentation). Images were acquired with a 63X/1.40 NA oil objective and Metamorph software (Molecular Devices). Design of the acquisition journals and system integration were done by Quorum Technologies. All subsequent image processing was performed on ImageJ.

**Confocal and bright-field image acquisition and analysis**. Details of software used for image acquisition and analysis are listed in Supplementary Table 4.

*Image acquisition*. Images were acquired using a Zeiss Axio Observer Z1 inverted microscope equipped with a Laser Scanning Microscope (LSM) 700 module for confocal images and an AxioCam MRm Rev.3 monochromatic digital camera (Zeiss) for bright-field pictures, with the Zen 2010 software (Zeiss). Identical settings were used for imaging samples within each experiment (gain, pinhole, and filter sets for confocal microscopy; light intensity and exposure time for light microscopy). Confocal images were acquired in separate channels for each fluorophore. All image analyses were performed on ImageJ. In Fig. 8e, due to the high intensity of the CRE signal (labeled with AlexaFluor 405) and the resulting moderate bleed-through into the other channels, the bleed-through signal originating

from the CRE channel was subtracted from the VIP channel (labeled with AlexaFluor 488) and the GM130 channel (labeled with AlexaFluor 647) using ImageJ's "image calculator" post-acquisition and prior to data analysis.

*Intensity measurements*. For quantification of staining intensity in the SCN, 10X bright-field images of the bilateral SCN were acquired. The area of each SCN was delineated using the polygon selection tool, and the average optical density was obtained using the "measure" function. For quantification of staining intensity/area in HEK293T cells, cultured mouse primary neurons, and fly brains, confocal images were acquired, the region of interest (ROI) was delineated using the polygon selection tool, and the fluorescence intensity/area was quantified using the "measure" function. For quantification of *Pdf*/PDF fluorescence intensity in projections and cell bodies of clock neurons, 10X and 40X confocal images were acquired, respectively. For quantification of PDF fluorescence intensity in fly brains, the entire PDF-immunoreactive cytoplasmic region of the cell, excluding the nucleus, was delineated as the ROI and quantified. For quantification of PER fluorescence intensity in fly brains, the entire PER-immunoreactive region of the cell was delineated as the ROI and quantified. When PER intensity was severely reduced or absent, PDF staining was used to delineate individual clock cells. For quantification of staining intensity and *cis*-Golgi/TGN area in HEK293T cells and primary neurons, 40X or 63X confocal images were acquired and the Golgi complex was delineated using Golgi marker stains (GM130 or p230). Primary neurons were delineated with either the neuronal marker, MAP2, or mCherry. For fluorescent signals, confocal images were converted to maximum intensity projections prior to intensity measurements. For all intensity measurements, background staining was measured in a non-immunoreactive region adjacent to the ROI and subtracted from the immunoreactive intensity of that region.

*Cell counts*. For immunoreactive cell counts, the SCN area was first measured and cropped using the polygon tool. For VIP$^+$, AVP$^+$, and PER2$^+$ cell counts, 10X or 20X bright-field images of the SCN were converted to binary and processed with the "Watershed" filter to delineate individual cells. Immunoreactive cells at each threshold were counted using the "analyze particle" function with a size set at ≥30 μm$^2$. The number of PDF-positive cells was manually scored from the acquired confocal images by an experimenter blind to genotype.

**Label-free mass spectrometry (MS)**. Cell processing for MS, LC-MS analyses, and database search was performed by the Donnelly Mass Spectrometry Center at the University of Toronto. Details of resources and software used in MS experiments and data analyses are listed in Supplementary Table 4.

*Cell culture and sample processing*. Four wild-type and four *UBR4* KO HEK293T cell samples were processed. Briefly, HEK293T cells seeded on 15-cm dishes were washed once in ice-cold PBS (pH 7.4). Cells were then scraped in fresh PBS, pelleted, snap-frozen, and stored at –80 °C until further processing. All subsequent processing steps were performed at 4 °C unless otherwise stated. AFC buffer (10 mM Tris-HCl, pH 7.9, 420 mM NaCl, 0.1% NP-40) supplemented with protease inhibitors (Sigma, S8830-20TAB) and phosphatase inhibitors (10 mM NaF, 1 mM sodium orthovanadate, 2 mM sodium pyrophosphate) was added to frozen cell pellets. To lyse the cells, samples were subjected to three freeze-thaw cycles by transferring the samples between ethanol/dry ice and a 37 °C water bath, with frequent mixing to prevent sample temperature from rising above 4 °C. Lysates were then sonicated (20x, 0.3 s on, 0.7 s off) and centrifuged at 15,000 × g for 30 min at 4 °C. Following protein quantification using a NanoDrop spectrophotometer (absorbance measured at 280 nm), protein samples were prepared at 1 mg/ml stocks, precipitated using the ProteoExtract kit (MillliporeSigma), dried, and subsequently subjected to a trypsin digestion protocol. Specifically, dried samples were reconstituted in 50 mM ammonium bicarbonate, reduced using 2 mM TCEP-HCl, alkylated with 11 mM iodoacetamide, and digested with 5 μg trypsin O/N at 37 °C. Peptides were acidified using acetic acid, desalted using C-18 ZipTips, and dried prior to LC-MS analysis.

*LC-MS analyses*. Peptides were reconstituted in 20 μl of 1% formic acid and 5 μl was loaded onto the column. Peptides were separated on a reverse-phase Acclaim PepMap trap column and EASY-Spray PepMap analytical column using the EASY-nLC 1200 system (Proxeon). The organic gradient was driven by the EASY-nLC 1200 system using buffers A and B. Buffer A contained 0.1% formic acid (in water), and buffer B contained 80% acetonitrile with 0.1% formic acid. The separation was performed in 180 min at a flow rate of 220 nl/min, with a gradient of 5 to 25% buffer B in 155 min, followed by 25 to 100% buffer B in 9 min, and 100% buffer B for 15 min. Eluted peptides were directly sprayed into a Q Exactive HF mass spectrometer (ThermoFisher Scientific) with collision-induced dissociation (CID) using a nanospray ion source (Proxeon). The EasySpray ion source (ThermoFisher Scientific) was used to directly ionize peptides injected into a Q Exactive HF mass spectrometer (ThermoFisher Scientific). For each selected MS1 full scan mass spectrum in profile mode, 20 MS2 data-dependent scans were acquired with HCD fragmentation at 32% normalized collision energy. For MS, the maximum injection time was 70 ms, and for MS/MS it was 25 ms. The dynamic exclusion range was set to 15 s. The full MS scan ranged from 300 to 1650 *m/z* and was followed by a data-dependent MS/MS scan of the 20 most intense ions. The resolutions of the full MS and MS/MS spectra were 60,000 and 15,000, respectively. The data-dependent

mode was used for MS data acquisition with target values of 3E + 06 and 1E + 05 for MS and MS/MS scans, respectively. All data were recorded with the Xcalibur software 4.1.31.9 (ThermoFisher Scientific).

*Data processing and bioinformatic analysis.* Raw MS files were processed using MaxQuant (version 1.6.6.0). Database search was performed with the built-in Andromeda (version 1.6.6.0) search engine using the decoy human Uniprot database (release date 2011_07, decoy mode set to 'revert'). The following parameters were used: methionine oxidation (M) and protein N-terminal acetylation were set as variable modifications, and carbamidomethylation of cysteine residues was set as a fixed modification. Enzyme specificity was set to trypsin/P, with a maximum of 2 missing cleavages allowed. The "match between runs" option was enabled in order to transfer identification between different LC-MS runs based on the peptides' mass and retention time following retention time alignment. The search mass tolerance was set at 20 ppm for the first search, and 4.5 ppm for the main search. A false discovery rate (FDR) of 1% was used to filter the data at the peptide and protein levels and a minimum length of seven amino acids was used for peptide identification.

The proteinGroups file was imported to Perseus (version 1.6.6.0) for differential protein analysis. First, the raw dataset (5,416 proteins) was filtered to remove proteins that were only identified by site, reverse, and contaminant proteins. This resulted in a truncated list of 5,326 proteins. The label-free quantification (LFQ) intensities were then Log2-transformed, and the dataset filtered to only include proteins with valid entries in at least 4 MS measurements, resulting in a list of 4,026 proteins. Missing values were replaced using imputation based on the assumption of the normal distribution with a downshift of 1.8 standard deviations and a width of 0.3 of the original normal distribution. Pairwise Pearson's correlation analysis yielded high $r$ values (>0.90) for all comparisons, indicating good reproducibility within the dataset. To examine differentially expressed proteins (DEPs), a two-sample Student's $t$ test was performed with a Benjamini–Hochberg FDR set at 0.05, and S0 set at 0. KO1 was excluded from the analysis of DEPs due to a high UBR4 LFQ intensity in the sample (indicating potential contamination during sample processing).

FAT Gene Ontology (GO) enrichment analyses were performed using the Database for Annotation, Visualization, and Integrated Discovery (DAVID) Bioinformatics Resources (version 6.8). Fisher's exact test was used to assess significant enrichment in GO Biological Processes and Cellular Components relative to the background dataset of 4026 stringently quantified proteins ($p < 0.05$, and a threshold of at least 3 proteins/GO term).

**Statistics and reproducibility.** The detail of the software used for statistical analyses in this paper is listed in Supplementary Table 4.

Data were analyzed using one- and two-sample $t$ test, one-way analysis of variance (ANOVA), two-way ANOVA, linear mixed-effects modeling, and chi-square test. Post hoc significance of pairwise comparisons was assessed using the Bonferroni test with α set at 0.05. When assumptions of normality were not met as assessed by Shapiro–Wilk test, Mann–Whitney $U$ test, or Kruskal–Wallis test with Dunn's post hoc were used (α set at 0.05). Where applicable, tests were two-tailed. Control and KO/KD groups that lack an asterisk (*) notation were not significantly different. GraphPad Prism (version 8.3.1) was used to perform all statistical analyses. Sample sizes reported for all experiments represent "n" following the exclusion of outliers. All experiments reported in this paper were repeated at least twice with similar results.

**Reporting summary.** Further information on research design is available in the Nature Research Reporting Summary linked to this article.

## Data availability

The mass spectrometry proteomics data generated in this study have been deposited to the ProteomeXchange Consortium (http://proteomecentral.proteomexchange.org) via the PRIDE[58] partner repository with the dataset identifier PXD020630. The human UniProt database used in this study is publicly available at https://www.uniprot.org/help/uniprotkb. All data generated during this study that support our findings are available within the article and its Supplementary Information files. Further information and requests for data, resources, and reagents should be directed to and will be fulfilled by the corresponding authors. Source data are provided with this paper.

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

## Acknowledgements

The authors wish to thank N. Cermakian, R.-H. Chen, I. Edery, M. Holmes, I. Meinertzhagen, M. Nitabach, O. Shafer, P. Taghert, S. Tripathi, and D. Weaver for sharing reagents and equipment. Many thanks to J. Butler, B. Calvieri, Y. Chen, A. Chong, H. Hong, E. Marcon, and S. Pu for technical assistance on TEM, time-lapse imaging, and MS analyses. We are grateful to M. Phillips and all members of the Cheng and Levine laboratories for their insights and helpful discussion. This work was supported by operating grants from the Canadian Institutes of Health Research (CIHR) (grant numbers MOP-286265, PJT-166046, and PJ8-162479, H.-Y.M.C.; PJT-148679-S, J.D.L.; PJT-175131, H.-L.M.C.), the Natural Sciences and Engineering Research Council (NSERC) of Canada (grant numbers RGPIN-2016-05563, H.-Y.M.C.; RGPIN-2016-06218, J.D.L.; RGPIN-2019-06137, H.-L.M.C) and Japan Society for the Promotion of Science (JSPS) Kakenhi (grant numbers JP18K06119 and JP21K06088, T.T.). S.H. and A.H.C. are supported by NSERC post-graduate scholarships. This study is dedicated to the memory of Harrod H. Ling, whose love of chronobiology and all things related to UBR4 inspired me (H.-Y.M.C.) to persevere to the end.

## Author contributions

S.H. designed and conducted experiments, analyzed and interpreted data, prepared figures, and wrote the first draft of the manuscript. A.H.C., H.H.L., J.R.G., S.S., and Z.A. conducted experiments and analyzed data. T.T. generated the UBR4 plasmid constructs. J.L. performed stereotaxic injections under the supervision of B.-h.L. D.A.S. assisted with TEM imaging. T.B. and M.A. conducted the polysome fractionation experiment under the supervision of N.S. S.A. analyzed data. K.N. assisted with fly dissections. J.J.K. assisted with fly experiments and provided supervision. H.-L.M.C. supervised and funded the TEM experiments. J.D.L. supervised and funded the study. H.-Y.M.C. conceived, funded, and supervised the study, designed and conducted experiments, interpreted data, and wrote the final version of the manuscript with input from all authors.

## Competing interests

The authors declare no competing interests.
