## [Peer Review File · Nature Communications]

UBR4/POE facilitates secretory trafficking to maintain circadian clock synchronyREVIEWER COMMENTS

Reviewer #1 (Remarks to the Author):

This is an interesting manuscript that investigates the role of ubiquitin ligase, UBR4/POE, in the control of circadian rhythmicity. The main finding is that the principal role of UBR4/POE is regulating the intracellular traffic of secreted proteins, specifically in the traffic out of the Golgi (and, possibly, also from the ER to the Golgi), and not regulating the stability of the core clock proteins (PER/TIM/CRY). In the context of the circadian clock, the consequences of disrupting UBR4/POE result from the incorrect trafficking of neuropeptides, AVP/VIP and PDF, in the SCN and the fly clock circuit, respectively. The defects are relatively subtle in mice, affecting the speed of re-entrainment. In flies, by contrast, knockdown of POE in clock neurons causes immediate arrhythmicity under conditions of constant darkness (DD) and also defective responses under light:dark entrainment (LD). When knockdown is restricted to the PDF neuropeptide-expressing neurons the behavioral defects mirror those of pdf null mutants. In both organisms lack (or knockdown) of UBR4/POE function causes abnormal accumulation of neuropeptides. Proteomic analyses followed by rescue experiments place the blame for this on defective Coronin 7 function, which is required for budding of Golgi-derived transport vesicles.

The manuscript is a tour de force, combining mouse and fly work, and behavioral, molecular, and proteomic analyses. The text is well written, the narrative makes sense, and the results are well documented and presented. I worked hard to find anything useful to comment. Below are the meager results of my efforts.

Minor comments

1- The results, especially the live imaging, clearly show that UBR4/POE plays an acute function in protein traffic. Yet, it appears that it may also play a role during development, as indicated by the fact that (line 252) "... these flies [tim>Dcr2; poeRNAi] had fewer LNvs than either tim>Dcr2 or poeRNAi controls." It would be interesting to demonstrate this "developmental" function directly by examining the consequences of expressing poe RNAi in these neurons only during development. I recognize that this would mean extra experiments; I leave it to the Editor to decide if they would be required.

2- The manuscript presents UBR4/POE as a protein needed for proper circadian function. Yet, its function is much more general because UBR4/POE is involved in protein trafficking along both the regulated and the constitutive secretory pathways. The relevance of this function to other processes is demonstrated by, for instance, the fact that UBR4 null mutant mice die as embryos, with defects in many organ systems. Given this role, I am not sure that directing the work towards a "clock readership" (at least in the title and the abstract) is such a good idea, since there are many more readers interested in protein trafficking than in clocks. So if I were in the authors' shoes I would use the clock as a sensitive assay to detect defects in protein trafficking, rather than emphasize the role of UBR4/POE on clock function. But I am not in the author's shoes, and they may choose to leave the emphasis intact, which is their prerogative.

Micro-comments

3- Could the authors comment on why ablation of *ubr4/poe* causes damping of PER2 oscillations in the murine SCN and severe damping and phase delay of dPER rhythms in the fly clock neurons? Is this due to changes in the stability of the core clock proteins (PER/TIM/CRY) or is it an indirect effect due to incorrect intracellular neuropeptide traffic?

4- Also, do the authors have an explanation for why the traffic speed from the ER to the Golgi is increased when UBR4 function is impaired?

5- The Supplementary data clearly detail the sources of all reagents including stocks and antibodies. I would ask the authors to refer the reader to this file for information on these sources at the start of the relevant sub-sections of the Methods section.

6- It is not clear how the authors quantified the PER-IR (e.g., Fig. 5 and Supplementary Fig. 3) since the signal is annular. Please provide a few more details (did the ROI consider the entire cell or only a piece of its rim, etc).

7- Regarding Fig. 5 and Supplementary Fig. 3 I would recommend reducing the number of panels. I admit that the data are very beautiful, but showing the pattern of immunoreactivity over many timepoints and for all genotypes means that each image is very tiny. Since the "hard data" are in the associated graphs and histograms, the images are really only illustrative. Thus, I would recommend reducing the number of panels, so that each can be shown in a larger size (as done for other panels that show staining, e.g., Figs. 4, 6 and 7).

Reviewer #2 (Remarks to the Author):

This manuscript by Hegazi et al. demonstrated that UBR4/POE is essential for trafficking of neuropeptides at SCN and exporting peptides from Golgi apparatus. They identified Coronin7 bound to Golgi complex membranes as being regulated by UBR4 for cargo export from the Golgi apparatus. Overall, this work elucidated a novel function of UBR4/POE in secretory trafficking and highlighted the potential for application in circadian rhythms-related diseases. However, the manuscript suffers from a lack of mechanistic details and is largely descriptive in nature. I feel the current manuscript is too premature for publication in Nature Communications.

Major points

1. It is claimed that UBR4 regulates the level of Coronin7(CRN7) on a translational level, as opposed to transcriptional or degradative in nature (361-364). However, the premise of this claim is descriptive

since the authors fail to present convincing data and instead rely on observations that other differentially expressed proteins (DEPs), including eukaryotic transcription factors, are similarly affected by the absence of UBR4. Even if this were the case, the authors fail to show exactly why or how UBR4 deficiency causes such a phenomenon.

2. It is claimed that VIP and AVP trafficking was impaired in UBR4-deficient SCN. While it seems reasonable that UBR4 is essential for trafficking of those neuropeptides, the authors did not address the possibility that UBR4 may not directly affect VIP/AVP trafficking since the synchronization process of SCN is affected from various signals. Defects in synchronization process might be an indirect consequence of UBR4-mediated signaling, in which case the authors should have provided more mechanistic insight.

3. The authors claim that Coronin7(CRN7) mediates exporting cargo from the Golgi apparatus, and that UBR4 regulates the translational level of CRN7. The authors should confirm whether recovery from impaired clock synchrony and chronic jetlag in UBR4-deficient mice is actually due to the decreased levels of CRN7.

4. Even though the authors claim a novel function of the N-recognin UBR4 in secretory trafficking for circadian clock synchrony, they do not show any experiments to confirm whether this novel function is indeed due to the role of UBR4 as an N-recognin. If this novel function is indeed attributable to the N-recognin properties of UBR4, the authors should have tested the role of the UBR box within UBR4 in circadian clock synchrony.

Minor points

1. It is recommended to remove “N-recognin” in the title, as the paper in its current format does not show any evidence that this novel function of UBR4 is related to its N-recognin properties within the N-degron pathway.

2. The word, “ubr4 KO” in HEK293T cell, should be replaced with “UBR4 KO” because HEK293T cell is originated from Homo sapiens.

3. Same as 2., “fmr1” and “Crn7” should be replaced with “FMR1” and “CRN7”.

4. In figure 6, to show generalization to SCN, the authors could show that neuropeptide Y or other secretory proteins were restricted to the Golgi apparatus in SCN of control and Ubr4 cKO mice.

5. The authors mentioned that UBR4 deficiency would have consequences on timely communication between cells (307-309). To confirm that secretion of proteins is affected by depletion of UBR4 associated with exporting system from Golgi apparatus, the author can compare amounts of secreted proteins like NPY from control and UBR4 KO cells.

Reviewer #3 (Remarks to the Author):

The authors have produced an exceptionally thorough study of the loss of function effects for the gene UBR4 (poe) on the circadian timing system underlying rhythmic locomotion. They study the problem in not just one but two different model systems (mice and flies) and analyze the experiments at a multitude of scales, including whole animal behavior, cellular properties, ultrastructure and gene expression. The results appear robust and there is an interesting mix of negative and positive data presented, which increases confidence in the objectivity of the overall evaluation. My only criticism is with the elaboration of a very specific hypothesis that tries to unify all the observations in support of a single molecular mechanism – disruption of neuropeptide trafficking through the regulated secretory pathway via coronin7. The authors are of course entitled to interpret the data according to their judgement as long as the data are fairly presented – and I emphasize that the data are by and large fairly presented. Still the preponderant emphasis on a single mechanism, in the face of general cellular disruption with loss of UBR4, and of discordant results between flies and mice, argues in favor of less strident advocacy for the favored hypothesis. Here I offer a few specific comments and questions in hopes of improving this excellent manuscript.

L 371 “Furthermore, our ubr4 cKO mice and poe KD flies exhibit perturbed behavioral rhythms that are best explained by a central pacemaker with weak network synchrony, a likely consequence of defective neuropeptide trafficking.” This is an interesting hypothesis, and the additional observation on a specific candidate mediator (coronin7) is also interesting, but I feel that a lot of complex cellular physiology is being simplified in pursuit of offering a concise (albeit plausible) mechanism. It is recognized by the authors that both regulated and constitutive pathways are down-regulated (line 306) and that other pathways (like ER to Golgi) are likewise affected. It is further shown that many genes are either up or down regulated in the genetic states studied. It also mentioned that flies with mutations in poe (aka pushover) gene display physiological phenotypes in motorneurons that are consistent with disruptions in trafficking of ion channel components. In other words, many things are changing substantively in neurons lacking this protein.

Behaviorally, effects on mice and flies are substantial in both case but surprisingly different. In mice effects are seen at extreme environmental light conditions (LL), while in flies, they occur preponderantly in DD. The authors argue this may be due to differences in cell number between the two different animals, but that surprising point is simply invoked without substantiation. Furthermore, at least in the case of the fly, the poe behavioral phenotypes do not correspond precisely to what is seen with the absence of PDF, (contrary to what is mentioned on line 224) except in so far as both phenotypes display some delayed arrhythmicity. As far as I can tell, poe KD flies do not display shortened periods. In mice, *ubr4* cKO leads to more rapid adjustment to phase advances, although not phase delays. Importantly, it still takes the *ubr4* cKO mouse a few days to adjust. The authors analogize the rapid adjustment to two studies to argue that similar adjustment phenotypes are shown by deficient mice deficient in the neuropeptide receptors AVPR and VIPR. However those studies indicate a different picture according to my reading. Lack of AVPR leads to more rapid adjustment to both phase advances and phase delays. Lack of VIPR leads to immediate adjustment of locomotor activity phases (unlike the *ubr4* cKO mice) indicative of little contribution by the internal clock and near complete control by masking effects of light.

Moreover, regarding neuropeptide trafficking changes: There is substantial IHC analysis showing a change in the balance of neuropeptide content favoring cell bodies and away from processes, in both flies and mice. That evidence is consistent with the favored hypothesis. However the small LNV neurons display significantly increased neuropeptide content at the subjective night time point, which is anomalous and not consistent. Herraro et al (2020, Current Biology) recently reported that chronic depolarization of these same neurons increased neuropeptide staining levels in the same dorsal terminals at this time point – they argued that depolarization likely recruits increased PDF containing vesicles to the distal aspects. That hypothesis is diametrically opposite to the one favored in the MS under review.

There are rather substantial effects on the molecular oscillator in both systems studied. The authors clearly document these effects but do not emphasize their potential contributions to disruptions of rhythmic locomotor outputs. Especially in the case of the fly, this seems unwarranted. There are also effects on PDF cell number, which implies effects on cell viability. There was no mention of counts for PER+ cells, which could be a useful measure and I believe the data is in hand from the careful systematic time of day analysis of PER staining. Likewise it would be useful to employ a measure of cell morphology (e.g., *cd8-GFP*) distinct from neuropeptide levels to assess POE contributions to overall neuronal differentiation.

All of this leads me to urge more moderation in the enthusiasm with which the authors promote their favored hypothetical mechanism centered on *UBR4*>*coronin7*>Neuropeptides.

Small points

In the images of the fly brain, panel A displaying PDF IHC and poe in situ signals at very low magnification conveys little useful information. In panel B, in situ signals in control and RNAi knockdown conditions indicate a difference but the background levels remain substantial so as to diminish appreciation of “loss” of poe RNA signals.

Line 375 mentions “unpublished” data: I believe that is no longer a supported mechanism

Reviewer #4 (Remarks to the Author):

General Comments

My review for the manuscript NCOMMS-20-34785 “The N-Recognin UBR4/POE facilitates secretory trafficking to maintain circadian clock synchrony” is focused on the mass spectrometry/proteomics contribution to the manuscript. The mass spectrometry data presented is suitable, however more details in the method needs to be included and I have noted some curiosities and weaknesses in the results.

Specific Comments

1. In the results from the label-free quantitative MS where you compare ubr4 KO vs WT HEK293T cells I am curious why out of 471 differentially expressed proteins only one protein, Coronin 7, was of interest.
2. The presented results for this protein are very vague. The manuscript only states that the CRN7 is downregulated and the western blot supports the MS results. While I agree the western does show slightly less abundant bands, however, what are the p-values for both the MS and western data for this protein and what is the actual fold change from both the MS and western data. It seems like a relatively small change. I am used to seeing corresponding ratio bar graphs with the western blot image showing the standard deviation and p-value.
3. The Methods and Materials need to be more specific and or edited.
 - a. Page 32 line 772 – please specify which protease and phosphatase inhibitors were used.
 - b. Page 32- 33 line 791 - 796 – I am not clear what is being done after the cell pellets were lysed using the AFC buffer in this section. Is the three freeze-thaw cycle a protein precipitation method? You state protein stock solutions were prepared at 1 mg/ml – how was that measured? Bradford?
 - c. Page 33 Section beginning on line 801 – was dynamic exclusion utilized? If so what were the parameters? What was the injection time on the Q Exactive set to for both the MS and MSMS scan event? What was the HCD fragmentation energy set to?

d. Page 33 line 80 – you alkylated the cysteines, yet I do not note that alkylated cysteine was a considered modification. What was the mass accuracy limits in the search? What database was used to do the search? Which version?

4. I note that the Proteomic data has not been uploaded to a data depository.

POINT-BY-POINT RESPONSE (in blue text)

Reviewer #1 (Remarks to the Author):

This is an interesting manuscript that investigates the role of ubiquitin ligase, UBR4/POE, in the control of circadian rhythmicity. The main finding is that the principal role of UBR4/POE is regulating the intracellular traffic of secreted proteins, specifically in the traffic out of the Golgi (and, possibly, also from the ER to the Golgi), and not regulating the stability of the core clock proteins (PER/TIM/CRY). In the context of the circadian clock, the consequences of disrupting UBR4/POE result from the incorrect trafficking of neuropeptides, AVP/VIP and PDF, in the SCN and the fly clock circuit, respectively. The defects are relatively subtle in mice, affecting the speed of re-entrainment. In flies, by contrast, knockdown of POE in clock neurons causes immediate arrhythmicity under conditions of constant darkness (DD) and also defective responses under light:dark entrainment (LD). When knockdown is restricted to the PDF neuropeptide-expressing neurons the behavioral defects mirror those of pdf null mutants. In both organisms lack (or knockdown) of UBR4/POE function causes abnormal accumulation of neuropeptides. Proteomic analyses followed by rescue experiments place the blame for this on defective Coronin 7 function, which is required for budding of Golgi-derived transport vesicles.

The manuscript is a tour de force, combining mouse and fly work, and behavioral, molecular, and proteomic analyses. The text is well written, the narrative makes sense, and the results are well documented and presented. I worked hard to find anything useful to comment. Below are the meager results of my efforts.

Minor comments

1- The results, especially the live imaging, clearly show that UBR4/POE plays an acute function in protein traffic. Yet, it appears that it may also play a role during development, as indicated by the fact that (line 252) "... these flies [*tim*>*Dcr2*; *poe*^{RNAi}] had fewer LN_vs than either *tim*>*Dcr2* or *poe*^{RNAi} controls." It would be interesting to demonstrate this "developmental" function directly by examining the consequences of expressing *poe* RNAi in these neurons only during development. I recognize that this would mean extra experiments; I leave it to the Editor to decide if they would be required.

As the reviewer points out, there is likely a developmental function of *poe*, as there are fewer LN_vs in the *tim*>*Dcr2*; *poe*^{RNAi} flies compared to controls. The role of *poe* in fly development is a very interesting question, however, one that we chose not to pursue in our first investigation of this gene. The primary goal of our current study is to leverage the clock system to hopefully gain new insights on the molecular and cellular functions of *Ubr4/poe*. This in part is the reason why we opted to carry out most of our *poe* analysis using the *Pdf* >

Dcr2; *poe*^{RNAi} flies, to avoid the confounding effects of altered development. Following consultation with the editor, we have decided that it would be better to reserve a full investigation on the potential developmental roles of *poe* for a future study.

2- The manuscript presents UBR4/POE as a protein needed for proper circadian function. Yet, its function is much more general because UBR4/POE is involved in protein trafficking along both the regulated and the constitutive secretory pathways. The relevance of this function to other processes is demonstrated by, for instance, the fact that UBR4 null mutant mice die as embryos, with defects in many organ systems. Given this role, I am not sure that directing the work towards a “clock readership” (at least in the title and the abstract) is such a good idea, since there are many more readers interested in protein trafficking than in clocks. So if I were in the authors’ shoes I would use the clock as a sensitive assay to detect defects in protein trafficking, rather than emphasize the role of UBR4/POE on clock function. But I am not in the author’s shoes, and they may choose to leave the emphasis intact, which is their prerogative.

We thank the reviewer for this very helpful comment. We agree that the goal of our study was to shed light on the cellular functions of UBR4/POE and the clock system has allowed us to accomplish this. In the revised manuscript, we have shifted the emphasis in the discussion from the circadian phenotype of *Ubr4* cKO and *poe* KD flies, to the more general function of UBR4/POE in vesicular trafficking. This is reflected in the order in which these topics are now discussed: the cellular function of UBR4/POE is discussed first, followed by its effects on circadian behavior and clock network, and finally our views on how its function in vesicular trafficking may be related to phenotypes that have been previously described in organisms where the orthologous gene is disrupted or perturbed.

Micro-comments

3- Could the authors comment on why ablation of *ubr4/poe* causes damping of PER2 oscillations in the murine SCN and severe damping and phase delay of dPER rhythms in the fly clock neurons? Is this due to changes in the stability of the core clock proteins (PER/TIM/CRY) or is it an indirect effect due to incorrect intracellular neuropeptide traffic?

This is a very interesting question. The damping of PER2/dPER oscillations in *Ubr4* cKO mice and *poe* KD flies, respectively, may be due to either reduced translation or increased protein degradation of PER. Note that, based on our qRT-PCR analyses, it is unlikely that transcription or mRNA stability is altered. The V5-PER2 data in mouse Neuro-2a cells (Supplementary Fig. 1b) would favor increased protein degradation as the mechanism. However, as the reviewer points out, it is also possible that the PER phenotype is related to perturbed neuropeptide trafficking. Our current data do not allow us to distinguish

between changes in translation, or effects on protein degradation that are either dependent or independent of neuropeptide trafficking. In the revised discussion, we have acknowledged these possible explanations:

“Further investigations are also needed to establish the mechanism by which Ubr4/poe deficiency triggers a decrease in PER protein abundance. Plausible explanations include reduced translation of Per transcripts and reduced stability of PER proteins. The latter may be due to indirect effects of improper neuropeptide trafficking, as PDF has previously been shown to enhance PER stability⁴³.”

4- Also, do the authors have an explanation for why the traffic speed from the ER to the Golgi is increased when UBR4 function is impaired?

At the moment, we have no explanation for why ER-to-Golgi trafficking is faster in the absence of UBR4. To our knowledge, there are no obvious candidate genes in our MS data that might explain this effect. This is one of the phenotypes that we hope to explore in much further mechanistic detail in future studies.

5- The Supplementary data clearly detail the sources of all reagents including stocks and antibodies. I would ask the authors to refer the reader to this file for information on these sources at the start of the relevant sub-sections of the Methods section.

We thank the reviewer for this suggestion. We have directed the reader to Supplementary Table 4 at relevant sub-sections of the Methods section.

6- It is not clear how the authors quantified the PER-IR (e.g., Fig. 5 and Supplementary Fig. 3) since the signal is annular. Please provide a few more details (did the ROI consider the entire cell or only a piece of its rim, etc).

We apologize for the lack of clarity in our description of the PER- and PDF-IR quantification. Given the annular signal of PDF, we drew the ROI such that it excluded the cell nucleus; thus, only the cytoplasmic region containing PDF immunoreactivity was quantified. For PER, we drew the ROI to encompass the region of PER immunoreactivity. The methods section has been revised as shown below:

“For quantification of PDF fluorescence intensity in fly brains, the entire PDF-immunoreactive cytoplasmic region of the cell, excluding the nucleus, was delineated as the ROI and quantified. For quantification of PER fluorescence intensity in fly brains, the entire PER-immunoreactive region of the cell was delineated as the ROI and quantified. When PER

intensity was severely reduced or absent, PDF staining was used to delineate individual clock cells."

7- Regarding Fig. 5 and Supplementary Fig. 3 I would recommend reducing the number of panels. I admit that the data are very beautiful, but showing the pattern of immunoreactivity over many timepoints and for all genotypes means that each image is very tiny. Since the "hard data" are in the associated graphs and histograms, the images are really only illustrative. Thus, I would recommend reducing the number of panels, so that each can be shown in a larger size (as done for other panels that show staining, e.g., Figs. 4, 6 and 7).

We agree with the reviewer that those figures are quite busy. For Figure 5, we have moved the panels corresponding to CT5 and CT20 to Supplementary Figure S3G and increased the size of the remaining panels. For old Supplementary Fig. 3 (current Supplementary Fig. 4), we have reduced the number of panels by omitting those that show PDF immunoreactivity only, as PDF is used primarily as a marker of LN_s and the more critical data are the PER immunofluorescence intensities. The remaining panels, PER and the merge of PER and PDF, have been enlarged.

Reviewer #2 (Remarks to the Author):

This manuscript by Hegazi et al. demonstrated that UBR4/POE is essential for trafficking of neuropeptides at SCN and exporting peptides from Golgi apparatus. They identified Coronin7 bound to Golgi complex membranes as being regulated by UBR4 for cargo export from the Golgi apparatus. Overall, this work elucidated a novel function of UBR4/POE in secretory trafficking and highlighted the potential for application in circadian rhythms-related diseases. However, the manuscript suffers from a lack of mechanistic details and is largely descriptive in nature. I feel the current manuscript is too premature for publication in Nature Communications.

Major points

1. It is claimed that UBR4 regulates the level of Coronin7(CRN7) on a translational level, as opposed to transcriptional or degradative in nature (361-364). However, the premise of this claim is descriptive since the authors fail to present convincing data and instead rely on observations that other differentially expressed proteins (DEPs), including eukaryotic transcription factors, are similarly affected by the absence of UBR4. Even if this were the case, the authors fail to show exactly why or how UBR4 deficiency causes such a

phenomenon.

The reviewer raises a valid point. In the original submission, we suggested the UBR4 regulates CRN7 on a translational level based on a process of elimination (i.e., transcription, mRNA stability, and protein stability were not affected) and the fact that several translational regulators were identified as being differentially expressed in *UBR4* KO HEK293T cells. In the revised manuscript, we provide direct evidence that translation of *CRN7* is reduced in these cells. Using the polysome profiling assay, a popular method to monitor the translation status of mRNAs (based on the association of mRNA with either one, a few, or many ribosomes), we found that *CRN7* transcripts from *UBR4* KO HEK293T cells sedimented with the lighter-polysome fractions in sucrose density gradient centrifugation compared to control cells (Supplementary Fig. 6e and 6f).

2. It is claimed that VIP and AVP trafficking was impaired in *UBR4*-deficient SCN. While it seems reasonable that *UBR4* is essential for trafficking of those neuropeptides, the authors did not address the possibility that *UBR4* may not directly affect VIP/AVP trafficking since the synchronization process of SCN is affected from various signals. Defects in synchronization process might be an indirect consequence of *UBR4*-mediated signaling, in which case the authors should have provided more mechanistic insight.

The reviewer raises an important point on whether the effects of *UBR4* on VIP/AVP trafficking are direct and cell-intrinsic/cell-autonomous, or whether they are an indirect consequence of the synchronization process that occurs between clock neurons. To show that the effects of *UBR4* on VIP trafficking in SCN neurons are direct and cell-autonomous, we examined VIP localization in primary cultures of dispersed SCN neurons where the connectivity that naturally exists within the SCN is absent. Two experiments were initially performed, one looking at endogenous VIP and another where VIP was overexpressed using an AAV construct. We also conducted a third experiment to assess the effects of *Crn7* overexpression. The excerpt below describes the results from these experiments:

*"To demonstrate that cargo proteins are retained in SCN neurons in a cell-autonomous fashion, we prepared dissociated SCN neuronal cultures from *Ubr4^{fl/fl}* neonates and virally transduced them with Cre recombinase to ablate *Ubr4* (Supplementary Fig. 7a). In Cre-transduced cultures, we could easily identify cells with endogenous VIP expression, as VIP was highly localized to the Golgi (Supplementary Fig. 7b). In contrast, we had difficulty finding VIP-expressing neurons in GFP-transduced control cultures, as the VIP signal was very weak and diffuse, presumably as a result of efficient trafficking and secretion (Supplementary Fig. 7b). In order to quantify VIP localization, we performed a dual transduction using adeno-associated viral (AAV1) constructs that expressed Cre recombinase and the murine *Vip* gene*

under the control of the human Synapsin (hSyn) promoter (Fig. 8c). Similar to endogenous VIP, ectopically expressed VIP was highly concentrated in the Golgi of Cre-transduced neurons relative to GFP-transduced controls (Fig. 8c and 8d). Importantly, in triple transduction experiments, ectopically expressed VIP was no longer localized in the Golgi of Cre-transduced neurons when murine CRN7 was overexpressed (Fig. 8e and 8f, Supplementary Fig. 7a)."

3. The authors claim that Coronin 7 (CRN7) mediates exporting cargo from the Golgi apparatus, and that UBR4 regulates the translational level of CRN7. The authors should confirm whether recovery from impaired clock synchrony and chronic jetlag in UBR4-deficient mice is actually due to the decreased levels of CRN7.

To determine whether CRN7 downregulation is causal to the chronic jetlag phenotype of *Ubr4* cKO mice, we stereotaxically injected AAV constructs expressing CRN7 into the bilateral SCN of these animals. Our results show that CRN7 overexpression in the SCN rescues the jetlag phenotype, promoting desynchronized behavior in 50% of the *Ubr4* cKO mice. The excerpt below summarizes our findings:

*"Lastly, to determine whether defects in CRN7-mediated protein trafficking in the SCN underlie the resistance of *Ubr4* cKO mice to jetlag, we examined the effects of CRN7 overexpression in the *ChrA*^{6/2} jetlag paradigm. *Ubr4* cKO mice received bilateral SCN injections of AAV1 constructs encoding CRN7-2A-mCherry under the control of the CMV promoter or a control AAV1-CMV-mCherry vector prior to activity monitoring. Post-mortem analysis revealed robust expression of mCherry in the majority of, but not all, SCN neurons (Supplementary Fig. 7c). All *Ubr4* cKO mice (9/9) injected with the mCherry control vector showed a single, entrained component of ~21 h (period: 21.00 ± 0.01h) under the *ChrA*^{6/2} schedule (Fig. 8g and 8h). In contrast, only 50% of *Ubr4* cKO mice (4/8) injected with the CRN7 vector were entrained to the *ChrA*^{6/2} schedule (period: 21.02 ± 0.02h), while the remaining 50% (4/8) exhibited desynchronized behavior with two rhythmic components (period: 20.96 ± 0.02h and 23.69 ± 0.15h) (Fig. 8g and 8h, Supplementary Fig. 7d)."*

4. Even though the authors claim a novel function of the N-recognin UBR4 in secretory trafficking for circadian clock synchrony, they do not show any experiments to confirm whether this novel function is indeed due to the role of UBR4 as an N-recognin. If this novel function is indeed attributable to the N-recognin properties of UBR4, the authors should have tested the role of the UBR box within UBR4 in circadian clock synchrony.

The reviewer raises an excellent point. To address the contribution of the N-recognin properties of UBR4 on the trafficking phenotype in *UBR4* KO cells, we overexpressed either wild-type (full-length) or mutant UBR4 in *UBR4* KO HEK293T cells and examined the localization of NPY-GFP. The two mutants that we tested either contained a partial deletion

of the UBR box (Del) or carried multiple Ala substitutions in place of the conserved Cys/His residues in the UBR box (Ala). The excerpt below summarizes our findings:

Lastly, we determined whether the effects of UBR4 on cargo export from the Golgi are dependent on the UBR-box domain, which is required for the recognition of N-degrons³². Overexpression of full-length (FL), wild-type UBR4 in UBR4 KO HEK293T cells rescued the Golgi export phenotype, as evident by the broad distribution of NPY-GFP throughout the cytoplasm and reduced localization in the Golgi relative to the rest of the cell (Fig. 6k and 6l, Supplementary Fig. 5i). Surprisingly, the phenotype was also rescued by overexpression of two mutant forms of UBR4 in which the function of the UBR-box was abrogated: one mutant contained a partial deletion of the UBR-box domain (Del) while the other carried multiple His/Cys→Ala mutations in the UBR-box (Ala) (Fig. 6k and 6l, Supplementary Fig. 5i).

Minor points

1. It is recommended to remove “N-recognin” in the title, as the paper in its current format does not show any evidence that this novel function of UBR4 is related to its N-recognin properties within the N-degron pathway.

We agree with the reviewer and have removed “N-recognin” from the title. The new title is “UBR4/POE facilitates secretory trafficking to maintain circadian clock synchrony”

2. The word, “ubr4 KO” in HEK293T cell, should be replaced with “UBR4 KO” because HEK293T cell is originated from Homo sapiens.

We apologize for the oversight and have used “UBR4 KO” throughout the manuscript when referring to the human gene.

3. Same as 2., “fmr1” and “Crn7” should be replaced with “FMR1” and “CRN7”.

In the revised manuscript, *Crn7* has been replaced with *CRN7* when referring to the human gene. We have chosen to omit the original FMR1 data from the revised manuscript, as they provided little additional mechanistic insight to our study.

4. In figure 6, to show generalization to SCN, the authors could show that neuropeptide Y or other secretory proteins were restricted to the Golgi apparatus in SCN of control and Ubr4 cKO mice.

We appreciate the reviewer’s suggestion. Approximately 10% of SCN neurons express endogenous VIP and ~20% express AVP. To rephrase the question, as we examined only VIP and AVP by immunohistochemistry, it leaves open the possibility that the Golgi export

phenotype is restricted solely to these two cell populations. However, we do not believe this to be the case. When we ectopically expressed VIP in primary SCN neuronal cultures using an AAV1 construct that is not biased in its infectivity towards a certain subtype of neuron, we consistently observed localization of VIP in the Golgi of *Ubr4*-deficient neurons (Fig. 8c and 8d). This result suggests that the Golgi export phenotype is generalized to all or most SCN neurons.

5. The authors mentioned that UBR4 deficiency would have consequences on timely communication between cells (307-309). To confirm that secretion of proteins is affected by depletion of UBR4 associated with exporting system from Golgi apparatus, the author can compare amounts of secreted proteins like NPY from control and UBR4 KO cells.

In the revised manuscript, to directly assay for secretion, we examined the amount of NPY-GFP protein that was secreted from *UBR4* KO and WT HEK293T cells into the culture media using ELISA. Our data (Supplementary Fig. 5j) show that secretion is reduced in the absence of UBR4.

Reviewer #3 (Remarks to the Author):

The authors have produced an exceptionally thorough study of the loss of function effects for the gene *UBR4* (*poe*) on the circadian timing system underlying rhythmic locomotion. They study the problem in not just one but two different model systems (mice and flies) and analyze the experiments at a multitude of scales, including whole animal behavior, cellular properties, ultrastructure and gene expression. The results appear robust and there is an interesting mix of negative and positive data presented, which increases confidence in the objectivity of the overall evaluation. My only criticism is with the elaboration of a very specific hypothesis that tries to unify all the observations in support of a single molecular mechanism – disruption of neuropeptide trafficking through the regulated secretory pathway via coronin7. The authors are of course entitled to interpret the data according to their judgement as long as the data are fairly presented – and I emphasize that the data are by and large fairly presented. Still the preponderant emphasis on a single mechanism, in the face of general cellular disruption with loss of *UBR4*, and of discordant results between flies and mice, argues in favor of less strident advocacy for the favored hypothesis. Here I offer a few specific comments and questions in hopes of improving this excellent manuscript.

We appreciate the reviewer's thoughtful comments. We have revised the discussion to provide a more balanced consideration of the potential mechanisms that could contribute

to the behaviors that we have observed in our *Ubr4* cKO mice and *poe* KD flies. A sample excerpt from our manuscript where we discuss alternative mechanisms is provided below:

“While the evidence aligns with the favored hypothesis that defective neuropeptide trafficking underlies the behavioral perturbations, we cannot exclude the possibility that other factors, or mechanisms, may also contribute to the circadian phenotype of Ubr4 cKO mice and poe KD flies. Given that both the regulated and constitutive secretory pathways are affected in UBR4 KO HEK293T cells, there may be other proteins with important functions in clock neurons whose trafficking is impacted by Ubr4/poe ablation. In addition to effects on protein trafficking, Ubr4/poe deficiency leads to a damping of PER oscillations, modest in the case of the murine SCN and severe in the s-LN_vs of Pdf>Dcr2; poe^{RNAi} flies. The extent to which damped PER oscillations contribute to the behavior of our animal models is unclear. Interestingly, selective disruption of per in PDF neurons, which approximates the effects of our Pdf>Dcr2; poe^{RNAi} flies in terms of per expression, does not lead to arrhythmicity in the majority of animals, unlike poe knockdown^{41,42}. This suggests that disrupted per expression in our flies is not the primary cause of the behavioral deficits, although a more rigorous examination of this hypothesis is warranted. Further investigations are also needed to establish the mechanism by which Ubr4/poe deficiency triggers a decrease in PER protein abundance. Plausible explanations include reduced translation of Per transcripts and reduced stability of PER proteins. The latter may be due to indirect effects of improper neuropeptide trafficking, as PDF has previously been shown to enhance PER stability⁴³. Lastly, it is important to consider not only the immediate, direct effects of a defect in neuropeptide trafficking on signaling and communication between cells, but also the indirect consequences of disrupted signaling on the connectivity and plasticity of clock networks. For example, Herrero et al. (2020) recently showed that PDF is necessary for the dynamic remodeling of axonal arbors of s-LN_vs across the circadian timescale⁴⁴. Their finding raises the possibility that there may be more far-reaching effects of poe ablation on the clock network than presently envisioned.”

L 371 “Furthermore, our *ubr4* cKO mice and *poe* KD flies exhibit perturbed behavioral rhythms that are best explained by a central pacemaker with weak network synchrony, a likely consequence of defective neuropeptide trafficking.” This is an interesting hypothesis, and the additional observation on a specific candidate mediator (*coronin7*) is also interesting, but I feel that a lot of complex cellular physiology is being simplified in pursuit of offering a concise (albeit plausible) mechanism. It is recognized by the authors that both regulated and constitutive pathways are down-regulated (line 306) and that other pathways (like ER to Golgi) are likewise affected. It is further shown that many genes are either up or down regulated in the genetic states studied. It also mentioned that flies with mutations in *poe* (aka pushover) gene display physiological phenotypes in motorneurons that are

consistent with disruptions in trafficking of ion channel components. In other words, many things are changing substantively in neurons lacking this protein.

We agree with the reviewer that the cellular physiology of *Ubr4*-deficient cells is complex. The discussion has been modified to reflect this complexity. However, we would like to draw the reviewer's attention to new data showing that overexpression of *Crn7* in the SCN of *Ubr4* cKO mice is able to restore/promote desynchronized behavior in a subset of animals under chronic jetlag, similar to wild-type mice (Fig. 8g and 8h). Therefore, while we agree that there may be other mechanisms that contribute to the perturbations in circadian behavior, the effects of *Ubr4* ablation on *Crn7* expression and vesicular trafficking are likely to play an important role in determining the phenotype.

Behaviorally, effects on mice and flies are substantial in both case but surprisingly different. In mice effects are seen at extreme environmental light conditions (LL), while in flies, they occur preponderantly in DD. The authors argue this may be due to differences in cell number between the two different animals, but that surprising point is simply invoked without substantiation. Furthermore, at least in the case of the fly, the *po*e behavioral phenotypes do not correspond precisely to what is seen with the absence of PDF, (contrary to what is mentioned on line 224) except in so far as both phenotypes display some delayed arrhythmicity. As far as I can tell, *po*e KD flies do not display shortened periods. In mice, *ubr4* cKO leads to more rapid adjustment to phase advances, although not phase delays. Importantly, it still takes the *ubr4* cKO mouse a few days to adjust. The authors analogize the rapid adjustment to two studies to argue that similar adjustment phenotypes are shown by deficient mice deficient in the neuropeptide receptors AVPR and VIPR. However, those studies indicate a different picture according to my reading. Lack of AVPR leads to more rapid adjustment to both phase advances and phase delays. Lack of VIPR leads to immediate adjustment of locomotor activity phases (unlike the *ubr4* cKO mice) indicative of little contribution by the internal clock and near complete control by masking effects of light.

The reviewer raises several excellent points. In the revised manuscript, we have removed the argument of the difference in cell number between the mouse and fly central clock as a way to explain the differences in behavioral phenotypes. The reviewer is correct in noting that *po*e KD flies do not display shortened periods and that they are similar to the *Pdf*-null flies in so far as they both exhibit delayed arrhythmicity. We have clarified this point in our manuscript:

"In the fly model, poe KD in PDF neurons leads to complex DD behaviors including delayed arrhythmicity in the majority of mutant flies, which partially overlaps with the phenotype of Pdf⁰¹ mutants ⁴."

Regarding our previous comparison between *Ubr4* cKO mice and mice deficient for AVPR and VIPR, we agree that our mice do not completely mirror the phenotypes of either model, even though there are some areas where they are similar. The reviewer also raises an excellent point about the masking effects in *Vipr*-deficient mice. In the revised manuscript, we have omitted the comparison of *Ubr4* cKO mice with *Vipr*-deficient animals, and we have clarified where our mouse model overlaps with the *Avpr*-deficient mice in terms of their behavioral phenotype. The relevant excerpt is provided below:

"The pattern of behavior elicited by Ubr4/poe deficiency also suggests that clock network synchrony, which relies heavily on neuropeptide-based communication, is compromised. For example, Ubr4 cKO mice are more susceptible to LL-induced arrhythmicity and are resistant to acute and chronic jetlag, features that are consistent with weakened intra-SCN coupling^{7,25}. Ubr4 cKO mice somewhat resemble V1a/V1b double knockouts, which lack AVP receptors, with respect to their rapid re-entrainment to an abrupt LD advance⁷. However, we do not expect Ubr4 cKO mice to perfectly mirror the phenotype of any knockout model where a particular neuropeptide (or its receptor) is absent, as Ubr4 deficiency is likely to reduce, but not eliminate, the secretion of a variety of neuropeptides."

Moreover, regarding neuropeptide trafficking changes: There is substantial IHC analysis showing a change in the balance of neuropeptide content favoring cell bodies and away from processes, in both flies and mice. That evidence is consistent with the favored hypothesis. However, the small LNV neurons display significantly increased neuropeptide content at the subjective night time point, which is anomalous and not consistent. Herraro et al (2020, Current Biology) recently reported that chronic depolarization of these same neurons increased neuropeptide staining levels in the same dorsal terminals at this time point – they argued that depolarization likely recruits increased PDF containing vesicles to the distal aspects. That hypothesis is diametrically opposite to the one favored in the MS under review.

The reviewer raises interesting points. Our PDF immunostaining clearly shows that there is more PDF in the dorsal terminals of *poe* KD flies relative to controls at CT14. While that may appear to be contradictory to our hypothesis that Golgi export is impaired in these animals, it is important to note that, in our trafficking experiments in *UBR4* KO HEK293T cells, Golgi exit was slower but it was not blocked. Hence, the PDF might leave the Golgi in the soma of s-LNVs at a slower rate, and it may accumulate at the dorsal terminals due to another deficit, possibly the fusion of secretory vesicles with the plasma membrane. This is discussed in the excerpt below:

"Although we elected to focus on the mechanism underlying the Golgi export phenotype, there is some evidence to suggest that UBR4 may affect other vesicular trafficking events. For example, ER-to-Golgi trafficking in HEK293T cells appears to be accelerated in the absence of

UBR4. Exocytosis might also be reduced or delayed, given the conspicuous accumulation of large, post-Golgi, GPI-GFP⁺ membrane-enclosed compartments in the cytoplasm of UBR4 KO HEK293T cells, and the abnormally high accumulation of PDF in the dorsal terminals of Pdf>Dcr2; poe^{RNAi} flies at CT14. Further investigations are warranted to fully characterize the effects of UBR4 along the entire biosynthetic secretory pathway."

We thank the reviewer for pointing us to the Herrero et al. study. Upon reading it, it gave us fresh ideas on what other possible effects a deficit in PDF trafficking might have on the fly clock. We have cited the Herrero study in the excerpt below:

"Lastly, it is important to consider not only the immediate, direct effects of a defect in neuropeptide trafficking on signaling and communication between cells, but also the indirect consequences of disrupted signaling on the connectivity and plasticity of clock networks. For example, Herrero et al. (2020) recently showed that PDF is necessary for the dynamic remodeling of axonal arbors of s-LN_vs across the circadian timescale⁴⁴. Their finding raises the possibility that there may be more far-reaching effects of poe ablation on the clock network than presently envisioned."

There are rather substantial effects on the molecular oscillator in both systems studied. The authors clearly document these effects but do not emphasize their potential contributions to disruptions of rhythmic locomotor outputs. Especially in the case of the fly, this seems unwarranted. There are also effects on PDF cell number, which implies effects on cell viability. There was no mention of counts for PER⁺ cells, which could be a useful measure and I believe the data is in hand from the careful systematic time of day analysis of PER staining. Likewise, it would be useful to employ a measure of cell morphology (e.g., cd8-GFP) distinct from neuropeptide levels to assess POE contributions to overall neuronal differentiation.

In the revised manuscript, we have examined the morphology of PDF neurons in *poe* KD flies using the CD2-HRP membrane marker (Supplementary Fig. 3f). We observed no obvious differences in axonal morphology of PDF neurons between *Pdf>Dcr2; poe^{RNAi}* flies and controls, indicating that, at least in these animals, POE does not affect neuronal differentiation. The data also allow us to rule out the possibility that altered PDF distribution in the *Pdf>Dcr2; poe^{RNAi}* flies is due to structural defects of the axonal projections.

With regards to PER cell counts, we received further clarification from the reviewer (via the editor) on this issue:

"My question concerns whether the cells are alive in the knockdown condition – is POE necessary for cell viability? I do not have the MS at hand – my remembrance is that they had performed PER antibody staining in a *tim*>POE RNAi background. That would be best to measure cell viability across the whole group (counting PER+ nuclei). But if that experiment was only done with *Pdf*>POE RNAi, then it will still be informative to count just PDF cell bodies. Because this is an endpoint assay (cell death), only one time point is necessary (meaning, if cells died, they likely died many days prior to dissection): so then the best single time point is at or near the max for PER protein. If PER levels are not predictable enough, then an anti-peptide antibody stain for PDF cells could be used to count that subset."

To clarify, we only imaged the region of the fly brain that included the PDF neurons in both the *tim*>*Dcr2*; *poe*^{RNAi} and *Pdf*>*Dcr2*; *poe*^{RNAi} lines. Therefore, it is not possible for us to quantify all PER⁺ nuclei in the fly brain using the previously acquired images. We did quantify PDF⁺ cell bodies and found that *tim*>*Dcr2*; *poe*^{RNAi} flies had fewer PDF cells compared to controls (Supplementary Fig. 4d and 4e), while *Pdf*>*Dcr2*; *poe*^{RNAi} flies had the same number (Supplementary Fig. 3b and 3c). As a consequence of these findings, we elected to characterize the PDF trafficking phenotype in the *Pdf*>*Dcr2*; *poe*^{RNAi} line, which did not show obvious developmental deficits.

All of this leads me to urge more moderation in the enthusiasm with which the authors promote their favored hypothetical mechanism centered on UBR4>coronin7>Neuropeptides.

We sincerely thank the reviewer for their thoughtful comments. We hope that by moderating our advocacy of our favored hypothesis and by providing a more balanced discussion that we have strengthened our manuscript.

Small points

In the images of the fly brain, panel A displaying PDF IHC and *poe* in situ signals at very low magnification conveys little useful information. In panel B, in situ signals in control and RNAi knockdown conditions indicate a difference but the background levels remain substantial so as to diminish appreciation of "loss" of *poe* RNA signals.

Panel A showing *Pdf* and *poe* ISH signals in the whole fly brain has now been moved to Supplementary Fig. 3a. The intent of this panel is to illustrate the ubiquitous expression of *poe* in the fly brain. With respect to panel B (currently Fig. 4a), we appreciate that the loss of *poe* ISH signal appears to be modest when considering the entire micrograph. However, as the knockdown of *poe* is only in PDF neurons, it is expected that the signal is not lost elsewhere. To assist the reader in observing the loss of *poe* signal in PDF neurons, we have demarcated the boundary of a PDF neuron in each of the micrographs.

Line 375 mentions “unpublished” data: I believe that is no longer a supported mechanism.

We apologize for this and have removed mention of unpublished data from the manuscript.

Reviewer #4 (Remarks to the Author):

General Comments

My review for the manuscript NCOMMS-20-34785 “The N-Recognin UBR4/POE facilitates secretory trafficking to maintain circadian clock synchrony” is focused on the mass spectrometry/proteomics contribution to the manuscript. The mass spectrometry data presented is suitable, however more details in the method needs to be included and I have noted some curiosities and weaknesses in the results.

Specific Comments

1. In the results from the label-free quantitative MS where you compare *ubr4* KO vs WT HEK293T cells I am curious why out of 471 differentially expressed proteins only one protein, Coronin 7, was of interest.

We appreciate the reviewer’s comment. Even though we identified several phenotypes at the behavioral and cellular level that are stemming from the disruption of UBR4/POE, we elected to focus on investigating the molecular mechanisms by which UBR4 promotes Golgi exit of secretory proteins. Following careful examination of the 471 DEPs, Coronin 7 emerged as our top-ranked candidate, given its well-studied role in Golgi export. Several studies have shown that mammalian CRN7 is a cytosolic protein that is recruited to the cytoplasmic side of Golgi membranes where it promotes normal Golgi organization and biogenesis of TGN-derived transport carriers required for Golgi export (Bhattacharya et al., 2016⁴⁰; Rybakin, 2008³⁵; Rybakin et al., 2004, 2006, 2008^{33,34,39}; Yuan et al., 2014³⁶). Therefore, the downregulation of CRN7 in *UBR4* KO HEK293T cells seemed to us to be a plausible explanation for the impaired Golgi exit phenotype. Our pursuit of this hypothesis proved to be fruitful, as overexpression of CRN7 not only rescued the Golgi exit phenotype in *UBR4* KO HEK293T cells, but it also rescued the behavioral phenotype of *Ubr4* cKO mice under chronic jetlag.

2. The presented results for this protein are very vague. The manuscript only states that the CRN7 is downregulated and the western blot supports the MS results. While I agree the western does show slightly less abundant bands, however, what are the p-values for both the MS and western data for this protein and what is the actual fold change from both the MS and western data. It seems like a relatively small change. I am used to seeing

corresponding ratio bar graphs with the western blot image showing the standard deviation and p-value.

We apologize for the oversight. In the revised manuscript, we have included bar graphs showing the levels of CRN7 obtained from MS as well as Western blot analyses, with corresponding p-values provided (**Fig. 7b, 7d, 7e**). In addition, details regarding the fold-change (FC) are now incorporated in the relevant results section:

*“Consistent with our quantitative MS data, Western blot analyses revealed a significant downregulation of CRN7 in UBR4 KO HEK293T cells and in SCN tissues extracted from Ubr4 cKO mice (MS: $\text{Log}_2(\text{fold-change (FC)})$ of CRN7 = -0.9; HEK293T WB: CRN7 FC = 0.62; SCN WB: CRN7 FC = 0.39) (**Fig. 7b, 7d and 7e**).”*

3. The Methods and Materials need to be more specific and or edited.

a. Page 32 line 772 – please specify which protease and phosphatase inhibitors were used.

We apologize for the lack of detail. The protease inhibitor cocktail (catalog number) and phosphatase inhibitors used are now specified in the Methods section:

“All subsequent processing steps were performed at 4 °C unless otherwise stated. AFC buffer (10 mM Tris-HCl, pH 7.9, 420 mM NaCl, 0.1% NP-40) supplemented with protease inhibitors (Sigma, S8830-20TAB) and phosphatase inhibitors (10mM NaF, 1mM sodium orthovanadate, 2mM sodium pyrophosphate) was added to frozen cell pellets.”

b. Page 32- 33 line 791 - 796 – I am not clear what is being done after the cell pellets were lysed using the AFC buffer in this section. Is the three freeze-thaw cycle a protein precipitation method? You state protein stock solutions were prepared at 1 mg/ml – how was that measured? Bradford?

We have now added more details on the steps taken following the addition of AFC buffer to the cell pellets. The relevant part of the Methods section is shown below:

“To lyse the cells, samples were subjected to three freeze-thaw cycles by transferring the samples between ethanol/dry ice and a 37 °C water bath, with frequent mixing to prevent sample temperature from rising above 4 °C. Lysates were then sonicated (20x, 0.3s on, 0.7s off) and centrifuged at 13,000 rpm for 30 min at 4 °C. Following protein quantification using a NanoDrop spectrophotometer (absorbance measured at 280nm), protein samples were prepared at 1 mg/ml stocks, precipitated using the ProteoExtract kit (MilliporeSigma), dried, and subsequently subjected to a trypsin digestion protocol. Specifically, dried samples were reconstituted in 50 mM ammonium bicarbonate, reduced using 2 mM TCEP-HCl, alkylated

with 11 mM iodoacetamide, and digested with 5 µg trypsin O/N at 37 °C. Peptides were acidified with acetic acid, desalted using C-18 ZipTips, and dried prior to LC-MS analysis."

c. Page 33 Section beginning on line 801 – was dynamic exclusion utilized? If so what were the parameters? What was the injection time on the Q Exactive set to for both the MS and MSMS scan event? What was the HCD fragmentation energy set to?

Again, we apologize for the omission of key details. The additional details requested have now been added to the revised manuscript, as demonstrated by the excerpt below from the Methods section:

"The EasySpray ion source (ThermoFisher Scientific) was used to directly ionize peptides injected into a Q Exactive HF mass spectrometer (ThermoFisher Scientific). For each selected MS1 full scan mass spectrum in profile mode, 20 MS2 data-dependent scans were acquired with HCD fragmentation at 32% normalized collision energy. For MS, the maximum injection time was 70 ms, and for MS/MS it was 25 ms. The dynamic exclusion range was set to 15 s. The full MS scan ranged from 300 – 1650 m/z and was followed by data-dependent MS/MS scan of the 20 most intense ions. The resolutions of the full MS and MS/MS spectra were 60,000 and 15,000, respectively. Data-dependent mode was used for MS data acquisition with target values of 3E+06 and 1E+05 for MS and MS/MS scans, respectively."

d. Page 33 line 80 – you alkylated the cysteines, yet I do not note that alkylated cysteine was a considered modification. What was the mass accuracy limits in the search? What database was used to do the search? Which version?

Alkylated cysteine (carbamidomethylation of cysteine residues) was considered a fixed modification; we apologize for omitting this detail from the initial version of the manuscript. Raw MS files were processed using MaxQuant (version 1.6.6.0). Database search was performed with the built-in Andromeda search engine (version 1.6.6.0) using the decoy human Uniprot database (release date 2011_07). The search mass tolerance was set at 20 ppm for the first search, and 4.5 ppm for the main search. The following excerpt from the methods section addresses this comment:

"Raw MS files were processed using MaxQuant (version 1.6.6.0). Database search was performed with the built-in Andromeda search engine (version 1.6.6.0) using the decoy human Uniprot database (release date 2011_07, decoy mode set to 'revert'). The following parameters were used: methionine oxidation (M) and protein N-terminal acetylation were set as variable modifications, and carbamidomethylation of cysteine residues set as a fixed modification. Enzyme specificity was set to trypsin/P, with a maximum of 2 missing cleavages allowed. The "match between runs" option was enabled in order to transfer identification between different

LC-MS runs based on the peptides' mass and retention time following retention time alignment. The search mass tolerance was set at 20 ppm for the first search, and 4.5 ppm for the main search. A false discovery rate (FDR) of 1% was used to filter the data at the peptide and protein levels and a minimum length of seven amino acids was used for peptide identification."

4. I note that the Proteomic data has not been uploaded to a data depository.

We are very sorry to hear that you were unable to access the raw data. The data were deposited prior to the original submission, and we have checked the functionality of the reviewer login information. The data can be accessed using the information given below:

The mass spectrometry proteomics data have been deposited to the ProteomeXchange Consortium (<http://proteomecentral.proteomexchange.org>) via the PRIDE partner repository with the dataset identifier PXD020630, and can be accessed by the reviewer using the following details:

Link: <http://www.ebi.ac.uk/pride>

Username: reviewer16017@ebi.ac.uk

Password: 20L88wlQ

We have, on rare occasion, not been able to access our own data using the login information above for a few hours, up to a day. We received confirmation from the PRIDE repository team that this is due to a temporary issue with the website itself.

REVIEWER COMMENTS

Reviewer #1 (Remarks to the Author):

The authors have responded satisfactorily to my comments. I have no additional comments. I think it's a work of interest to cell biologists in general, not only to circadian biologist.

Reviewer #2 (Remarks to the Author):

The authors have addressed all of my concerns. This manuscript now can be accepted with some minor revisions.

1. It is better to mark 'ns (not significant)' in Figure1d-g to show similar results between groups more clearly.
2. Using a protein ladder designed for high-weight proteins (e.g. LC5699, Thermo Fisher) would be better to indicate the size of UBR4 in western blot panels.
3. At Supplementary Figure 6b, it is recommended to use GAPDH instead of mouse gapdh primer as a normalization control for it was conducted in human cells.

Reviewer #3 (Remarks to the Author):

As before, I commend the authors on attacking this experimental problem with so much energy and for bringing in so many experimental methods and models. The scale of their consideration ranges from ultrastructure to behavior and the authors are not at all hesitant to pursue further (better) understanding and to test their model by whatever means necessary. Further the experiments appear technically sound (at least in the case of those that I know something about – which is not all). Finally the experimental detail to support their contention that Ubr4 regulates post-Golgi trafficking is impressive. My only remaining concern is my original one - I don't think the model (that secretory peptide signaling is the primary broken element) is well-chosen. I detail this conclusion with criticisms about the interpretations of behavioral and molecular experiments.

Major concerns

#1. The interpretation of behavioral phenotypes

Ubr4 in the Mouse.

“Ubr4 cKO mice were similar to controls with respect to period, amplitude, and activity levels under LD and DD. “

This description does not support a conclusion that Ubr4 plays an important role in regulating normal locomotor rhythmic behavior or normal secretory peptide signaling in the SCN.

In the case of the mouse, the authors again claim ‘partial overlap’ of phenotypes, Ubr4 loss of function with V1a/V2a double knockouts (“they somewhat resemble”) but I do not find this persuasive. Previously, I offered specific and substantive differences in the phenotypes of the pertinent mouse mutants, which the authors now acknowledge. Still they remain in support of their original hypothesis based on these interpretations. To quote:

“However, we do not expect Ubr4 cKO mice to perfectly mirror the phenotype of any knockout model where a particular neuropeptide (or its receptor) is absent, as Ubr4 deficiency is likely to reduce, but not eliminate, the secretion of a variety of neuropeptides.”

I agree - I suspect the UBR4 behavioral phenotype does not mirror any single secretory peptide mutation. But (to re-state a previous opinion) that is mainly because Ubr4 affects so many diverse cellular processes, beyond the secretion of neuropeptides.

Finding no disturbance of normal locomotor rhythm in Ubr4 mice either in LD or in DD, the authors tested behavior in LL and in response to phase shifts, and also employed a ‘chronic jetlag’ experimental design. They report that Ubr4 mice display a LL phenotype (less sensitive to light and disruption of adaptation to schedule shifts. Further they exhibit de-synchronization and that it is reversed (rescued) with CRN7 over-expression. The authors state:

491 The most direct evidence in support of our hypothesis is the re-emergence of desynchronized behavior in Ubr4 cKO mice under chronic jetlag following overexpression of CRN7.

These are interesting observations; I have one reservation about the desynchronization experiments:

Their interpretation invokes impaired communication between oscillators based on discussion from authors in the original 2012 CJL report (thus supporting this manuscript's model of poor neuropeptide signaling). However, and as explained in the original CJL report, another explanation for desynchronization under these extremely un-natural conditions is that the range of entrainment for one oscillator is compromised. Poe changes PER amplitude severely in the fly and Ubr does do mildly in the mouse. An effect on the oscillator should be considered and may therefore contribute to changes in range of entrainment and the observed phenotype. The authors do not discuss this possibility.

I also expected to see results from similar CRN7 overexpression on the very clear effects of Ubr4 cKO in LL and in response to phase shifts. Rescue of all these phenotypes would more strongly support their principle hypothesis.

Poe in *Drosophila*.

“In the fly model, *poe* KD in PDF neurons leads to complex DD behaviors including delayed arrhythmicity in the majority of mutant flies, which partially overlaps with the phenotype of Pdf01 mutants 4.”

Line 226 Pdf>Dcr2; *poe* RNAi flies were indistinguishable from controls in terms of their ability to anticipate the onset of morning and evening under LD conditions (Supplementary Fig. 2e-2h).

By my reading, using the phrase “partial overlap” implies a significant mechanistic overlap between *poe* and *pdf* mutant states because both produce some arrhythmicity. I submit that there is nothing especially unifying about phenotypes affecting the % of locomotor rhythmicity in DD: in this age of RNAi screening, there are numerous reports of *Drosophila* genes that when mutated or diminished increase behavioral arrhythmicity. Thus I see no justification to infer any mechanistic linkage between *poe* and *pdf* based on that non-specific phenotypic category. The behavior of *poe* RNAi flies in LD further confirms the lack of correspondence. I still do not see a basis to ascribe the behavioral phenotypes to disrupted secretory peptide signaling. Yes PDF staining is shifted within the neurons, but many things are also disrupted, including the molecular clock. Perhaps the phenotype results from the aggregate of effects?

Given the fact that Ubr4 helps the mouse clock system sense light, I was expecting to see experiments with poe RNAi testing the ability of the fly to sense LL and to respond to phase shifts. There may be a mechanistic parallel between the Ubr4/poe systems in the contributions to circadian physiology, albeit not involving neuropeptide secretion.

#2. Molecular model: Ubr4 locomotor behavior by regulating post-Golgi flow via CRN7

Reviewer 4 originally asked - why focus on CRN7 out of 471 differentially expressed genes? The authors responded by saying only that the choice was “fruitful” – they chose 1 in 471 and saw no reason to explain the premise or the logic. I am not convinced by their model that UBR4 works through CRN7 to significantly affect rhythmic locomotor output. The actual effects of poe/Ubr4 appear diverse and I strongly suspect some or many of the other 470 genes have roles to play, along with CRN7. The authors include words to this effect in their discussion, but retain nevertheless a singular emphasis on their primary hypothesis - the role of CRN7 and post-Golgi trafficking of secretory peptides. A narrow focus on CRN7 will I think limit future understanding of poe/Ubr4 biology.

Ubr4 mutants have lowered amounts of CRN7 yet CRN transcription and protein levels appear normal. To their credit, the authors next tested potential regulation of CRN7 translation and claim to find evidence for such. The data on this point then assumes importance as providing the mechanism to explain the authors’ principle hypothesis. Regarding the design of and actual data from this experiment, I have the following three concerns:

a. I found no statistical analysis of the results.

b. Further, if GAPDH and CRN7 are statistically different (which is at least implied), what should we understand from that? What is the actual consequence for translation rates if RNA is enriched in one density domain versus another? Knowing little about this subject I was hoping for guidance, yet authors offer only the following:

468 “Future experiments are needed to confirm the effects of UBR4 ablation on the translation of CRN7 and to establish the precise molecular mechanisms.”

c. Finally, GAPDH levels (the control RNA) appear the same across the different polysome fractions in either genotype, while UBR4 levels appear different across polysome fractions according to genotype – what is the significance of that? I would have thought the control RNA should also display different proportions across different polysome fractions in order to serve as a proper point of comparison. Perhaps the control RNA is not well chosen?

Reviewer #4 (Remarks to the Author):

Thank you for addressing my questions and concerns. I am satisfied with the changes.

Point-by-Point Response to the Reviewers

(response in blue)

Reviewer #2 (Remarks to the Author):

The authors have addressed all of my concerns. This manuscript now can be accepted with some minor revisions.

1. It is better to mark 'ns (not significant)' in Figure1d-g to show similar results between groups more clearly.

The changes have been made in the revised version of the manuscript as recommended by the reviewer.

2. Using a protein ladder designed for high-weight proteins (e.g. LC5699, Thermo Fisher) would be better to indicate the size of UBR4 in western blot panels.

We appreciate the reviewer's recommendation. The ladder suggested by the reviewer covers a wider range of protein sizes (30-460 kDa) than the one we had used in our experiments. However, it is still not ideal, since its maximum limit is below the size of UBR4 (~600 kDa).

3. At Supplementary Figure 6b, it is recommended to use GAPDH instead of mouse gapdh primer as a normalization control for it was conducted in human cells.

The legend for Supplementary Figure 6b has been adjusted to reflect the fact that human *GAPDH* primers were used (note: the use of lowercase letters for the human gene was an oversight on our part that has been corrected in the revised version).

Reviewer #3 (Remarks to the Author):

As before, I commend the authors on attacking this experimental problem with so much energy and for bringing in so many experimental methods and models. The scale of their consideration ranges from ultrastructure to behavior and the authors are not at all hesitant to pursue further (better) understanding and to test their model by whatever means necessary. Further the experiments appear technically sound (at least in the case of those that I know something about – which is not all). Finally the experimental detail to support their contention that Ubr4 regulates post-Golgi trafficking is impressive. My only remaining concern is my original one - I don't think the model (that secretory peptide signaling is the primary broken element) is well-chosen. I detail this conclusion with criticisms about the interpretations of behavioral and molecular experiments.

Major concerns

#1. The interpretation of behavioral phenotypes

Ubr4 in the Mouse.

“Ubr4 cKO mice were similar to controls with respect to period, amplitude, and activity levels under LD and DD. “

This description does not support a conclusion that Ubr4 plays an important role in regulating normal locomotor rhythmic behavior or normal secretory peptide signaling in the SCN.

We appreciate the reviewer’s comment. In the manuscript, it was not our intention to imply that UBR4 is required for “normal” locomotor rhythmic behavior under ecologically relevant light conditions, nor do we expect (or believe) that the changes to neuropeptide trafficking/signaling that are evident in these mice will impact all aspects of circadian behavior. Please bear in mind that *Ubr4* ablation reduces, but does not eliminate, neuropeptide secretion from cells. In a recent study, we showed that *Avp* and *Vip* hypomorphic mice that expressed the corresponding neuropeptides at 50% of WT levels did not have any defects in LD or DD behavior (Cheng et al., 2019; PMID: 31452438). In the revised discussion, we have added the following sentences:

“...the extent to which reduced vesicular transport in central clock neurons impacts the circadian behavior of Ubr4 cKO mice and poe KD flies has not been directly examined, except in the case of the chronic jetlag paradigm. While we believe that the defect in the trafficking and secretion of neuropeptides contributes to at least some of the behavioral phenotypes of our animal models, we by no means claim that it is the sole determinant or that all phenotypes (or lack thereof) are equally impacted by it.”

In the case of the mouse, the authors again claim ‘partial overlap’ of phenotypes, Ubr4 loss of function with V1a/V2a double knockouts (“they somewhat resemble”) but I do not find this persuasive. Previously, I offered specific and substantive differences in the phenotypes of the pertinent mouse mutants, which the authors now acknowledge. Still they remain in support of their original hypothesis based on these interpretations. To quote:

“However, we do not expect Ubr4 cKO mice to perfectly mirror the phenotype of any knockout model where a particular neuropeptide (or its receptor) is absent, as Ubr4 deficiency is likely to reduce, but not eliminate, the secretion of a variety of neuropeptides.”

I agree - I suspect the UBR4 behavioral phenotype does not mirror any single secretory peptide mutation. But (to re-state a previous opinion) that is mainly because Ubr4 affects so many diverse cellular processes, beyond the secretion of neuropeptides.

The reviewer raises valid points: other mechanisms may also contribute to the *Ubr4* cKO phenotype. Furthermore, AVP signaling is unlikely to be completely abrogated in *Ubr4* cKO mice, making it challenging to compare their phenotypes with those of neuropeptide- or neuropeptide receptor-deficient mouse strains. We do not expect these phenotypic comparisons to produce an exact match.

Given these points, we have removed the sentences referring to the *V1a/V2a* double knockouts from the discussion. We have also included a new paragraph on what alternative mechanisms may be operating in our animal models:

“Further investigations are needed to decipher the causal mechanisms underlying the individual circadian phenotypes of Ubr4 cKO and poe KD flies. For example, it would be worthwhile to determine if overexpression of Crn7 is sufficient to rescue the acute jetlag or LL phenotypes of Ubr4 cKO mice, and if the fly ortholog, POD-1, plays a similar role in the s-LNs to regulate PDF trafficking and circadian behavior. Although our current study focused on the effects of Ubr4 ablation on the transport of neuropeptides, it is clear that both the regulated and constitutive secretory pathways are perturbed in UBR4 KO HEK293T cells. Thus, there may be other proteins whose trafficking is also impaired in Ubr4/poe-deficient clock neurons and which contribute to the behavioral phenotype. Beyond the role of UBR4 in protein trafficking, it is important to determine the contributions of damped PER oscillations to the behavior of Ubr4 cKO mice and poe KD flies, and to identify the molecular underpinnings for the decrease in PER protein abundance in these animal models. Plausible mechanisms include reduced translation of Per transcripts and reduced stability of PER proteins. If UBR4 indeed regulates PER stability, it would be important to ask whether this is a direct effect that relies on the N-recognin activity of UBR4, or alternatively an indirect effect stemming from improper neuropeptide trafficking⁴⁵. On the other hand, if UBR4 regulates Per translation, as it appears to do for Crn7, a full examination of the potential links between UBR4 and the translational machinery, including its regulators, is warranted. Finally, a deeper exploration of the 471 DEPs in UBR4 KO cells may reveal additional factors that mediate the effects of UBR4 on vesicular trafficking or other cellular processes.”

Finding no disturbance of normal locomotor rhythm in *Ubr4* mice either in LD or in DD, the authors tested behavior in LL and in response to phase shifts, and also employed a ‘chronic jetlag’ experimental design. They report that *Ubr4* mice display a LL phenotype (less sensitive to light and disruption of adaptation to schedule shifts. Further they exhibit de-synchronization and that it is reversed (rescued) with CRN7 over-expression. The authors state:

491 The most direct evidence in support of our hypothesis is the re-emergence of desynchronized behavior in *Ubr4* cKO mice under chronic jetlag following overexpression of CRN7.

These are interesting observations; I have one reservation about the desynchronization experiments: Their interpretation invokes impaired communication between oscillators based on discussion from authors in the original 2012 CJL report (thus supporting this manuscript's model of poor neuropeptide signaling). However, and as explained in the original CJL report, another explanation for desynchronization under these extremely un-natural conditions is that the range of entrainment for one oscillator is compromised. *Poe* changes PER amplitude severely in the fly and *Ubr* does do mildly in the mouse. An effect on the oscillator should be considered and may therefore contribute to changes in range of entrainment and the observed phenotype. The authors do not discuss this possibility.

As the reviewer insightfully points out, the chronic jetlag phenotype of *Ubr4* cKO mice could also be explained by changes in the range of entrainment. Oscillator properties (period, amplitude) can affect

the range of entrainment, as can neuropeptide signaling (i.e., coupling interactions), as demonstrated in two computational studies by Ananthasubramaniam et al., 2014 (PMID: 24743470) and Abraham et al., 2010 (PMID: 21119632). We have removed the paragraph discussing the chronic jetlag phenotype (and the other behaviors) and replaced it with the following:

“The pattern of behavior elicited by Ubr4/poe deficiency suggests that clock network synchrony is perturbed. However, the extent to which reduced vesicular transport in central clock neurons impacts the circadian behavior of Ubr4 cKO mice and poe KD flies has not been directly examined, except in the case of the chronic jetlag paradigm. While we believe that the defect in the trafficking and secretion of neuropeptides contributes to at least some of the behavioral phenotypes of our animal models, we by no means claim that it is the sole determinant or that all phenotypes (or lack thereof) are equally impacted by it. Under chronic jetlag, Ubr4 cKO mice entrain efficiently to the Zeitgeber cycle and only upon overexpression of Crn7 in the SCN will a subset of these animals desynchronize. These results, along with our Crn7 rescue experiments in SCN neuronal cultures, suggest that defective neuropeptide trafficking in Ubr4 cKO mice is responsible for their inability to desynchronize under extreme conditions of chronic jetlag. If we consider forced desynchrony under the ChrA^{6/2} paradigm as desynchronization of a dual oscillator system, the resistance of Ubr4 cKO mice to desynchrony suggests that Ubr4 ablation affects the range of entrainment in such a way that the Zeitgeber period is now within the range of entrainment of both oscillators. Entrainment range depends on various factors including the amplitude and period of the oscillator as well as neuropeptide-mediated coupling of oscillators^{41,42}. Given that the amplitude of PER2 rhythms is mildly reduced in the SCN of Ubr4 cKO mice, we cannot exclude the possibility that the chronic jetlag phenotype arises from the combined effects of altered oscillator properties and impaired communication between oscillators.”

I also expected to see results from similar CRN7 overexpression on the very clear effects of Ubr4 cKO in LL and in response to phase shifts. Rescue of all these phenotypes would more strongly support their principle hypothesis.

We agree that it would be wonderful to perform rescue experiments across all our behavioral paradigms. However, based on the editor’s decision letter, we believe that they are beyond the scope of the present study. In our paragraph on future directions, we mention the importance of attempting to rescue the acute jetlag and LL phenotypes with *Crn7* overexpression (see above).

Poe in *Drosophila*.

“In the fly model, poe KD in PDF neurons leads to complex DD behaviors including delayed arrhythmicity in the majority of mutant flies, which partially overlaps with the phenotype of Pdf01 mutants 4.”

Line 226 Pdf>Dcr2; poe RNAi flies were indistinguishable from controls in terms of their ability to anticipate the onset of morning and evening under LD conditions (Supplementary Fig. 2e-2h).

By my reading, using the phrase “partial overlap” implies a significant mechanistic overlap between *poe* and *pdf* mutant states because both produce some arrhythmicity. I submit that there is nothing especially unifying about phenotypes affecting the % of locomotor rhythmicity in DD: in this age of RNAi screening, there are numerous reports of *Drosophila* genes that when mutated or diminished increase behavioral arrhythmicity. Thus I see no justification to infer any mechanistic linkage between *poe* and *pdf* based on that non-specific phenotypic category. The behavior of *poe* RNAi flies in LD further confirms the lack of correspondence. I still do not see a basis to ascribe the behavioral phenotypes to disrupted secretory peptide signaling. Yes PDF staining is shifted within the neurons, but many things are also disrupted, including the molecular clock. Perhaps the phenotype results from the aggregate of effects?

We appreciate the reviewer’s comment. Yes, in the absence of rescue experiments, we cannot say with certainty that impaired PDF signaling is the underlying mechanism for the *poe* KD phenotype, even though both *Pdf>Dcr2; poe^{RNAi}* flies and *Pdf⁰¹* mutants show the same peculiar delayed arrhythmicity (and not the more commonly observed, immediate arrhythmicity under DD). We agree with the reviewer that the *poe* KD phenotype is likely the result of the aggregate effects of multiple deficits, including the drastic changes in PER oscillations. The previous discussion of the *poe* KD phenotype has been removed and replaced with the following paragraph:

*“Our fly model reaffirms the effects of Ubr4/poe ablation on neuropeptide trafficking, but also provides strong evidence for its effects on the molecular clock. Unlike Ubr4 cKO mice, Pdf>Dcr2; poe^{RNAi} flies exhibit profound changes in the amplitude and phase of PER oscillations in the s-LN_vs. The majority of Pdf>Dcr2; poe^{RNAi} flies display delayed-arrhythmic behavior under DD, emerging after a brief period of rhythmicity: this phenotype has also been observed in Pdf⁰¹ mutants⁴. While it may be tempting to infer that impaired PDF signaling is the cause of the delayed arrhythmicity of Pdf>Dcr2; poe^{RNAi} flies, it is possible that this behavior reflects the aggregate effects of damped molecular oscillations and less PDF signaling. Interestingly, selective disruption of *per* in PDF neurons does not lead to arrhythmicity in the majority of flies, suggesting that the changes in PER oscillations in Pdf>Dcr2; poe^{RNAi} flies, on their own, might not be sufficient to trigger delayed arrhythmic behavior under DD^{43,44}. It is also important to point out that not all phenotypes exhibited by Pdf⁰¹ mutants, namely, lack of morning anticipation and advanced evening anticipation under LD as well as short DD period, are observed in Pdf>Dcr2; poe^{RNAi} flies. The fact that Pdf>Dcr2; poe^{RNAi} and Pdf⁰¹ flies do not perfectly mirror each other with respect to their behavior is not surprising, given that 1) *poe* ablation is unlikely to result in the complete loss of PDF signaling, and 2) other mechanisms besides aberrant PDF signaling may also help to shape the behavior of Pdf>Dcr2; poe^{RNAi} flies under different environmental light conditions.”*

Given the fact that Ubr4 helps the mouse clock system sense light, I was expecting to see experiments with *poe* RNAi testing the ability of the fly to sense LL and to respond to phase shifts. There may be a mechanistic parallel between the Ubr4/*poe* systems in the contributions to circadian physiology, albeit not involving neuropeptide secretion.

These are interesting experiments indeed, but we believe that they are beyond the scope of the present study, based on the editor’s decision letter.

#2. Molecular model: Ubr4 locomotor behavior by regulating post-Golgi flow via CRN7

Reviewer 4 originally asked - why focus on CRN7 out of 471 differentially expressed genes? The authors responded by saying only that the choice was “fruitful” – they chose 1 in 471 and saw no reason to explain the premise or the logic. I am not convinced by their model that UBR4 works through CRN7 to significantly affect rhythmic locomotor output. The actual effects of *poe/Ubr4* appear diverse and I strongly suspect some or many of the other 470 genes have roles to play, along with CRN7. The authors include words to this effect in their discussion, but retain nevertheless a singular emphasis on their primary hypothesis - the role of CRN7 and post-Golgi trafficking of secretory peptides. A narrow focus on CRN7 will I think limit future understanding of *poe/Ubr4* biology.

We agree with the reviewer that our proteomics data may provide even greater insights into UBR4 biology than currently appreciated, in that there may be other DEPs that also contribute to the *UBR4* KO phenotype. Hence, we have added the following sentence to the discussion to point the reader in this direction as they consider future investigations: *“Finally, a deeper exploration of the 471 DEPs in UBR4 KO cells may reveal additional factors that mediate the effects of UBR4 on vesicular trafficking or other cellular processes.”*

In terms of the premise or logic that led us to focus on CRN7 in the current manuscript, it is explained below in our original response to Reviewer #4:

Even though we identified several phenotypes at the behavioral and cellular level that are stemming from the disruption of UBR4/POE, we elected to specifically focus on investigating the molecular mechanisms by which UBR4 promotes Golgi exit of secretory proteins. Following careful examination of the 471 DEPs, Coronin 7 emerged as our top-ranked candidate, given its well-studied role in Golgi export. Several studies have shown that mammalian CRN7 is a cytosolic protein that is recruited to the cytoplasmic side of Golgi membranes where it promotes normal Golgi organization and biogenesis of TGN-derived transport carriers required for Golgi export (Bhattacharya et al., 2016⁴⁰; Rybakin, 2008³⁵; Rybakin et al., 2004, 2006, 2008^{33,34,39}; Yuan et al., 2014³⁶). Therefore, the downregulation of CRN7 in *UBR4* KO HEK293T cells seemed to us to be a plausible explanation for the impaired Golgi exit phenotype – a hypothesis that we pursued further by conducting several rescue experiments, as described in the manuscript.

The well-documented roles of CRN7 in Golgi export are highlighted in the following excerpts taken from the results and discussion sections of the manuscript:

“To identify a candidate mediator of the trafficking phenotype in UBR4 KO cells, we carefully evaluated the list of DEPs for proteins that had previously been implicated in Golgi export. Coronin 7 (CRN7), a protein that binds to the cytosolic side of Golgi complex membranes, has been shown to mediate cargo export from the Golgi and is required for normal Golgi morphology^{33–35”}

“Mechanistically, we show that UBR4 facilitates Golgi export by supporting the high expression of CRN7. Mammalian CRN7 is a cytosolic protein that is recruited to the cytoplasmic side of Golgi membranes where it promotes normal Golgi organization and biogenesis of TGN-derived transport carriers required for Golgi export^{33–36,39,40”}

Ubr4 mutants have lowered amounts of CRN7 yet CRN transcription and protein levels appear normal. To their credit, the authors next tested potential regulation of CRN7 translation and claim to find evidence for such. The data on this point then assumes importance as providing the mechanism to explain the authors’ principle hypothesis. Regarding the design of and actual data from this experiment, I have the following three concerns:

a. I found no statistical analysis of the results.

We have included statistical analysis (2-way ANOVA) of these results in the revised manuscript. Fractions C and E are significantly different between *UBR4* WT and KO cells in terms of *CRN7* transcript abundance.

b. Further, if GAPDH and CRN7 are statistically different (which is at least implied), what should we understand from that? What is the actual consequence for translation rates if RNA is enriched in one density domain versus another? Knowing little about this subject I was hoping for guidance, yet authors offer only the following:

468 “Future experiments are needed to confirm the effects of UBR4 ablation on the translation of CRN7 and to establish the precise molecular mechanisms.”

GAPDH mRNA sedimentation is not different between *UBR4* WT and KO cells, as we expected based on the fact that it is a housekeeping gene and its expression should not differ between the two genotypes. On the other hand, *CRN7* mRNA sedimentation is statistically different between *UBR4* WT and KO cells, with the transcript being preferentially associated with the higher polysome fraction in WT cells relative to *UBR4* KO cells. Note that the polysome gradient separates the mRNAs based on the number of ribosomes loaded on them: therefore, the higher polysome (i.e., heavier) fractions contain more ribosomes bound to each mRNA. This suggests that the translation rate of *CRN7* is higher in WT cells compared to *UBR4* KO cells. The significance of the *GAPDH* and *CRN7* results are summarized in this sentence: “This result suggests that the rate of translation of *CRN7* is **selectively** reduced in the absence of *UBR4*.”

c. Finally, GAPDH levels (the control RNA) appear the same across the different polysome fractions in either genotype, while UBR4 levels appear different across polysome fractions according to genotype – what is the significance of that? I would have thought the control RNA should also display different proportions across different polysome fractions in order to serve as a proper point of comparison. Perhaps the control RNA is not well chosen?

Housekeeping genes such as *GAPDH* and *ACTIN* are commonly used as control RNA in polysome fractionation studies. Although we did expect to observe more variation in *GAPDH* sedimentation across different fractions, by collapsing 14 fractions into 5 pooled fractions (to increase the manageability of downstream cDNA synthesis and qPCR reactions), this may have the unintended effect of diluting/obscuring the differences in individual fractions. Nevertheless, it is important to appreciate that the sedimentation of *CRN7* is markedly different between *UBR4* WT and KO cells, and this corresponds with the observed decrease in *CRN7* protein levels in the KO cells.

REVIEWERS' COMMENTS

Reviewer #3 (Remarks to the Author):

The authors have addressed all my concerns with thoughtful responses and revisions to the text.